# A single-cell atlas enables mapping of homeostatic cellular shifts in the adult human breast

Austin D. Reed[1,2,10], Sara Pensa[1,2,10], Adi Steif[3], Jack Stenning [1,2],
Daniel J. Kunz [4], Linsey J. Porter[1,2], Kui Hua [3], Peng He[4,5],
Alecia-Jane Twigger[1,2], Abigail J. Q. Siu [1,2], Katarzyna Kania [3],
Rachel Barrow-McGee[6], Iain Goulding[6], Jennifer J. Gomm[6], Valerie Speirs[7,8],
J Louise Jones[6], John C. Marioni [3,4,5,9] ✉ & Walid T. Khaled [1,2] ✉

Here we use single-cell RNA sequencing to compile a human breast cell atlas assembled from 55 donors that had undergone reduction mammoplasties or risk reduction mastectomies. From more than 800,000 cells we identified 41 cell subclusters across the epithelial, immune and stromal compartments. The contribution of these different clusters varied according to the natural history of the tissue. Age, parity and germline mutations, known to modulate the risk of developing breast cancer, affected the homeostatic cellular state of the breast in different ways. We found that immune cells from *BRCA1 or BRCA2* carriers had a distinct gene expression signature indicative of potential immune exhaustion, which was validated by immunohistochemistry. This suggests that immune-escape mechanisms could manifest in non-cancerous tissues very early during tumor initiation. This atlas is a rich resource that can be used to inform novel approaches for early detection and prevention of breast cancer.

One of the biggest challenges in treating breast cancer is the heterogeneous nature of the disease[1]. We have limited understanding of how early divergences from the homeostatic breast subtypes lead to tumor heterogeneity. While large-scale cancer genomic studies indicate that different breast cancer subtypes are enriched for certain mutations[2], not all phenotype variability and tumor behavior can be explained by mutations alone. Studies in mice have shown that the cell-of-origin may contribute to the phenotype of the resulting tumor[3]. Defining which cell type leads to which kind of tumor is complicated by a growing list of environmental and epidemiological risk factors, such as age, which not only affect overall incidence, but also outcome[4]. Such risk

factors can themselves be modulated by other factors; for example, the age-dependent risk in breast cancer is greatly reduced by pregnancy early in life[5], whereas predisposing germline mutations (for example, *BRCA1* or *BRCA2*) greatly increases age-associated risk[6]. How these risk factors interact and impact tissue homeostasis remains to be fully understood. To do this, it is crucial to characterize cell types and cellular states present under different physiological conditions.

To tackle this problem, studies in the mouse have leveraged single-cell genomics (that is, single-cell RNA sequencing (scRNA-seq)) to determine the gene expression profile of individual mammary epithelial cells across embryonic and adult developmental stages[7–10].

[1]Department of Pharmacology, University of Cambridge, Cambridge, UK. [2]Wellcome-MRC Cambridge Stem Cell Institute, University of Cambridge, Cambridge, UK. [3]CRUK, Cambridge Institute, University of Cambridge, Cambridge, UK. [4]EMBL European Bioinformatics Institute, Hinxton, UK. [5]Sanger Institute, Hinxton, UK. [6]Breast Cancer Now Tissue Bank, Centre for Tumour Biology, Barts Cancer Institute, John Vane Science Centre, Queen Mary University of London, London, UK. [7]Institute of Medical Sciences, University of Aberdeen, Aberdeen, UK. [8]Aberdeen Cancer Centre, Aberdeen, UK. [9]Present address: Genentech, San Francisco, CA, USA. [10]These authors contributed equally: Austin D. Reed, Sara Pensa. ✉e-mail: marioni@ebi.ac.uk; wtk22@cam.ac.uk

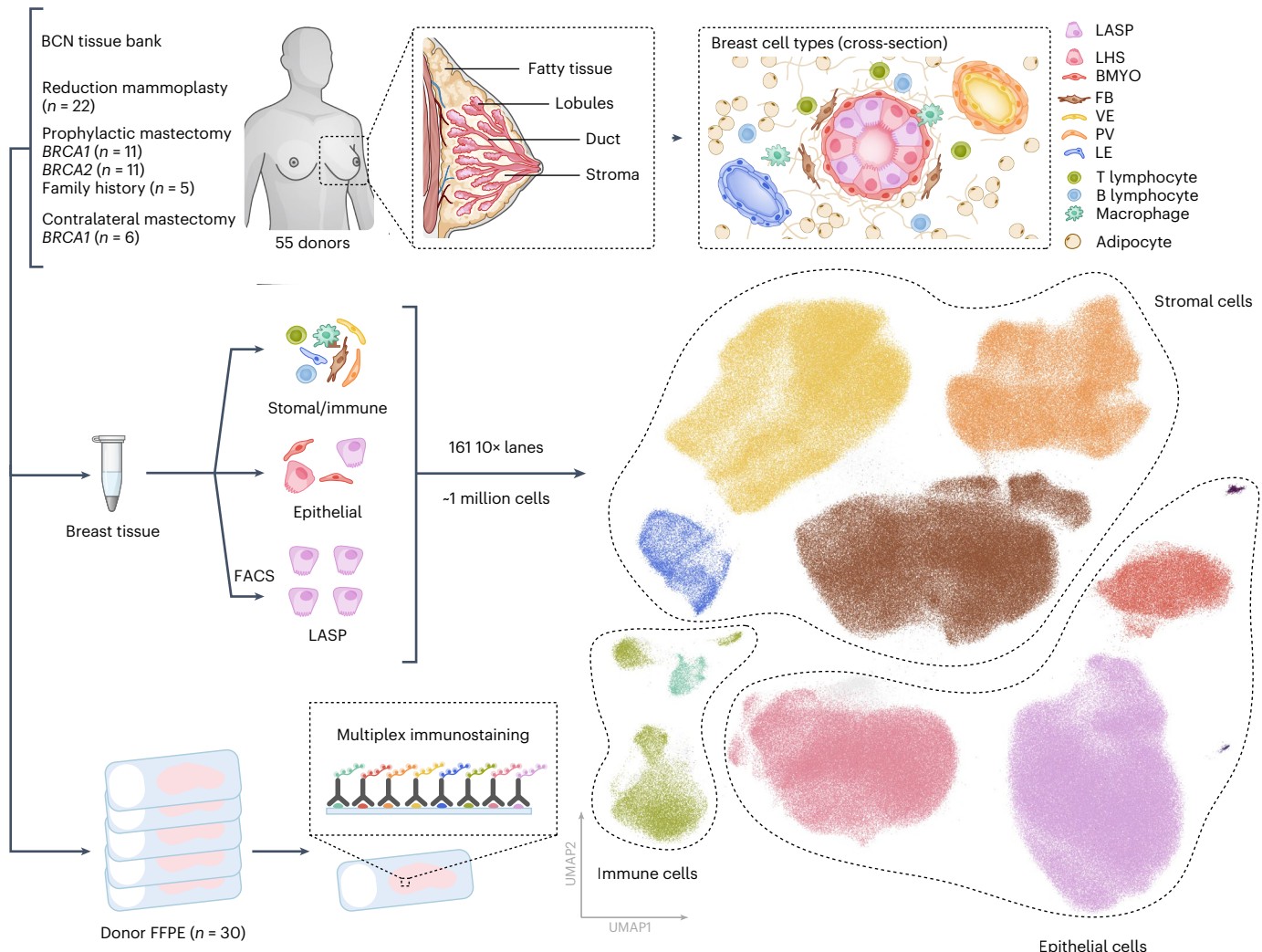

**Fig. 1 | Overview of the HBCA.** A schematic highlighting the overall experimental design and the cell types we aimed to capture in the atlas. The diagram highlights the overall number of donors sequenced and how they are distributed among the various subgroups alongside the number of donor FFPE tissues that have been analyzed by multiplex immunostaining. The global UMAP representation of the final dataset is colored by general cell types captured from all 161 samples processed as part of this atlas.

Similar to the mouse, after birth the human mammary gland continues to develop. At birth the human breast consists of a ductal structure with well-defined terminal ductal lobular units similar to those found in the adult[11]. Before puberty, the breast grows in proportion to the rest of the body. The onset of puberty triggers the expansion of the terminal ductal lobular unit to form adult lobules and to fill the surrounding stroma. Consisting of the vasculature system, fibroblasts and immune cells, the stromal compartment remains severely understudied[12]. Despite this, mounting evidence suggests that changes in the microenvironment are a major contributing factor in tumorigenesis and an important target for novel therapeutics[13]. Together, these motivate the study of the complex interactions between epithelial cells and the surrounding stromal.

Recent single-cell transcriptomic and proteomic studies of human reduction mammoplasty and tumor samples have begun to catalog the various epithelial compartments in the breast[12–20]. However, larger sample sizes, encompassing major developmental changes and risk-modulating factors, are needed to generate a comprehensive transcriptomic map of all adult breast cell subtypes and states. Here we report the use of scRNA-seq to compile a comprehensive Human Breast Cell Atlas (HBCA) of over 800,000 cells collected from 55 women across various adult development stages.

## Results

### Identification of a suitable tissue cohort

Here we present a comprehensive HBCA, enabling us to map how cellular composition changes as a function of various biological and environmental factors. To assist study of a wide range of these factors we identified a cohort of healthy breast tissue samples from the Breast Cancer Now Tissue bank that had (up to 80) health and lifestyle records available (Fig. 1 and Supplementary Table 1). Our cohort consisted of tissue samples from 22 women who had undergone reduction mammoplasties, 27 women carrying a *BRCA1* or *BRCA2* mutation or had a strong family history of breast cancer that was not attributed to known risk genes (who had risk reduction prophylactic mastectomies), and 6 women who had undergone contralateral mastectomies from *BRCA1* mutation carriers that had breast cancer in one breast and had the second breast removed to reduce the risk of further tumors (Fig. 1). The samples had a wide distribution of values across the various risk modifiers such as age, parity status and menopause. Additionally, we collected a range of metadata on further risk-modifying factors that may prove beneficial to future studies (Extended Data Fig. 1 and Supplementary Table 1). All samples were pre-processed by the tissue bank to isolate the epithelial and stromal/immune-enriched compartments (Fig. 1). These were then processed to single-cell level and viable cells

were loaded on a 10× chromium chip for single-cell capturing to enable scRNA-seq. For the epithelial-enriched compartment, we also used fluorescence-activated cell sorting (FACS) to enrich for luminal progenitor cells (EPCAM+ and CD49f+)[21] (Supplementary Figs. 1–3), which have been proposed to be the cell of origin for some breast cancers[3,21].

Following sequencing, more than one million cells were identified. After several quality control and computational doublet calling steps, 801,360 cells were taken forward for downstream analysis (Fig. 1 and Supplementary Figs. 4, 5). To combine our 45 separate sequencing batches we used scVI[22] to produce a batch corrected embedding for use in downstream analysis. More details on the sample preparation and analysis can be found in the methods section. Coarse cell-type annotation revealed that we sequenced over 350,000 epithelial cells, and 400,000 stromal and immune cells (Fig. 1).

### Major cell subtypes identified in the Human Breast Cell Atlas

Within the epithelial compartment we used canonical lineage markers to identify three major cell types: the luminal adaptive secretory precursor (LASP; also known as luminal progenitor, see Supplementary Table 2 for nomenclature summary), luminal hormone sensing (LHS), and basal-myoepithelial (BMYO) (Fig. 2). Iterative clustering identified several subclusters within each cell type based on unique gene expression profiles. The majority of these subclusters were found in all 55 donors, albeit at varying proportions (Supplementary Fig. 6). Of note, we found no distinct LASP cluster strongly marked by milk biosynthesis genes (*CSN2/3*, *LALBA*; Fig. 2b). Instead, we find most LASP heterogeneity is defined by proportions of standard marker gene expression. LASP1/2/3 show high expression of LASP specific markers (*ALDH1A3*, *SLPI*) in contrast to LASP4. Additionally, LASP2 is notable for its co-expression of both LASP (*ALDH1A3*) and BMYO markers (*KRT14*, *KRT5*) (Fig. 2b and Supplementary Fig. 7a,c). We also identified LASP5 as a small population of proliferating cells marked by the canonical proliferation marker *MKI67* and mitosis related genes (*AURKB*, *TOP2A*; Fig. 2b and Supplementary Fig. 7a). Similarly, within the LHS compartment, upregulation of *AREG* marked LHS1 cells, while upregulation of estrogen and progesterone receptors (*ESR1* and *PGR*) marked LHS2 cells. Subcluster LHS3 shows an expression pattern distinguished by high *SERPINA1* and *PIP* expression alongside increased expression of some LASP markers (*ALDH1A3*, *SLPI*; Fig. 2b and Supplementary Fig. 7b). The two BMYO subclusters are distinguished by expression of canonical BMYO markers *KRT5* and *KRT14* (high in BMYO1) as well as *OXTR* (high in BMYO2; Fig. 2b and Supplementary Fig. 7c).

Within the stromal compartment we identified four fibroblast (FB1–4), four vascular endothelial (VE) cell subclusters including VE venous (VEV), VE capillary (VEC), VE arterial (VEA) and VE angiogenic tip (VEAT) cells, two lymphatic endothelial cell (LE1–2) and five perivascular (PV1–5) subclusters, which were present across all donors (Fig. 2 and Supplementary Fig. 6). FB1 shows increased expression of genes related to extracellular matrix formation (*DCN*, *LUM* and *COL1A2*), whereas FB2 is marked by several genes related to extracellular matrix disassembly (*MMP2*, *MMP3*, *MMP10 MMP12* and *SH3PXD2B*). We found FB3 to generally possess a quite distinctive transcriptional profile compared with FB1, 2 and 4 subclusters and was marked by high *CLU* and *GREM1* expression (Fig. 2b and Supplementary Fig. 8a–c). The venous, arterial, capillary and angiogenic tip endothelial subclusters were identified using a range of previously described marker genes[23–25]. The lymphatic endothelial cell compartment, distinguished by canonical markers *CCL21* and *PDPN*, splits into two groups that can be separated by their expression of chemokines (*CXCL1* and *CXCL8*) and some angiogenic tip cell markers (*ANGPT2* and *PXDN*), both of which are highly expressed in LE1, but lowly expressed in LE2 (Supplementary Fig. 8b). The perivascular subclusters appear in two main groups, with PV1/2 having low expression of genes relating to muscular functions (*ACTA2* and *TAGLN2*) and pericyte markers (*RGS5*)[26], which are high in PV3/4/5 (Supplementary Fig. 8c). Using previously described immune

cell markers we were also able to identify a variety of lymphoid and myeloid cell types including five T cell subtypes, natural killer T (NKT) cells, natural killer (NK) cells, innate lymphoid cells (ILC), three B cell subtypes, plasma cells, dendritic cells and two macrophage subclusters including lipid-associated macrophages (Macro-lipo) (Fig. 2 and Supplementary Fig. 9a–c; see Supplementary Table 2 for cell-type nomenclature summary). To provide robust 100-gene signatures for each identified subcluster, we used pseudobulk one-versus-all differential expression testing to identify an extensive list of marker genes. These lists show high specificity to our cell-type and subcluster annotations and offer many cell-type markers for use of the community (Supplementary Fig. 10 and Supplementary Tables 3 and 4). We found all major cell types represented in all donors regardless of *BRCA1* or *BRCA2* mutation status, age or parity (Supplementary Fig. 6).

Our subclustering also identified two donor derived clusters (DDC), each consisting primarily of cells from just one donor (donor 17 and donor 37 respectively, Fig. 2a). Both DDC1 and DDC2 show high expression of LASP marker genes (*ALDH1A3* and *SLPI*, but not *KIT*, Fig. 2b). Despite this, they also show a mix of other epithelial lineage markers from LHS and BMYO cell types. Additionally, DDC1 shows particularly high expression of *MMP3*, as well as *PIP* and *MUCL1*, which were otherwise predominantly expressed in LHS3 within the epithelium (Fig. 2b). Despite the expression of mixed lineage markers, both DDC1 and DDC2 were not associated with batch, have low doublet scores and typical quality control metric distributions (Supplementary Fig. 7d). We carried out an additional quality control step to distinguish between single nuclei (stripped nuclei) from single cells using gene signatures described in a previous study[27]. This identified several such stripped nuclei clusters (colored gray in uniform manifold approximation and projections (UMAPs)); however, these did not overlap with the DDCs, suggesting that the DDCs are indeed single cells (Supplementary Figs. 7e, 8e and 9e). Together, this supports the DDC as genuine and distinct donor-specific cells.

Due to the mixed marker expression and rare patient specific nature of the clusters we then investigated the possibility these cells show early signs of transformation. To investigate the similarity with known tumor signatures, we looked at the four breast cancer subtype gene scores across our epithelial subclusters[28]. Both DDC clusters stood out from the remaining epithelial subclusters showing elevated gene scores for HER2+ and basal-like subtypes (Extended Data Fig. 2). To explore this further, we used inferCNV[29], which compares RNA expression binned across the genome between two groups of cells to predict copy number variations (CNVs) at a single-cell level. To maximize robustness against cell-type and donor-specific expression patterns across the genome, we used a reference set covering a range of epithelial, stromal and immune cells sampled from different mammoplasty donors alongside the non-epithelial cells of the DDC donors. The results of this analysis identified prominent CNV profiles in both DDC1 and DDC2 subclusters (Extended Data Fig. 3a). This analysis identified a range of deletions in chromosomes 3 and 10, duplications in chromosomes 1, 11, 12 and 17 as well as evidence of chromosome 19 duplication. Many of these aberrations were shared between DDC1 and DDC2. We additionally found a range of predicted duplications that appeared subcluster specific with DDC2 showing distinctive duplications on chromosomes 17 (containing *ERBB2*) and 22 (Extended Data Fig. 3a). When comparing the profiles of DDC1 and DDC2 to previously published CNV profiles of the four tumor subtypes, particularly luminal A and triple-negative breast cancer (TNBC), we observe several similarities, including the amplifications observed in chromosome 1 and duplication of chromosome 19 (ref. 30).

### Changes in healthy breast composition with age and parity

We first consider how natural factors such as age and parity impact the composition of the breast. To avoid confounding our results with changes resulting from differences in *BRCA1* or *BRCA2* mutation status,

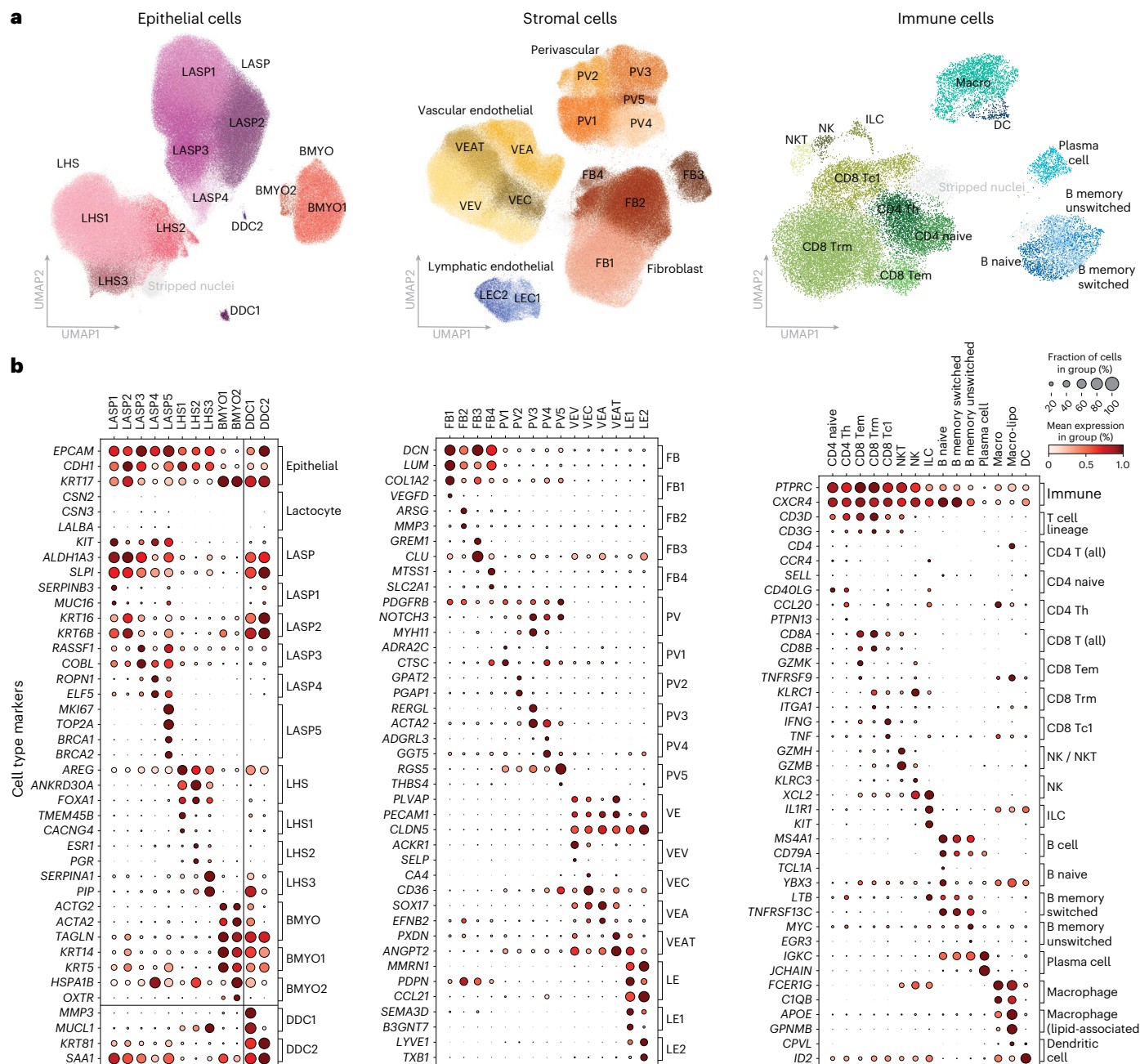

**Fig. 2 | Major cellular groups identified in the HBCA. a,** UMAP plots of the epithelial, stromal and immune cell compartments colored and labeled by subcluster annotations. Doublets/stripped nuclei are plotted in gray. **b,** Dot plots summarizing the known markers used to identify cell types and a selection of genes distinguishing each subcluster for the epithelial (left), stromal (middle) and immune (right) compartments. Each column corresponds to a specific cell subcluster and the rows correspond to a list of key marker genes (expression normalized per gene), brackets on the right side of each plot detail the cell type or subcluster that these genes mark.

we exclusively consider the 22 reduction mammoplasty donors. To gain high-resolution insight into cellular differential abundance and compositional shifts, we used Milo differential abundance testing[28]. This approach first groups cells into small overlapping neighborhoods based on similar gene expression (Methods). Then, in each neighborhood, we look for statistically significant enrichment of cells from each condition tested using flexible generalized linear models, thereby avoiding both statistical confounding and the inherent biases placed by fine cell-type clustering[31]. We found that most of the significant changes driven by age occur within the epithelial compartment. In the LASP compartment, older samples display an enrichment of LASP2/4 cells, while regions of the LASP1/3 subclusters are enriched in younger

individuals (Fig. 3a,b). The LASP clusters enriched in older women have reduced expression of traditional LASP markers, consistent with previous studies showing increased LASP lineage infidelity with age[15,32]. In the LHS compartment, we found an enrichment most prominently in the LHS2 and to a lesser extent LHS3 subclusters with age, while in the BMYO compartment we found a significant decrease in the proportion of BMYO1 cells (Fig. 3a,b). Outside the epithelial compartment, there are few significant differences with age except for an enrichment in a subset of FB1 cells and the FB3 subcluster, as well as depletion of plasma cells (Fig. 3a,b).

In contrast, parity has a more widespread impact on the cellular composition of the epithelial, stromal and immune compartments

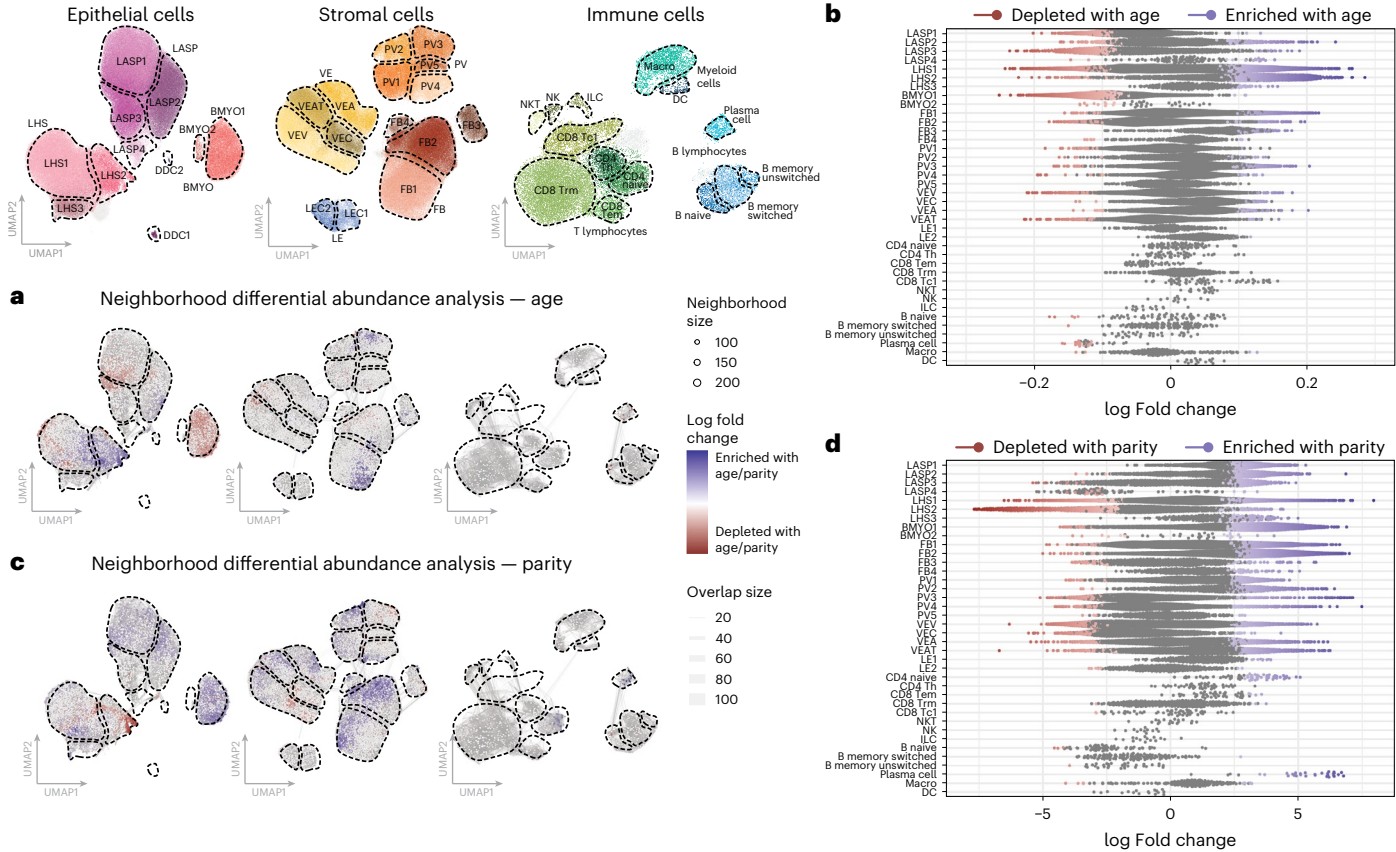

**Fig. 3 | Age and parity affect the homeostatic cellular state of the breast.**
**a**, Milo cell neighborhood differential abundance plots of the significant
(FDR <0.05) changes in the breast composition with age, blocking for the
effects of parity (mammoplasty donors; $n$ = 22). We test the effects of age as
a continuous scale ranging from 19 to 65 years, with the color gradient scale
representing log fold changes per year. Blue represents enrichment with age
while red denotes depletion with age. **b**, Beeswarm plot of the log fold changes
in the Milo neighborhoods with age, grouped into each cell-type subcluster.
Neighborhoods with a significant change in cellular abundance are colored
as indicated. Log fold changes are per year due to the continuous age scale tested.
**c**, Milo cellular neighborhood differential abundance plots of the significant
(FDR <0.05) changes in the breast composition with parity (that is, nulliparous
versus parous), blocking for the effects of ageing (mammoplasty donors; $n$ = 22).
Blue represents enrichment with parity while red denotes depletion with parity.
**d**, Beeswarm plot of the log fold changes of the Milo neighborhoods with parity,
grouped into each cell-type subcluster. Neighborhoods with a significant change
in cellular abundance are colored as indicated.

(Fig. 3c,d), which could reflect the large-scale tissue remodeling of
the breast that occurs during pregnancy. In the LASP compartment,
there is a decreased proportion of LASP4 cells as a function of parity,
despite enrichment for the LASP1/2/3 subclusters in parous women. In
the LHS and BMYO compartments, there is an enrichment of LHS3 and
BMYO1 subclusters in parous donors. We note that many of the changes
in cellular abundance seen within the epithelial compartment are in
the opposite direction when considering age or parity as the covari-
ate, which could contribute to contrasting breast cancer risk posed
by each. We see some evidence of this with overall increased propor-
tions of plasma, CD8 $T_{EM}$ and CD4 T cell types in the profiled parous
donors. In the stromal compartment, our findings suggest a transition
from *ACTA2* expressing PV3/4/5 toward PV1/2-type perivascular cells
as a function of parity, as well as increases in proportions of vascular
endothelial arterial (VEA) and vascular endothelial angiogenic tip
(VEAT) cells and an overall increase in most fibroblast (FB) subclusters
excluding FB3 (Fig. 3c,d).

**Impact of high-risk *BRCA1* and *BRCA2* germline mutations**
To determine mammary cell state shifts in high-risk (HR) donors com-
pared with average-risk (AR) donors, we analyzed cell profiles from
reduction mammoplasty donors ($n$ = 22) and donors with *BRCA1* ($n$ = 11)
and *BRCA2* ($n$ = 11) germline mutations (denoted HR-BR1 and HR-BR2,
respectively). First, we tested for differential expression between

AR- and HR-BR1, as well as between AR- and HR-BR2 cohorts, accounting
for the effects of both age and parity (see Methods for details, Extended
Data Fig. 4 and Supplementary Tables 5–10). Despite similar statistical
power, we noted that HR-BR1 epithelial cells have more significant tran-
scriptional changes than the respective HR-BR2 cells. This is particularly
clear in the BMYO compartment, which showed only one (*HLA-DQB1*)
significantly upregulated gene in HR-BR2 cells compared with the AR
cells. Of note in the LASP compartment, we see evidence for the upregu-
lation of milk biosynthesis (*CSN2/3*, *CSN1S1* and *LALBA*) in HR-BR1 LASP
cells particularly (Extended Data Fig. 4a,g–j). The upregulation of milk
biosynthesis genes is similarly interesting to tumor development, with
recent mouse studies proposing these genes as possible markers of
pre-tumor progression[10,29]. These results suggest a similar process may
also characterize early signs of malignant progression in the human
breast. Unsurprisingly, the expression of milk biosynthesis genes also
has a strong relationship to parity, with a strong upregulation in parous
donors regardless of BRCA status (Extended Data Fig. 4g–j).

To explore other cellular differences, particularly in the immune
and stromal compartment, we use Milo to compare AR reduction
mammoplasty donors against HR-BR1 or HR-BR2 donors and iden-
tify compositional changes that may contribute to the higher risk
of breast cancer development. To allow easy comparisons between
HR-BR1 and HR-BR2 induced changes we summarized this in a 'Milo
signature' plot averaging neighborhood log fold changes per donor

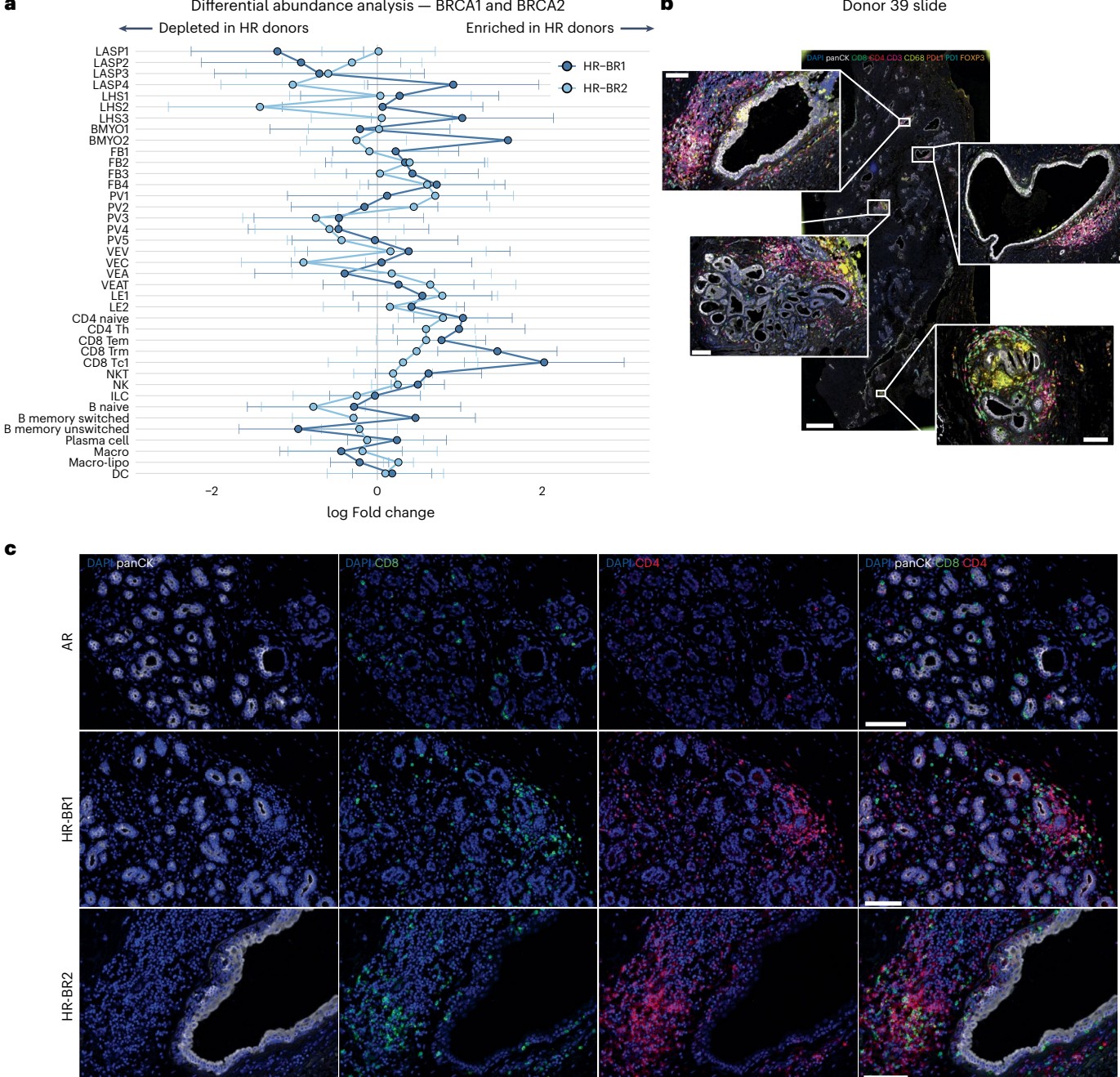

**Fig. 4 | HR-BR1 and HR-BR2 germline mutations associated with increased proportions of immune cells in the breast. a**, Summary plot of average Milo differential abundance results comparing the cellular composition of the breast from AR donors against HR-BR1 and HR-BR2 donors, blocking for the effects of both age and parity. To enable fair comparisons, neighborhoods were computed to be shared by AR/HR-BR1 and AR/HR-BR2 tests. The plot summarizes the mean and variance of log fold changes for each cell subcluster in both the HR-BR1 and HR-BR2 cohorts. A zero log fold change represents the same subcluster proportions as AR donor tissue on average, while positive (negative)

log fold changes denote subcluster enrichment (depletion) in the relative HR donor breast. See Extended Data Fig. 5 for full Milo results in HR comparisons. **b**, Example of whole section output using the Ultivue staining technology. DAPI, panCK, CD8, CD4, CD3, CD68, PDL1, PD1 and FOXP3 staining with colors indicated in figure are shown on a whole section and magnified insets from a representative donor. Scale bars, 100 μm in the insets, 2 mm in the whole section image. **c**, Ultivue staining showing panCK (white), CD8 (green), CD4 (red) and DAPI (blue) of breast sections from representative AR ($n = 10$), HR-BR1 ($n = 11$) and HR-BR2 ($n = 8$) donors as indicated. Scale bars, 100 μm.

for each subcluster (Fig. 4a and Extended Data Fig. 5). Both HR-BR1 and HR-BR2 donors (though more prominent in the latter) show shifts toward the VEAT and LE1 populations (Fig. 4a and Extended Data Fig. 5). However, one of the most significant changes in the HR cohorts is the large increase in the proportion of lymphocytes, particularly CD4, CD8 and NK/NKT cells (Fig. 4a). The most prominent changes are a twofold enrichment in the HR-BR1 donors of CD8 $T_{C1}$ cells, which are

characterized by high *IFNG* and *TNF* expression—both of which are known to be pro-inflammatory (Figs. 2b and 4a). To confirm this observation we obtained matched slides from 30 of the sequenced donors to perform orthogonal immunofluorescence validation (see Methods for details). This allowed us to assess up to eight markers simultaneously on tissue sections from samples taken from AR donors, and HR-BR1 and HR-BR2 donors (Fig. 4b). In agreement with Milo, we observe

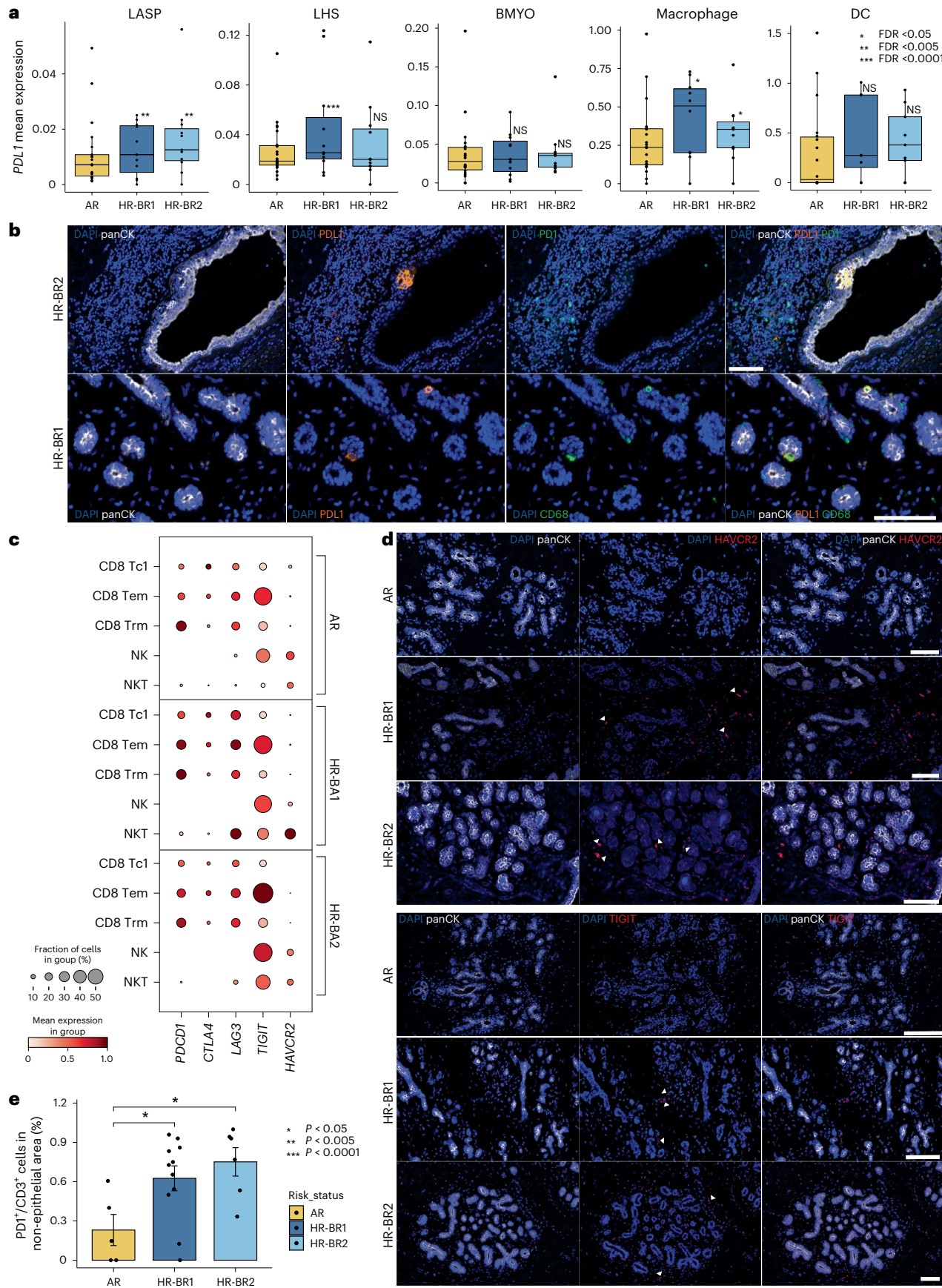

**Fig. 5 | HR-BR1 and HR-BR2 donor breast tissue display increased expression of immune checkpoint inhibition and exhaustion markers. a**, Boxplots visualizing the change in mean expression (log-transformed counts) for immune checkpoint ligand PDL1, identified as significantly upregulated in HR-BR1 (LASP, LHS and macrophage) and HR-BR2 (LASP and macrophage) donors with respect to AR controls. FDR values derived from edgeR pseudobulk differential gene expression testing for AR versus HR-BR1 and AR versus HR-BR2 (n = 33 in both comparisons; see Methods and Extended Data Fig. 4 for more details). The boxplot centers show median values while the minima/maxima show the 25th/75th percentiles, respectively, and whiskers extend to the most extreme datapoint within 1.5× interquartile range of the outer hinge of the boxplot. **b**, Top: Ultivue staining showing panCK (white), PDL1 (orange), PD1 (green) and DAPI (blue) of a breast section from an HR donor showing an example of epithelial expression of PDL1. Bottom: Ultivue staining showing panCK (white), PDL1 (orange), CD68 (green) and DAPI (blue) of a breast section from an HR donor showing examples of CD68 cells expressing PDL1. Scale bars, 100 μm.

**c**, Dot plot displaying the expression of several key immune checkpoint receptors expression in a range of cytotoxic lymphoid subclusters comparing between AR, HR-BR1 and HR-BR2 donor groups. Expression is normalized per gene. **d**, Top: immunofluorescence staining showing panCK (white), HAVCR2 (red) and DAPI (blue) of representative breast sections from AR and HR donors. Bottom: immunofluorescence staining showing panCK (white), TIGIT (red) and DAPI (blue) of a representative breast section from AR and HR donors. Arrowheads point at examples of positive cells. Each staining is representative of two (AR, HR-BR2) or three (HR-BR1) donor samples. Scale bars, 100 μm. **e**, Bar plot showing the percentage of PD1⁺/CD3⁺ double-positive cells located in non-epithelial epithelial areas (that is not intercalated with the epithelium) in HR-BR1/2 donor compared with AR donors in whole Ultivue slides (Methods). The P values are calculated with one-way non-parametric Wilcoxon tests for both AR versus HR-BR1 (n = 21, P = 0.02) and AR versus HR-BR2 (n = 18, P = 0.018) comparisons. Error bars show the standard error of the mean.

significantly increased proportions of CD8 positive cells (P < 0.05) alongside evidence of increased proportions of GZMH positive cells (NK/NKT) within regions of HR-BR1/2 donor tissue (Fig. 4c, Extended Data Figs. 6–9 and Supplementary Fig. 11).

A possible impact of this immune response was hinted to in the differential gene expression analysis of the three epithelial compartments (Extended Data Fig. 4a,b,d and Supplementary Tables 5–10). Further exploration of these results revealed increased expression of *PDL1* (also known as *CD274*) in HR-BR1/2 LASPs and macrophages as well as HR-BR1 LHS cells relative to AR controls (Fig. 5a). Despite *PDL1* expression being rare, we were able to identify PDL1⁺ epithelial and CD68⁺ cells in multiple HR donors using immunofluorescence (Fig. 5b and Supplementary Fig. 12). This observation led us to explore more immune checkpoint inhibitors and markers of immune cell exhaustion in our cohort, including *PDCD1* (also known as *PD1*), *CTLA4*, *LAG3*, *TIGIT* and *HAVCR2* (also known as *TIM3*)[33]. We see general increases in the frequency and level of expression across immune checkpoint/exhaustion genes in both HR-BR1 and HR-BR2 donors (Fig. 5c). There are modest increases in the expression of *PDCD1* (PDL1 receptor) itself in the CD8 $T_{EM}$, CD8 $T_{CI}$ and NKT (HR-BR1 specific) subclusters. We also see increased expression of *LAG3* and *TIGIT* in the CD8 $T_{CI}/T_{EM}$ and NK/NKT cell types of HR-BR1/2 donors alongside increased *HAVCR2* expression in NKT cells from HR donors (Fig. 5c). This is supported by immunofluorescence analysis showing evidence of increased exhaustion markers in matched HR donor tissue slides compared with AR controls (Fig. 5d, Extended Data Fig. 9 and Supplementary Figs. 13 and 14). Spatially, we found that immune cells marked by PD1 expression were preferentially localized outside the epithelium in HR-BR1 and HR-BR2 donors compared with AR counterparts (Fig. 5e). Collectively, these results suggest that, while there is increased immune activity in the HR donors, the immune system is showing signs of exhaustion and suppressed function in these tissues.

The Milo analysis also revealed several differences between the HR-BR1 and HR-BR2 donors. This can be seen in subclusters of the epithelium, where the HR-BR1 donors showed strong enrichments for LASP4, LHS3 and BMYO2 cell subclusters not shared with HR-BR2 donors. Instead, the HR-BR2 donors showed decreased LASP4 and LHS2 subclusters in comparison to AR donors (Fig. 4a).

## Comparison of annotations across single-cell breast studies

To facilitate the comparison of cell types and their subclusters defined within the HBCA cohort we integrated our data with six of the largest scRNA-seq studies[14–18,20] of the healthy breast to form the first integrated HBCA (iHBCA). The iHBCA includes both fresh and frozen tissue prepared using a range of different protocols across multiple labs totaling 2.1 million cells from 286 individuals (Fig. 6a). We used scVI to perform integration of the seven datasets to correct for any batch specific sequencing effects (see Methods for more details), which preserved the general cell-type structure from each of the seven studies (Extended Data Fig. 10a; see Supplementary Table 2 for nomenclature comparisons). This analysis highlighted a major lack of consensus in the cell nomenclature used across datasets. To address this, we utilized a CellTypist logistic regression classifier[30] (Methods), which we trained based on the subcluster annotations of our HBCA cohort. Using this model, we were able to assign an identity to each individual cell from all datasets, using our HBCA annotation as the reference. The resulting mapped cell-type annotations are visualized on the joint iHBCA UMAP (Fig. 6a). To quantitatively summarize the label mapping efficiency and nomenclature comparisons we created confusion matrices showing the proportion of cells from each original dataset label mapped onto the corresponding HBCA cell-type label (Fig. 6b). Overall, this shows strong concordance of cell-type annotations between datasets. In addition, we repeated the differential abundance testing on the iHBCA and confirmed many of the findings we reported earlier (Extended Data Fig. 10b).

To facilitate the usefulness of the iHBCA as a community resource, we have curated pretrained CellTypist models on the basis of the cell-type annotations provided by each dataset (see 'Data availability'). These models will enable other researchers to easily map iHBCA labels onto their own datasets using minimal computational resources. This stands as a comprehensive resource to compare cell-type and state annotations across different datasets.

## Discussion

In this study, we describe a scRNA-seq HBCA generated by sequencing over 800,000 cells from 55 donors. The scale of this dataset has enabled us to delve into the entire breast composition, encompassing not only the epithelium but also the surrounding microenvironment.

**Fig. 6 | The iHBCA provides quantitative comparison of cell-type annotations across seven of the largest scRNA-seq datasets for the breast. a**, A schematic showing the curation of the iHBCA combining seven of the largest scRNA sequencing datasets for the breast. The diagram highlights the composition and sample heterogeneity captured by the iHBCA. The central plot shows a global UMAP representation of the dataset colored by transferred subcluster annotations (Fig. 2) from the HBCA. Annotation labels were mapped using CellTypist logistic regression models (Methods). **b**, A set of six confusion matrices showing the cell type/subcluster comparisons between each of the published datasets and our own subcluster annotations. Each cell (row A, column B) shows the percentage of cells of type A in the original dataset that are mapped to cell type B in our HBCA subcluster annotations. Note: LC1/2 cells from the Twigger dataset are cells thought to appear only in the lactating gland and are hence absent from the HBCA cohort causing their nonsensical logistic regression mapping.

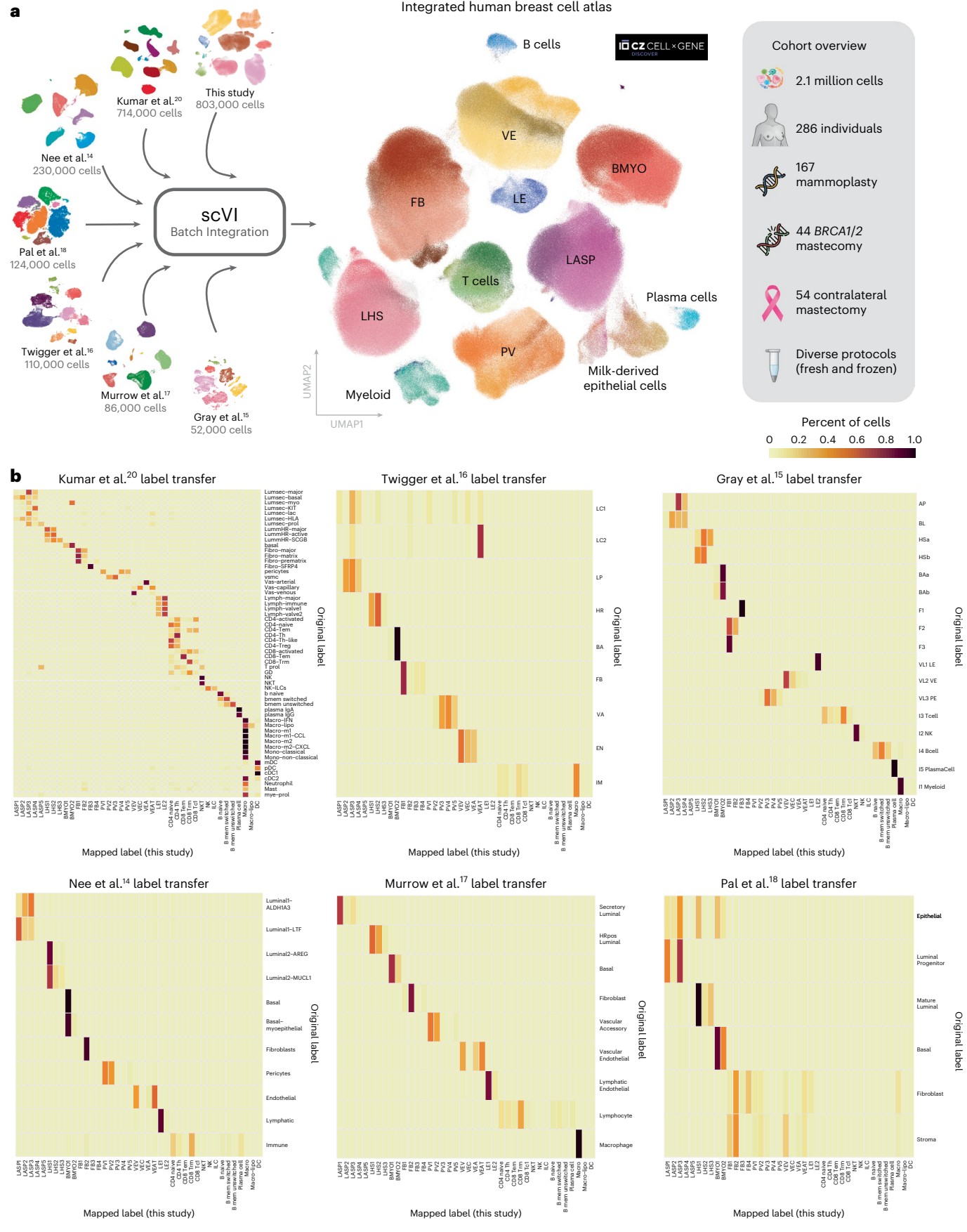

Due to the diversity of the samples, we were able to interrogate the data relative to several key breast cancer risk modifiers, such as age, parity and *BRCA1* or *BRCA2* germline mutations, enabling us to uncover cellular interactions and compositional changes associated with each factor.

As expected, both age and parity affect the homeostatic cellular state of the breast. Although the changes observed are not restricted to any one type of cell, we could identify unique features for the two risk factors. In contrast to parity, which has a widespread impact on breast composition, the main changes associated with age are concentrated within the epithelial compartment. Most notably, we observed an enrichment of the LASP2/4 clusters, which are characterized by mixed lineage expression and reduced fidelity toward traditional LASP (luminal progenitor) markers. This is consistent with previous observations of increased LASP lineage infidelity with age[15,32,34,35]. The global impact of parity can be seen across the epithelial, stromal and immune compartments. Many of the effects observed in the epithelium oppose those observed with age such as strong changes in BMYO1 and LHS2 proportions alongside more subtle general shifts in the total proportion of LASP cells. Outside the epithelium, we note increased proportions of plasma, CD8 $T_{EM}$ and CD4 T cells in parous women (Fig. 3). Overall, we found that the cellular composition changes observed with parity and age are complex and not restricted to any one type of cell. Thus, the impact of these changes needs to be considered collectively rather than in isolation given the contrasting impact of age and parity on breast cancer risk[4,36]. It would be interesting to explore the impact of other factors that will probably influence the homeostatic gland in combination with age and parity such as hormonal status or menopause, which due to limited sample size and metadata availability, could not be assessed in our study.

Once age and parity are accounted for, we do not identify cell populations that are exclusively associated with *BRCA1* or *BRCA2* germline mutation carriers, rather we observe shifts in the proportions of certain cell-type subclusters. One of the most diverse cell populations we captured in this study is the LASP compartment, which is also the subpopulation of epithelial cells most associated with breast cancer[21]. Some of the changes we observed here include decreased overall LASP proportions in both HR groups, with *BRCA1* samples having an enrichment of the LASP4 population, similar to that seen with age. Additionally, there was strong enrichment for the LHS3 and BMYO2 subclusters in HR-BR1 donors that was not seen in the HR-BR2 donors (Fig. 4a and Extended Data Fig. 5)

Some of the largest changes seen in our HR cohorts occurred in the immune compartment. In particular, we detected a significant immune expansion, prominently of the CD8 $T_{C1}$ cells in HR-BR1 donors, accompanied by increased expression of canonical immune checkpoint/exhaustion receptors in both HR cohorts (Figs. 4a and 5). A similar phenotype was reported in the Fallopian tube of *BRCA1* mutation carriers, suggesting commonalities across tissues[37]. Interestingly, when considering transcriptional shifts in HR donors compared with AR donors, we observe an increase in *PDL1* expression, mainly in LASPs and macrophages but also across some of the LHS cells (Fig. 5a,b and Supplementary Fig. 12). Previous studies have shown that high expression of *IFNG* in CD8 T cells, resembling what we observe in the CD8 $T_{C1}$ cell population, can directly drive *PDL1* expression in melanoma[38], suggesting that this expansion of CD8 $T_{C1}$ cells could be contributing to the *PDL1* induction in HR donors. The upregulation of *PDL1* could be particularly interesting due to its key role in immune evasion and supporting a tumorigenic microenvironment[39]. Given the observed increase in immune checkpoint/exhaustion markers in HR donor immune cells (Fig. 5b,c and Extended Data Fig. 9a–c and Supplementary Figs. 13 and 14) and the accumulating evidence of the involvement of LASPs in tumor initiation, our findings point toward an early epithelial immune-escape mechanism in the pre-malignant tissue driven by *BRCA1* and *BRCA2* germline mutations.

Within the stromal compartment, we found a strong enrichment of VEAT cells in HR donors, particularly HR-BR2 (Fig. 5a and Extended Data Fig. 5). Predictive cell–cell interaction analysis (Methods) suggests that SEMA3 and SEMA6 signaling pathways could be involved in mediating this enrichment (Supplementary Fig. 15). This agrees with a previous study where male *BRCA1* or *BRCA2* mutation carriers were found to have increased proportions of endothelial progenitor cells[40]. This warrants future investigations given the role of angiogenesis and vascular remodeling in tumorigenesis.

Finally, one of our primary objectives was to provide the community with a robust reference dataset that would enable seamless projection and integration with other datasets (Methods). As an example of this potential, we produced the iHBCA which integrates seven of the largest scRNA-seq datasets, representing a variety of sample types and tissue processing approaches (Fig. 6a). Overall, while we found strong concordance between cell types across datasets, the integration also revealed that the enrichment of the different cell types and subclusters varies greatly between datasets. We found that some of this variation could clearly be described by tissue storage and preparation differences between datasets. In particular, whether the cells were sequenced fresh following surgery or after collection from frozen tissue appeared to play a major role in the proportion of immune cell types and the transcriptional profile of the BMYO cells. We found that frozen tissue generally yielded more epithelial and stromal cells while fresh produced more immune cells (Extended Data Fig. 10c). Another difference worth noting for future studies is the number of cells sampled per individual, which varied greatly across the datasets (Extended Data Fig. 10d). Our study sequences the largest number of cells per individual, which allowed us to perform high-resolution differential abundance analysis through Milo as well as identify rare clusters in two of our donors (denoted DDC). Alongside mixed marker expression and increased tumor gene signatures, predictive CNV analysis suggested that these two clusters harbored genomic aberrations associated with early stages of tumorigenesis. Finally, the iHBCA presented here provides a framework for seamless integration of future sequencing datasets as we move toward larger cohorts that better represent the wider population.

## Online content

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

## Methods

### Human tissues

All frozen primary human female breast tissue was obtained from the Breast Cancer Now (BCN) tissue bank (REC 15/EE/0192) with collection done in compliance with all relevant ethical regulations. All participants' written, informed consent was provided to BCN. As all tissues were from female donors, findings apply only to female individuals and no sex-based analysis was performed. Metadata linked to the tissue bank samples was provided by BCN. Ethnicity was reported in Extended Data Fig. 1 and Supplementary Table 1 when available; however it was not taken into account in the data analysis as it was reported only for a fraction of the samples. All primary human breast tissue was derived from women undergoing reduction mammoplasties with no known genetic history ($n = 22$) and risk reduction prophylactic mastectomies from women with germline *BRCA1* or *BRCA2* mutations or other family histories ($n = 27$) and contralateral mastectomies from *BRCA1* mutation carriers that had breast cancer in one breast and had the second breast removed to reduce the risk of further tumors ($n = 6$). No specific age-range was selected. Participants were not specifically recruited for this study and are part of bigger cohorts where recruitment was not based on the parameters of interest for this analysis. Before freezing, fresh tissues were minced and digested overnight in 1 mg ml$^{-1}$ collagenase 1A and 1 mg ml$^{-1}$ hyaluronidase in an orbital shaker, washed two to three times and sedimented to allow for the separation between epithelial-enriched and stromal fractions, which were then cryostored.

### Batch design

Samples were divided into 45 batches (labeled by processing date in adata/sce object), where each batch represents a day in which two (for epithelial-enriched fractions) or four (for stromal-enriched fractions) samples were processed together. The batches were designed to minimise confounding with any of the main demographics of age and parity, or tissue conditions (surgery type or *BRCA1/2* mutation status). To achieve this, batches were designed to be either age or parity matched but with the corresponding tissue condition pseudorandomized.

### Mammary gland dissociation into single-cell suspension

Frozen vials of epithelial-enriched or stromal-enriched fractions were defrosted by gently diluting the material in 50 ml of cold tissue preparation medium (TPM, Roswell Park Memorial Institute 1640, 25 mM HEPES and 2 mM L-glutamine (Sigma R5886), 5% fetal bovine serum (FBS, Gibco), 100 units ml$^{-1}$ penicillin and 0.1 mg ml$^{-1}$ streptomycin sulfate (Gibco)) and washed in phosphate-buffered saline without calcium and magnesium (D8537, Sigma). Samples were centrifuged at 400 g for 5 min and resuspended in 2 ml of freshly prepared phosphate-buffered saline and 0.025% Trypsin, 0.1 g l$^{-1}$ EDTA (HyClone SV30031.01, Fisher Scientific) and 0.4 mg ml$^{-1}$ deoxyribonuclease 1 (DNase) (10104159001, Boehringer/Roche Diagnostics) previously warmed to 37 °C. Samples were then incubated at 37 °C with pipetting up and down for 30 s every 2–3 min until smoothly digested or up to a maximum of 10 min. Next, samples were washed in 40 ml of TPM and centrifuged for 20 min at 400 g with slow break. The pellet was resuspended by pipetting up and down in 200 µl of TPM and 10 µl of 10 mg ml$^{-1}$ DNase until homogeneous, then diluted in 25 ml of TPM and filtered through a 40 µm cell strainer (352354, Corning) into a 50 ml tube. After centrifugation for 5 min at 400 g, the pellet was resuspended by pipetting up and down in 200 µl of cell preparation medium (Roswell Park Memorial Institute 1640 plus 1% FBS, 100 units ml$^{-1}$ penicillin, 0.1 mg ml$^{-1}$ streptomycin sulfate) and 10 µl of 10 mg ml$^{-1}$ DNase until homogeneous, then washed in 3–6 ml of cell preparation medium. A total of 30,000 cells were resuspended into 48 µl of Hank's balanced salt solution (Gibco) and 1% FBS into low binding tubes. A total of 400 human mammary epithelial cells (Thermo Fisher Scientific, A10565) were added as spike-in, and samples were submitted for scRNA-seq (unsorted fraction). For the epithelial-enriched fraction only, the rest of the processed sample was stained with the following primary antibodies: CD45-APC (BioLegend, clone H130, 1:100, staining most hemopoietic cells), CD31-APC (BioLegend, clone WM-59, 1:100, staining endothelial cells), EPCAM-AF488 (BioLegend, clone 9C4, 1:50) and CD49f-PE/Cy7 (BioLegend, clone GoH3, 1 µg ml$^{-1}$, 1:200). DAPI was used to detect dead cells. Cells were filtered through a cell strainer (Partec) before sorting. Sorting of cells was done using a FACS Aria Fusion sorter and analyzed using FlowJo v10.9. Single-stained control cells were used to perform compensation manually and unstained cells were used to set gates. After doublets, dead cells and contaminating hematopoietic and endothelial cells (referred to as lineage) were gated out (Supplementary Fig. 1), up to 30,000 LASPs were sorted for scRNA-seq (with the addition of 400 human mammary epithelial cells as spike-in).

### scRNA-seq

Estimated equal numbers of cells per sample were processed for scRNA library preparation. Samples were processed for first-strand complementary DNA synthesis after sample defrosting and single-cell preparation. The remaining steps of library preparation were completed within the following 7 days.

### Library preparation and sequencing

scRNA-seq libraries were prepared in the Cancer Research UK Cambridge Institute Genomics Core Facility using the following: Chromium Single Cell 3′ Library and Gel Bead Kit v3, Chromium Chip B Kit and Chromium Single Cell 3′ Reagent Kits v3 User Guide (Manual Part CG000183 Rev C; 10X Genomics). Cell suspensions were loaded on the Chromium instrument with the expectation of collecting gel beads emulsions containing 8,000 single cells per channel. RNA from the barcoded cells for each sample were subsequently reverse transcribed in a C1000 Touch Thermal cycler (Bio-Rad) and all subsequent steps to generate single-cell libraries were performed according to the manufacturer's protocol with no modifications (13 cycles were used for cDNA amplification). cDNA quality and quantity were then measured using an Agilent TapeStation 4200 (High Sensitivity 5000 ScreenTape), after which 25% of material was used for gene expression library preparation.

Library quality was confirmed using an Agilent TapeStation 4200 (High Sensitivity D1000 ScreenTape to evaluate library sizes) and Qubit 4.0 Fluorometer (Thermo Fisher Qubit dsDNA LHS Assay Kit to evaluate double-stranded DNA quantity). Each sample was normalized and pooled in equal molar concentration. To confirm concentration, pools were analyzed with quantitative PCR using KAPA Library Quantification Kit on QuantStudio 6 Flex before sequencing.

Pools were sequenced on an Illumina NovaSeq6000 sequencer with the following parameters: 28 bp, read 1; 8 bp, i7 index; and 91 bp, read 2. Four S4 flowcells were run initially (ten samples per S4 lane) and further sequencing was added (depending on number of cells captured) aiming for 50,000 reads per cell across the whole dataset.

### Processing and quality control of scRNA-seq data

The same processing and analysis pipeline was used across all samples and all batches, using both R and Python with the conda/singularity environments specified for each step in the github repository. Read processing was performed using the 10X Genomics workflow. We used the CellRanger Single-Cell Software Suite (v6.0.2) to perform barcode assignment, demultiplexing and unique molecular identifier (UMI) quantification[41]. The reads were aligned to the 10X reference genome GRCh38 (ref-2020-A). To demultiplex our spike-in cells we used Vireo (v0.5.6)[42] genotyping with a reference lane of pure spike-in cells (sample SLX-20005-20446_SIGAE10), outputting the probability of being a spike-in or spike-in-doublet alongside the final classification as spike-in or not.

Barcodes that correspond to droplets with successfully captured cells were distinguished from empty droplets using the 'emptyDrops' function from DropletUtils (v1.12.1)[42] at a false discovery rate (FDR) of 0.001. Next, we performed various quality check steps to identify and remove low quality cells. We applied Scrublet (v0.2.3) in combination with an over-clustering approach by Pijuan-Sala et al. to computationally detect and remove doublets per sample[43,44]. After an initial rough filtering on low quality cells (remove any cell with UMIs <600 or mitochondrial content >15%), we further excluded cells with lower than 3 median-absolute deviance from the sample median for UMIs or counts, or those above 3 mean-absolute deviances in mitochondrial percentage. Additionally, we found sample SLX-19864-20262_SIGAA5, belonging to donor 50, to contain debris and have poor cell quality in general. Consequently, this sample was excluded from downstream analysis. Lastly, any spike-in cells were removed. After these filtering steps 803,283 cells were retained.

Single-cell analysis was performed using the scanpy pipeline (v1.8.2)[45]. Raw counts were log-normalized (scanpy.pp.normalize_per_cell(counts_per_cell_after = 1e4), scanpy.pp.log1p()) and subsequently used to select 5,000 highly variable genes (scanpy.pp.highly_variable_genes()). For dimensionality reduction and batch correction the raw counts of the highly variable genes were used to train the scvi-tools (v0.17.1) scVI VAE (parameters: batch_key = 'processing_date', n_latent = 20, gene_likelihood = 'nb', use_layer_norm = 'both', use_batch_norm = 'none', encode_covariates = True, dropout_rate = 0.2, n_layers = 2)[22,46]. The 20 latent dimensions were used to determine a KNN graph (scanpy.pp.neighbors(n_neighbors = 15)) for UMAP generation (scanpy.tl.umap()) and leiden clustering (scanpy.tl.leiden()).

## Clustering and annotation

We used multiple rounds of iterative leiden[47] clustering with scanpy.tl.leiden() to identify cellular partitions of our data. At each round we determined a new KNN graph and applied leiden clustering to the scVI batch corrected latent representation. This allowed us to label 'level0' and 'level1' annotations of the broad cellular compartments and general cell types. Then, we split our cells into multiple smaller groups, repeating the above steps to finalize the 'level2' subcluster annotations. Subclusters were labeled and identified using an array of known marker genes, quality control metrics and gene lists provided by 'scanpy.tl.rank_genes_groups'. We identified the optimal cluster resolution for each group by maximizing subcluster gene expression distinction, robustness and consistency (Supplementary Figs. 7–9). In some cases, small clusters of additional doublets were found, characterized by mixed lineage marker expression alongside an elevated mean scrublet scores. These cells were then (re-)labeled 'Doublets' at all levels of cell-type annotation. The original clustering, used in subsetting the data, was retained in the columns 'level0_global' and 'level1_global' for consistency.

## Differential expression and gene set enrichment analysis

Differential gene expression analysis was performed using edgeR (v3.36.0)[48]. A negative binomial generalized log–linear model was fitted to gene counts with the relevant subcluster or donor groups as the test covariate(s). The subcluster marker genes (Supplementary Fig. 10 and Supplementary Tables 3 and 4) used a one-versus-all approach across all samples (10× lanes) with the following model formula: '~ sample_type_coarse + subcluster_test' (subcluster_test being a Boolean for the subcluster being tested). Tests between AR- and HR-BR1/2 donors were completed separately in standard two group comparison approaches with samples again as the replicate with the following model formula: '~ sample_type_coarse + donor_age + parity + HR_test' ('HR_test' being a Boolean for the relevant HR donors, and 'parous' also being a Boolean for parity >0). For each the 'glmQLFTest' function was used to identify genes that have a log fold change significantly different from 0 at an FDR of 0.01.

## Milo and differential abundance analysis

To test for changes in cellular abundance at high resolution across various groups/conditions we used Milo's (v1.3.1) neighborhood abundance approach[31]. Due to the direct impact sorting methods have on cellular abundance we considered only unsorted (epithelial- and stroma-enriched) samples for differential abundance testing. For exploring age and parity, we considered the 22 (AR) mammoplasty donors (although one was excluded due to lack of parity metadata). We generated shared cell neighborhoods on the basis of the scVI corrected KNN graph for each of the epithelial, stromal and immune compartments. Specifically, we used the functions 'buildGraph' and 'makeNhoods' with parameters: 'k = 50', 'd = 20', 'prop = 0.3', 'refined=TRUE' and 'refinement_scheme = 'graph' in their respective inputs. The use of a graph refinement scheme significantly decreased the run time (and removed requirement to run 'calcNhoodDistance') when testing large numbers of cells and the remaining parameters were chosen on the basis of the recommended Milo summary plots and standards. The differential neighborhood abundance testing was completed using 'testNhoods' over samples with 'fdr.weighting = 'graph-overlap' and model formula '~ sample_type_coarse + milo_block + milo_test' with milo_block being donor_age/parous and milo_test being parous/donor_age respectively (here 'parous' is a Boolean for parity being greater than zero). To test for changes in HR donors compared with AR donors, a similar approach was taken while ensuring the neighborhoods were consistent across both AR- versus HR-BR1 and AR versus HR-BR2 tests to enable comparisons. The only differences arise in the model formula being: '~ sample_type_coarse + parous + donor_age + milo_test' where milo_test is a Boolean labeling the respective HR-BR1/2 samples. To create the 'Milo signature' summary we generated the average neighborhood log fold changes per donor for each subcluster. To account for the issue of overlapping neighborhoods, these averages were generated over the cells individually using the mean log fold change over all neighborhoods containing the cell.

## Cell–cell interactions

To predict and analyze the putative cell–cell interactions within our dataset we used Cell Chat (v1.6.0)[49]. Here we split cells into level2 subclusters (excluding the immune compartment which we grouped into lymphoid and myeloid lineages to simplify the analysis and increase readability). Separate Cell Chat objects were made for the AR, HR-BR1 and HR-BR2 cohorts to allow comparison between each. The communication probabilities were inferred using the 'computeCommunProb' function. Cell–cell communications for each cell signaling pathway were generated with the 'computeCommunProbPathway' function.

## Copy number inference

For predicting copy number aberration in donors 17 and 37 we used the inferCNV (v. 1.10.0) software package[50]. To map genes back to chromosomes we used the GRCh38 genome. In 'CreateInfercnvObject' we ignored chromosomes: 'GL000009.2', 'GL000194.1', 'GL000195.1', 'GL000213.1', 'GL000218.1', 'GL000219.1', 'KI270711.1', 'KI270713.1', 'KI270721.1', 'KI270726.1', 'KI270727.1', 'KI270728.1', 'KI270731.1', 'KI270734.1' and 'chrM'. In 'infercnv::run' we used the recommended cut-off = 0.1, cluster_by_groups = T, denoise=T, HMM = T, BayesMaxP-Normal = 0.5 and analysis_mode = 'subc'. We used a reference set covering range of epithelial, stromal and immune cells sampled from the mammoplasty donors alongside the non-epithelial cells of the DDC donors. These reference cells were used to estimate baseline expression profiles across the genome, in turn highlighting any abnormalities in cells comprising of donors 17 and 37 epithelium.

## Ultivue InSituPlex staining and imaging

Formalin-fixed, paraffin-embedded (FFPE) human tissue blocks were prepared as 4 µm sections, baked for 2 h at 60 °C and stained using Ultivue's Immuno8 FixVUE Panel (Ultivue Inc.; CD8, PD1, PDL1, CD68,

CD3, CD8, FoxP3 and pan-cytokeratin (panCK) cocktail). The kits contain pre-optimized reagents and buffers necessary to stain ten slides. Slides were stained following the manufacturer's instructions. In brief, FFPE tissue sections were loaded on to the Leica Bond RX platform, along with the Ultivue reagents that had been prepared according to the manufacturer's instructions. All eight of the DNA barcoded antibodies were applied as a cocktail, amplified and then detected in two rounds of four using fluorescent probes conjugated with complementary DNA barcodes.

Slides were scanned on the PhenoImager HT (Akoya Biosciences) at 20× magnification (resolution 0.5 μm per pixel) in two rounds; these images were then stacked in the Ultistacker software (Ultivue). HALO (v3.6.4134.137) and HighPlex FL (v4.2.14) modules were used for automated analysis of the images. Optical densities (as defined by the area quantification module on HALO) for positively stained cells used for the automated quantitative analysis of scanned sections were: PDL1 (cytoplasmic)−24.7295, CD68 (nuclear/cytoplasmic)−19.3527, CD8 (nuclear)−15.9375, PD1 (cytoplasm)−43.0822, FoxP3 (nuclear)−17.0759, panCK/Sox10 (cytoplasm)−15.9375, CD3 (nuclear)−20.4455, and CD4 (nuclear)−14.4804. Ten cellular phenotypes were identified: PD1 CD4 positive cells (+CD4/PD1); PD1 CD8 positive cells (+CD8/PD1); PD1 CD3 positive cells (+CD3/PD1); PD1 CD68 positive cells (+CD68/PD1); FoxP3 CD4 positive cells (+CD4/FoxP3); FoxP3 CD8 positive cells (+CD8/FoxP3); FoxP3 CD4 positive cells (+CD4/FoxP3); PD1 PanCK/Sox10 positive cells (+PD1/PanCK/Sox10); PDL1 CD68 positive cells (+CD68/PDL1); PDL1 CD4 positive cells (+CD4/PDL1); and PDL1 CD8 positive cells (+CD8/PDL1).

A random forest classifier was used to distinguish between epithelium, non-epithelium, immune hot and excluded regions within the images. Annotations were created manually on several images, and these were then used to train the classifier. The outputs of the classifier were used to define the tissue regions to give results for the ten phenotypes in these areas specifically. Images were analyzed using QuPath v0.4.3.

## iHBCA
Four of the six external scRNA-seq datasets[14–18,20] downloaded for the iHBCA, were acquired from Gene Expression Omnibus as processed counts matrices: GSE174588 (ref. [14]), GSE180878 (ref. [15]), GSE198732 (ref. [17]) and GSE161529 (ref. [18]). One dataset[20] was acquired as an AnnData object from CellxGene (here) and the counts of the final dataset[16] can be generated using the corresponding scRNA-seq analysis scripts on their GitHub[51]—raw data available on ArrayExpress: 'E-MTAB-9841' (batch 1), 'E-MTAB-10855' (batch 2) and 'E-MTAB-10885' (batch 3). Each of these datasets were then manually formatted to allow dataset merging with the processed data of the HBCA. During merging, we performed an intersection of the common genes (Ensemble ID) across all datasets and used this for dataset integration and dimension reduction to avoid adding artificial batch effects. Integration was performed using scVI over the sequencing batch variable and dimension reduction was performed with scanpy pipelines using identical (hyper)parameters as described above for the HBCA.

Quantitative cell-type comparisons across datasets were performed utilizing CellTypist (v0.1.9) logistic regression classifiers[30]. These were trained on the labels provided by the origin dataset authors and used to make predictions on the remaining datasets of the iHBCA. The predicted labels from each of the datasets (labeled by first author) cell-type labeling system are stored under the column 'map_celltype_map2[author]' with the label convention '[author]-[cell_type]' for predicted labels and 'True-[cell_type]' for the reference cells (in other words, those cells from the dataset whose labels are being projected).

Cell-type differential abundance testing on the iHBCA was completed using edgeR (v3.36.0)[48]. A negative binomial generalized log–linear model was fitted to the cell-type counts from the CellTypist mapping onto HBCA cell-type annotations. Age and parity tests were performed only on mammoplasty donors with the model formulas: 'dataset + live_sorted_boolean + parous + old_young' and 'dataset + live_sorted_boolean + old_young + parous', respectively. Here 'dataset' is a categorical variable denoting both the dataset of origin and sample_type, 'live_sorted_boolean' is a Boolean variable denoting whether or not the samples were FACS sorted (sample with any other cell sorting were removed), 'parous' is a Boolean for parity >0 and 'old_young' is a Boolean for donor_age >49 (as this was the only age information available for the Kumar et al. dataset). The HR-BR1 and HR-BR2 tests were performed on all mammoplasty and BRCA1 or BRCA2 mutant mastectomy donors with the following model formula: 'dataset + live_sorted_boolean + parous + old_young + risk_status', where 'risk_status' is a Boolean denoting BRCA1 or BRCA2 mutation carriers. For each the 'glmQLFTest' function was used to identify genes that have a log fold change significantly different from 0 at an FDR of 0.05.

## Immunofluorescence staining and analysis
Human mammary gland FFPE were immunostained with antibodies for PanCK (1:1,000, Novus Bio NBP2-29429) in combination with HAVCR2 (1:150, Abcam ab241332), TIGIT (1:200, Abcam ab243903) or GZMH (1:200, Atlas Antibodies HPA029200). Antigen retrieval was performed according to the antibody specification sheet or using sodium citrate at pH 6 if not stated. Secondary staining involved goat anti-rat AlexaFluor 488, or anti-mouse AlexaFluor 568 (1:200, Invitrogen). Nuclear content was stained using PureBlu Hoescht 33342 (1:100 of 2uM stock, Bio-Rad) and slides mounted using ProLong Gold Antifade Mountant (Thermo Fisher, P36934). Slides were then scanned on Akoya Biosciences PhenoImager.

Immunofluorescence images were analyzed using ImageJ v1.54f. HAVCR2, TIGIT and GZMH expression was quantified relative to DAPI. Split channels were converted into binary ('Convert to Mask') before excluding the epithelial/lumen regions (with 'Fill Holes' and the image calculator function 'Subtract Create'). The resulting images were then processed, using 'Gaussian Blur' ($\sigma = 1$ for DAPI and 1.75 otherwise) and 'Subtract Background' (rolling of 50 for DAPI and 30 otherwise). The DAPI channel was then converted to binary, while a threshold was set for the HAVCR2/TIGIT/GZMH channel (lower threshold of 30 and upper threshold of 255) before binary conversion to remove non-specific staining. The DAPI channel was then segmented to distinguish individual nuclei ('Watershed'). To identify true immune regions, only the total area of HAVCR2/TIGIT/GZMH expression colocalizing with DAPI (imageCalculator ('AND create')) was analyzed ('Analyze Particles'). The same was repeated for DAPI only, allowing the proportion of colocalised HAVCR2/TIGIT/GZMH area over DAPI to be calculated.

## Statistics and reproducibility
No statistical method was used to predetermine sample size as sample size was limited by availability of material. One 10× sequencing lane belonging to donor 50 was removed in early quality control stages due to the sample reading poor quality control metrics and showing large quantities of debris. This was determined a failed lane of sequencing and thus ignored from further analysis. No further data exclusions were made. We used randomization to group samples for sequencing to minimize possible batch effects (see 'Library preparation and sequencing' for more details). In statistical testing we blocked for effects of age and parity where possible to minimize confounding (see 'Milo and differential abundance analysis' above for specific details). The investigators were not blinded to allocation during experiments and outcome assessment as for this study no treatments were provided to the participants. For all immunofluorescence images, the number of individual donors analyzed per staining is indicated in the figure legend. Staining was performed on one whole section per donor, and a representative snapshot for each section is reported in the figures.

**Reporting summary**

Further information on research design is available in the Nature Portfolio Reporting Summary linked to this article.

## Data availability

The authors declare that all data supporting the findings of this study and unprocessed images are available within the article and its supplementary information. The raw sequencing data and CellRanger raw outputs are available on ArrayExpress with accession number E-MTAB-13664. Processed data from our study, as well as the iHBCA, can be found and explored using the user-friendly CELLxGENE at https://cellxgene.cziscience.com/collections/48259aa8-f168-4bf5-b797-af8e88da6637. The trained CellTypist logistic regression models for label transfer can be downloaded from https://doi.org/10.5281/zenodo.10044650 (ref. 52).

## Code availability

All code is available on GitHub at https://github.com/MarioniLab/hbca (ref. 53).

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

## Acknowledgements

We thank the staff at the Cambridge NIHR BRC Cell Phenotyping Hub and the Genomic and Histopathology Core at the CRUK Cambridge Institute for their constant support and assistance and S. Westermann for designing the schematic in Fig. 1. We acknowledge the role of the Breast Cancer Now Tissue Bank in collecting and making available the samples used in the generation of this publication and all the patients who donated. This study was primarily funded by an MRC project grant (MR/S036059/1) and supported by BBSRC project grant (BB/S006745/1), Breast Cancer Now project grant (2017MayPR907), CRUK career establishment award (17348) and CRUK programme foundation award (DCRPGF\100010) to W.T.K. and core funding from EMBL and CRUK (C9545/A29580) to J.C.M.

## Author contributions

A.D.R. performed the majority of the bioinformatic analysis and interpretation of the data. S.P. contributed to the study design, sample processing, analysis and interpretation of the data. J.S. contributed to the sample processing. D.J.K. and P.H. contributed to the data processing, batch correction and cell cluster identification. A.S. contributed to the design of the sample batches and contributed to the analysis of the raw data. A.J.T. contributed to the analysis of the data and Figure design. L.J.P. performed the immune histochemistry validations. K.H. assisted A.D.R. with the inferCNV analysis and interpretation. P.H. assisted with the subclustering of immune cells and scVI integration analysis. A.Q.S. performed the immunofluorescence quantification. K.K. performed all the scRNA-seq library preparation and sequencing. R.B.M., I.G., J.J.G., V.S. and J.L.J. provided the human tissues and the metadata from the 55 donors. A.D.R., S.P., J.C.M. and W.T.K. wrote the paper. J.C.M. and W.T.K. conceptualized and supervised the study.

## Competing interests

J.C.M. has been an employee of Genentech since September 2022. The other authors declare no competing interests.

## Additional information

**Extended data** is available for this paper at https://doi.org/10.1038/s41588-024-01688-9.

**Correspondence and requests for materials** should be addressed to John C. Marioni or Walid T. Khaled.

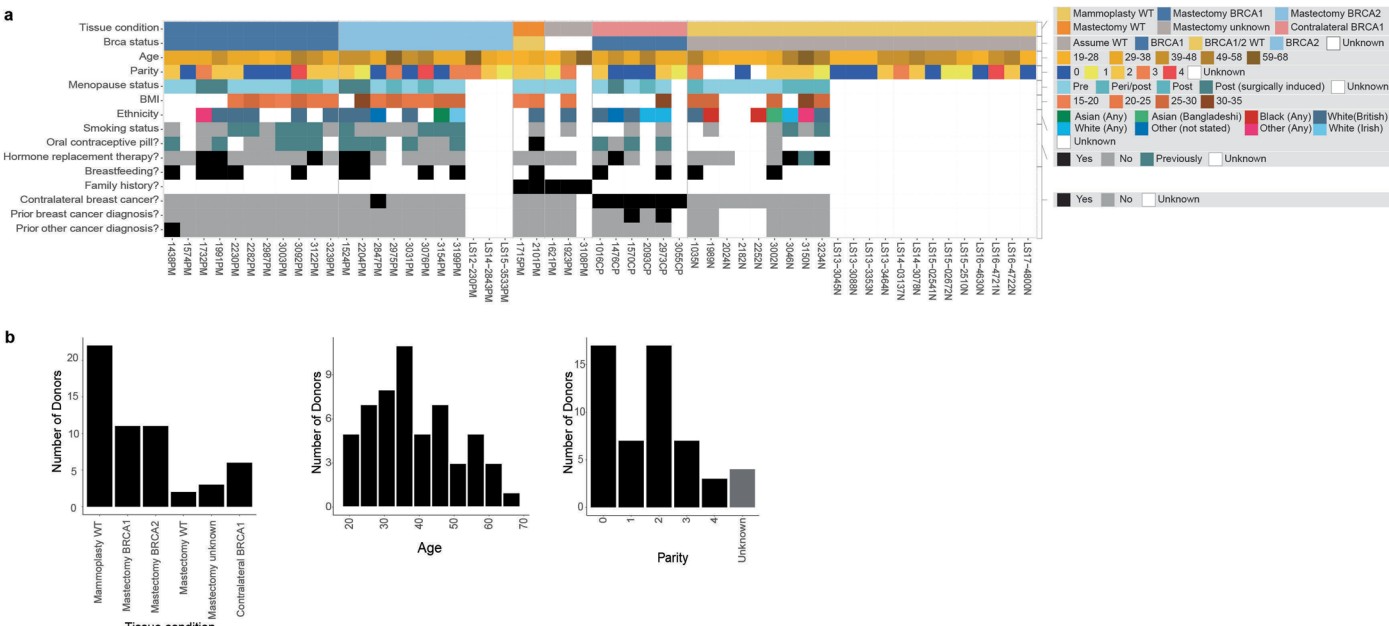

**Extended Data Fig. 1 | Summary of major demographic metadata.**
(**a**) Summary table visualising some of the main demographics metadata for each of our donors. Full metadata listed per sample (10X lane) in

Supplementary Table 1. (**b**) Bar plot/histograms showing the total distribution of our main three demographics explored in the paper – tissue condition (surgery and BRCA status), age and parity.

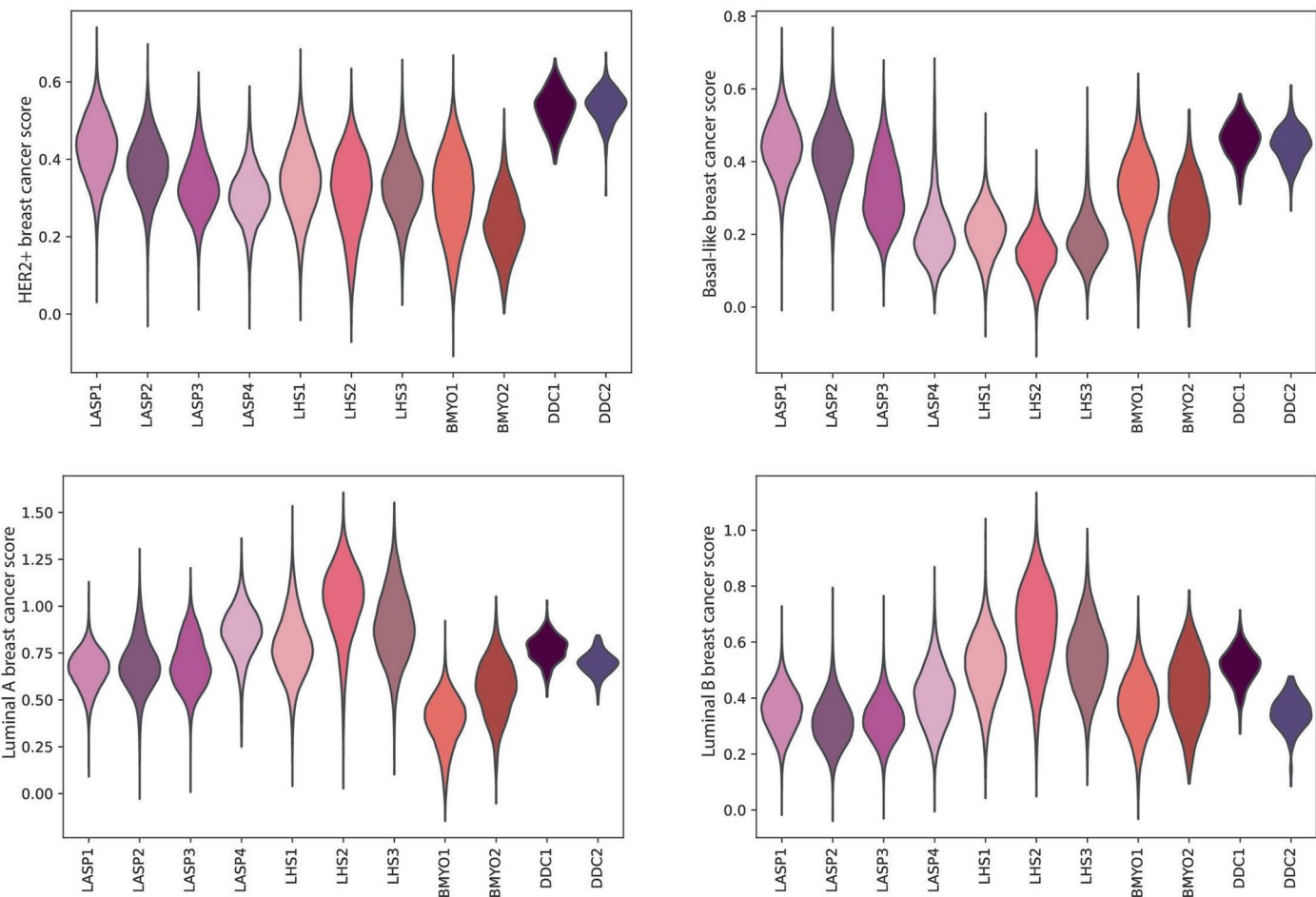

**Extended Data Fig. 2 | Specificity of subcluster gene signatures.** Gene scores for the four molecular subtypes of breast cancer on the epithelial subclusters based gene sets described in Wu et. al.[28].

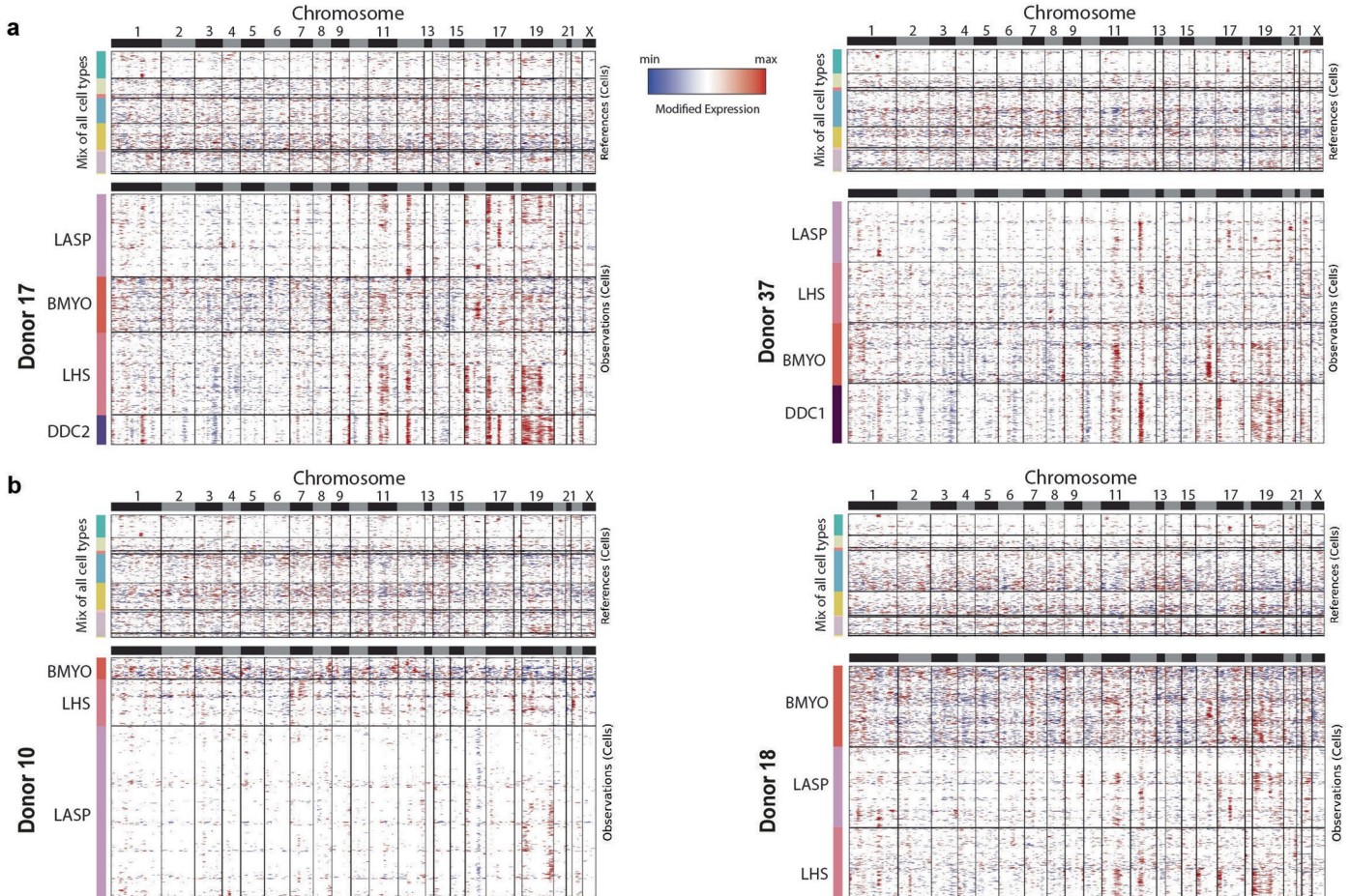

**Extended Data Fig. 3 | InferCNV predicts putative CNV profiles in the DDC1/2 clusters.** (**a**) Denoised inferCNV output plot predicting copy number variation (CNV) profiles per cell over the epithelium of Donors 17 (left) and 37 (right), both containing the unique donor derived clusters (DDCs). Red (blue) indicates increased (decreased) expression in the respective genomic region. For each

inferCNV test, a reference dataset of cells sampled from mammoplasty donors (excluding those tested) plus the remaining stroma/immune cells of the tested donor, were used to account for any cell type or donor specific expression profiles across the genome. (**b**) Similar analysis as above but for two randomly selected control mammoplasty donors.

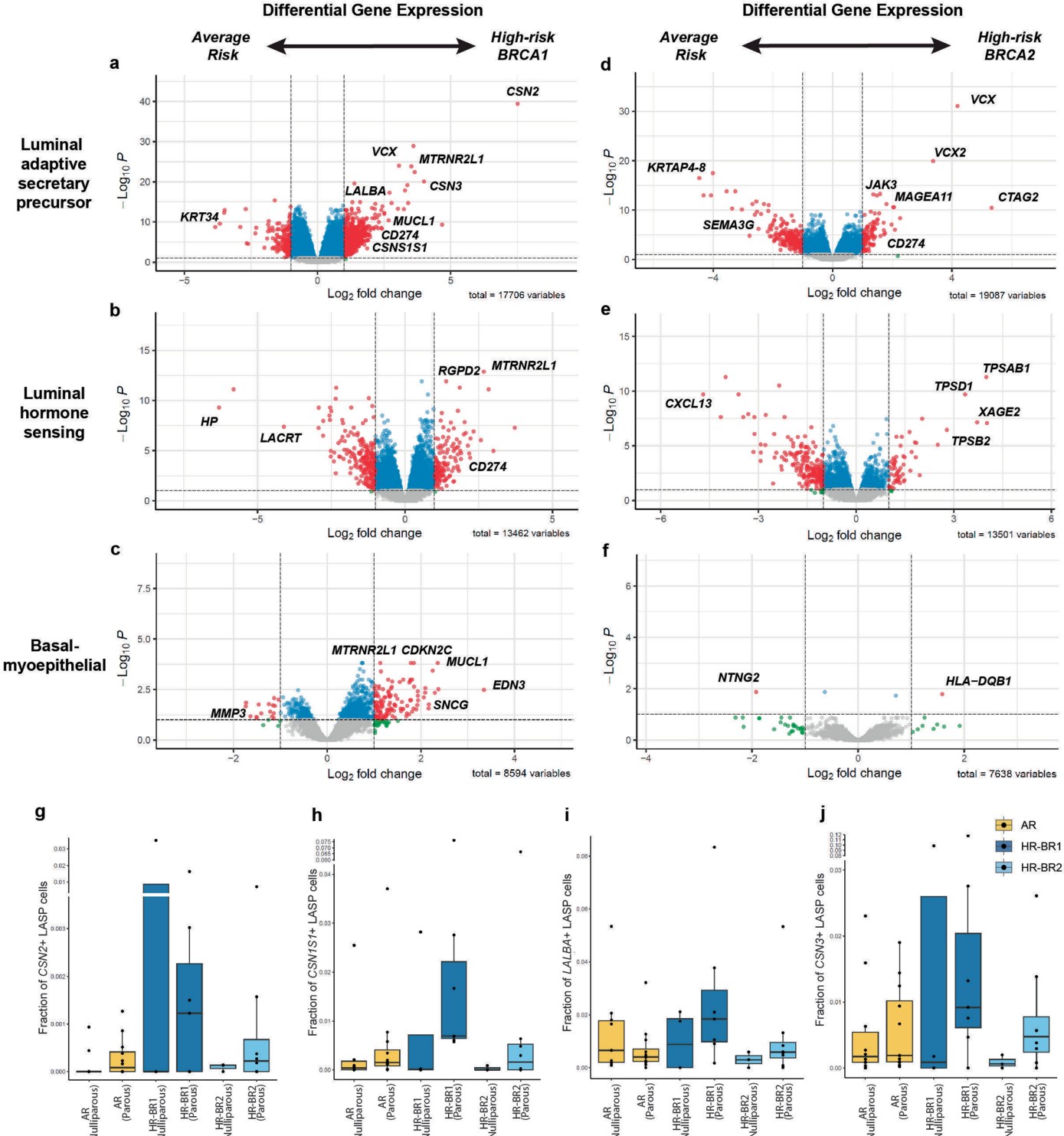

**Extended Data Fig. 4 | Differential gene expression of high-risk donor epithelium.** (a–c) Volcano plots showing the results of differential gene expression testing between average risk (AR) and high risk *BRCA1* germline (HR-BR1) donors within the luminal adaptive secretory precursor (LASP), luminal hormone sensing (LHS) and basal-myoepithelial (BMYO) compartments respectively. Green points have significant log fold changes, blue points have significant FDR (FDR < 0.1), red points have significant log fold change and FDR. (d–f) Volcano plots showing the results of differential gene expression testing between AR and high risk *BRCA2* germline (HR-BR2) donors within the LASP, LHS and BMYO compartments respectively. Green points have significant

log fold changes, blue points have significant FDR (FDR < 0.1), red points have significant log fold change and FDR. (g-j) Box plots showing the upregulation of milk biosynthesis related genes in HR donors compared to AR alongside the related impact of parity on the expression of these genes. These plots show the frequency of non-zero expression of each respective gene amongst the LASP population. Sample numbers: AR nulliparous (7), AR parous (15), HR-BR1 nulliparous (4), HR-BR1 parous (7), HR-BR2 nulliparous (3), HR-BR2 parous (8). The boxplot centers show median values while the minima / maxima show the 25th /75th percentiles respectively and whiskers extend to the most extreme datapoint within 1.5 × IQR (inter-quartile range) of the outer hinge of the boxplot.

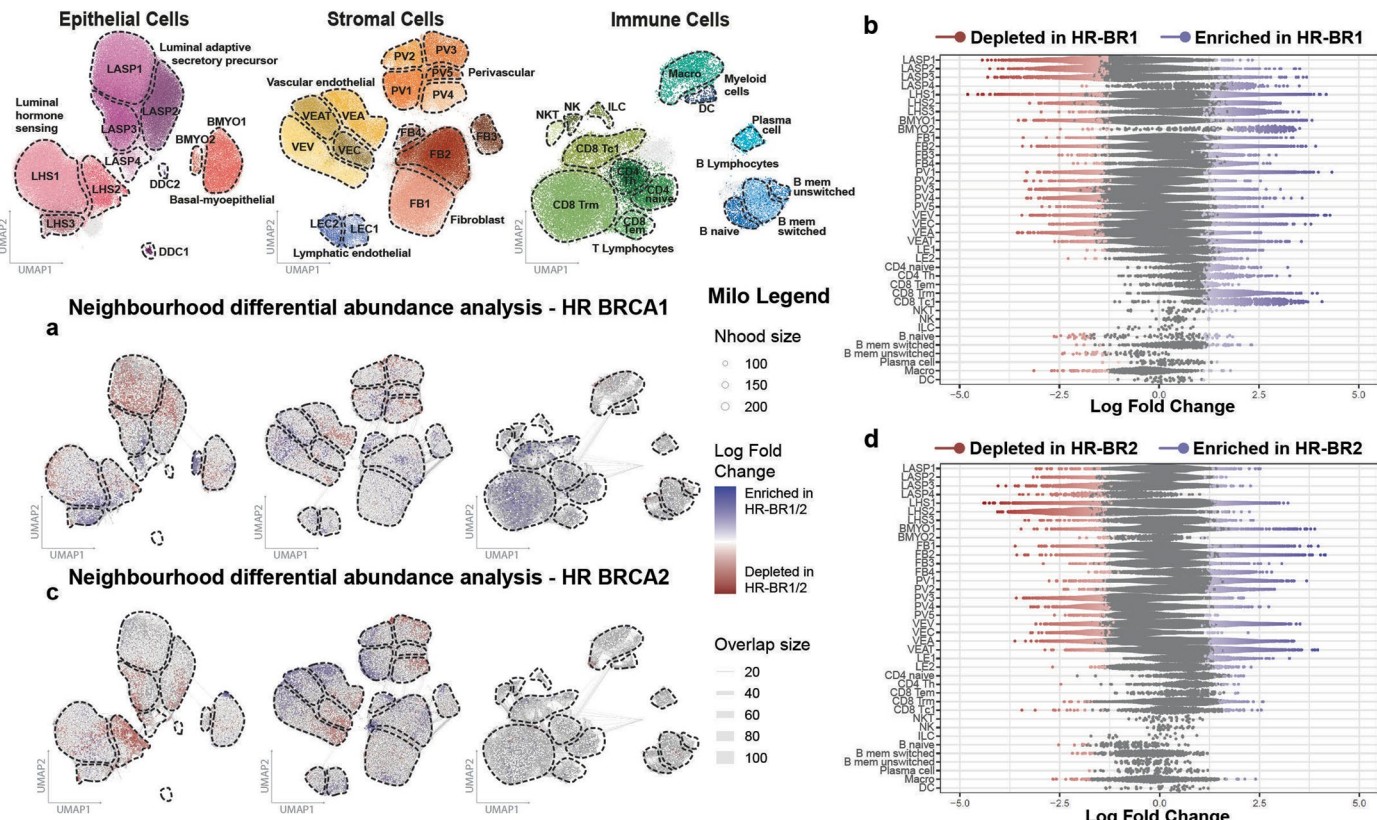

**Extended Data Fig. 5 | Impact of *BRCA1* and *BRCA2* germline mutations on the cellular composition of the breast. (a)** Milo cell neighbourhood differential abundance plots of the significant (FDR < 0.05) changes in the breast composition comparing average risk (AR) donors (n = 22) to the high risk *BRCA1* germline (HR-BR1) cohort (n = 11), blocking for the effects of age and parity. Blue represents enrichment with HR-BR1, whilst red denotes depletion with HR-BR1. **(b)** Beeswarm plot of the log fold changes of the Milo neighbourhoods grouped into each cell type subcluster for AR versus HR-BR1. Neighbourhoods with a significant change in cellular abundance are coloured

as indicated. **(c)** Milo cell neighbourhood differential abundance plots of the significant (FDR < 0.05) changes in the breast composition comparing AR donors (n = 22) to the high risk *BRCA2* germline (HR-BR2) cohort (n = 11), blocking for the effects of age and parity. Blue represents enrichment with HR-BR2, whilst red denotes depletion with HR-BR2. **(d)** Beeswarm plot of the log fold changes of the Milo neighbourhoods grouped into each cell type subcluster for AR versus HR-BR2. Neighbourhoods with a significant change in cellular abundance are coloured as indicated.

**AR**

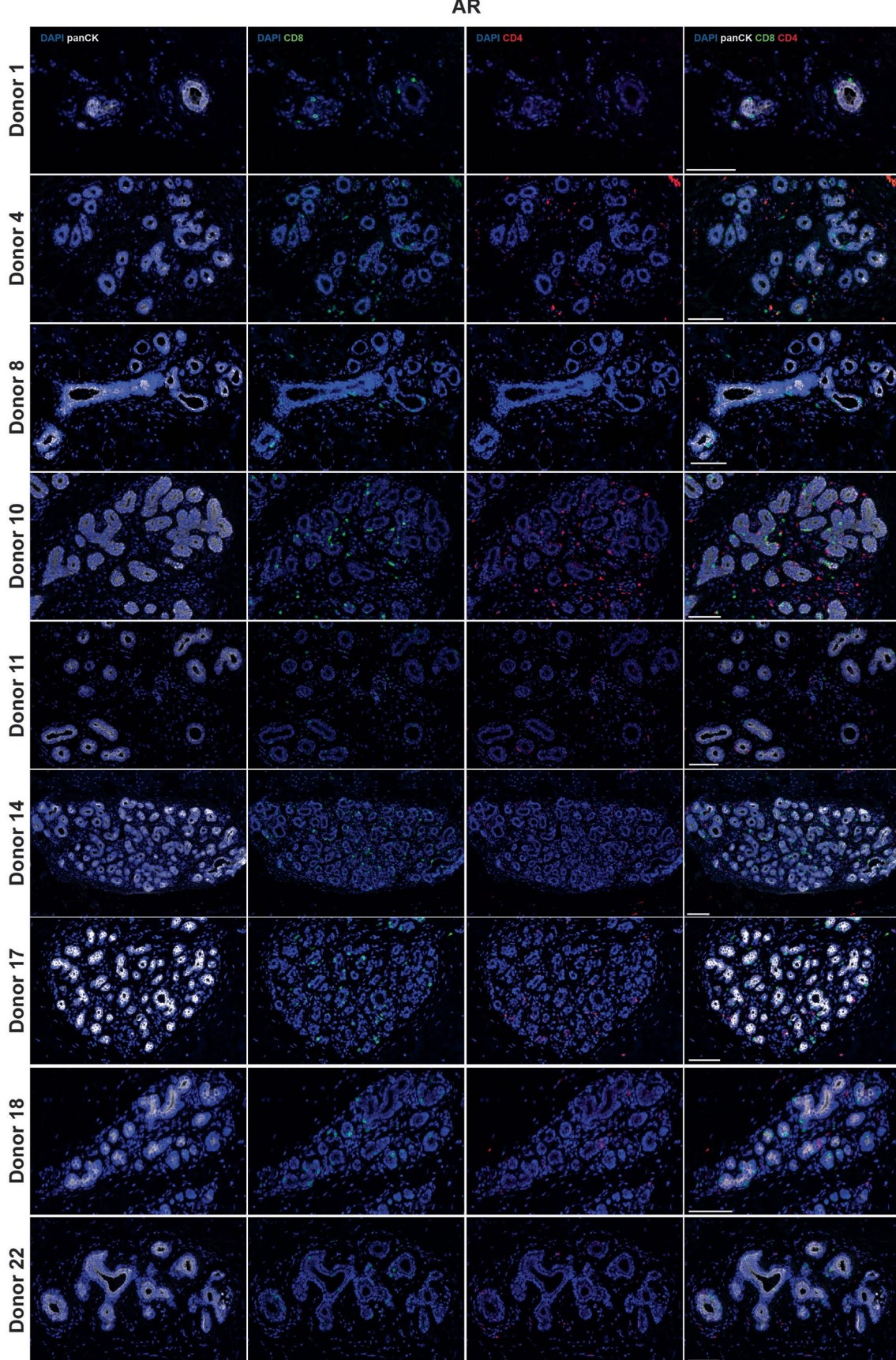

**Extended Data Fig. 6 | CD8 and CD4 expression in AR donors.** Ultivue staining showing panCK (white), CD8 (green), CD4 (red) and DAPI (blue) of representative breast sections from AR donors (n = 9). Scale bars represent 100 μm.

HR-BR1

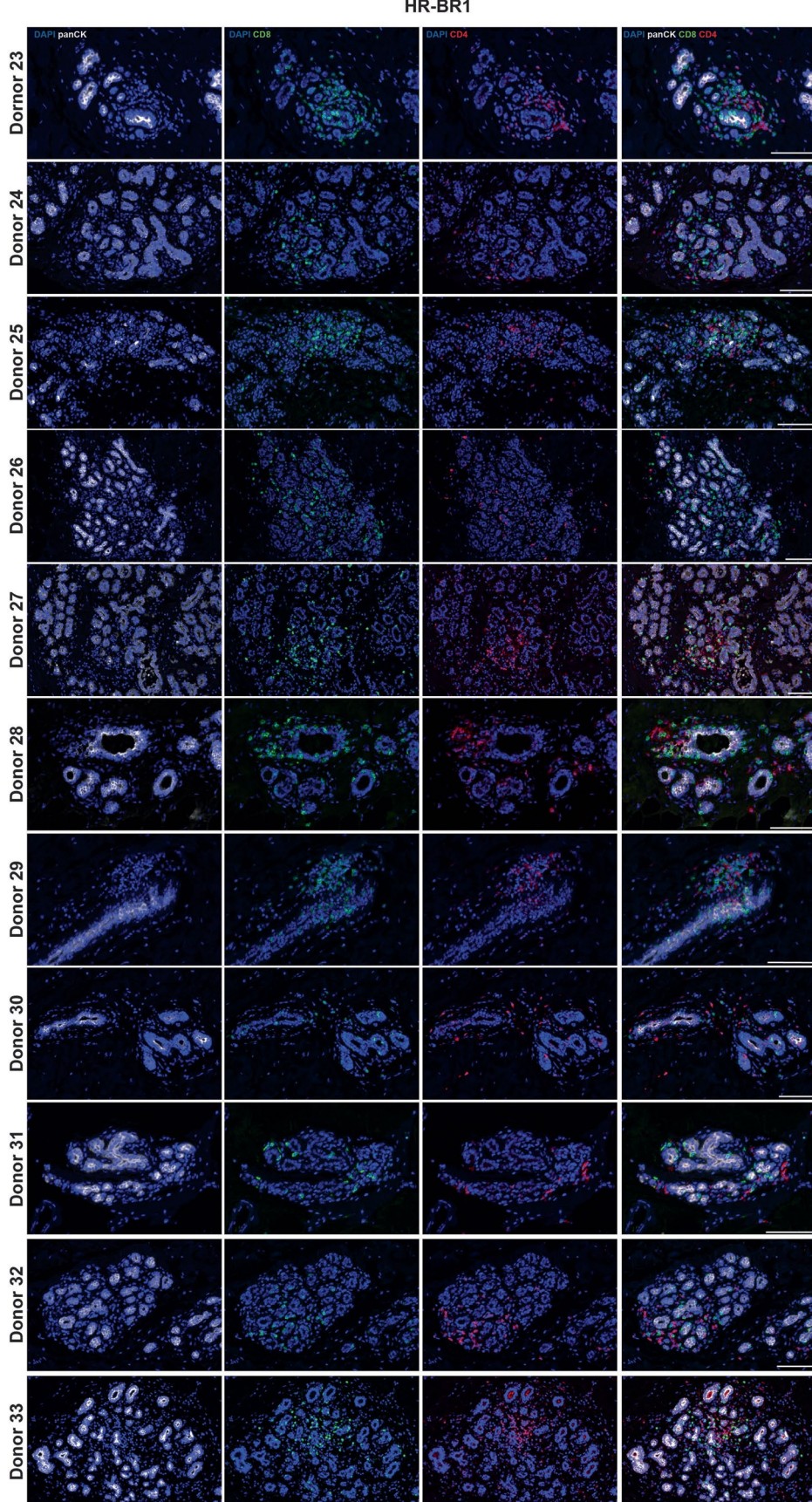

**Extended Data Fig. 7 | CD8 and CD4 expression in HR-BR1 donors.** Ultivue staining showing panCK (white), CD8 (green), CD4 (red) and DAPI (blue) of representative breast sections from HR-BR1 donors (n = 11). Scale bars represent 100 μm.

# HR-BR2

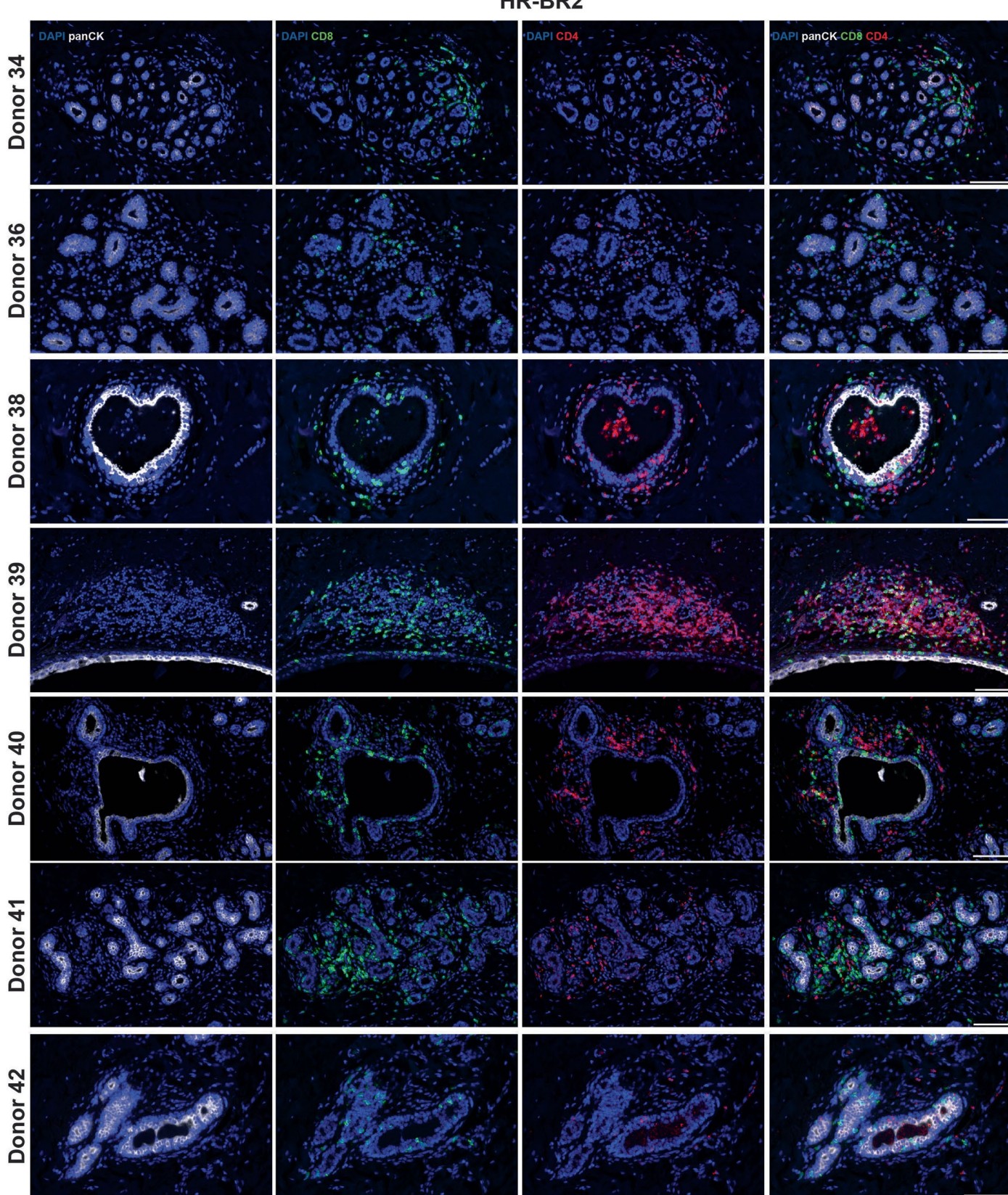

**Extended Data Fig. 8 | CD8 and CD4 expression in HR-BR2 donors.** Ultivue staining showing panCK (white), CD8 (green), CD4 (red) and DAPI (blue) of representative breast sections from HR-BR2 donors (n = 7). Scale bars represent 100 µm.

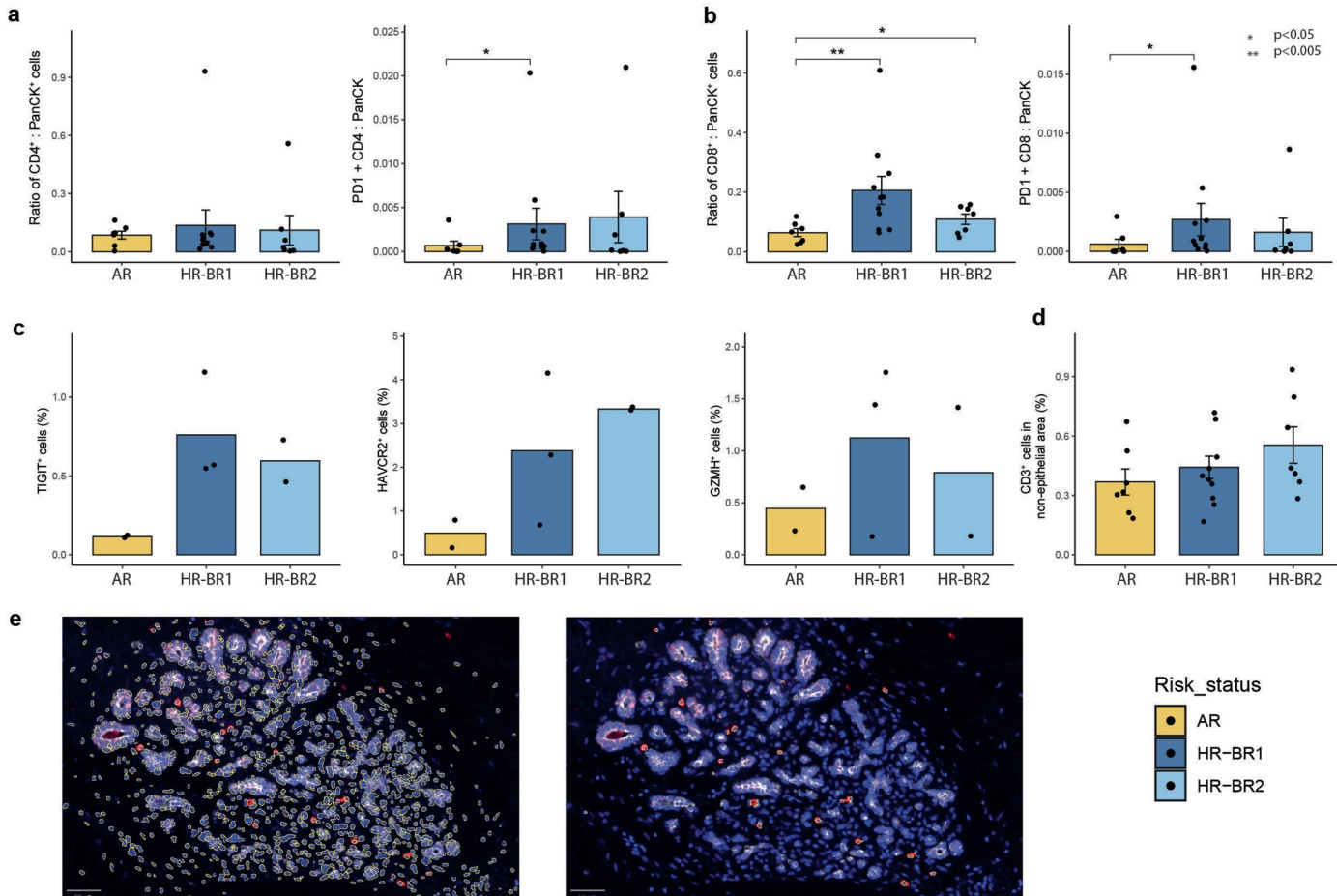

**Extended Data Fig. 9 | Quantification of Ultivue and Immunofluorescence images.** (**a**) Bar plots showing the ratio of CD4+ (left) and CD4+/PD1+ double-positive (right) cells to PanCK+ cells in whole Ultivue tissue slides from average risk (AR; n = 10), high-risk *BRCA1* (HR-BR1; n = 11, p=) and high-risk *BRCA2* (HR-BR2; n = 8) donors. The p-values are calculated with one-way non-parametric Wilcox test with * (**) indicating p < 0.05 (p < 0.005) respectively. The AR vs HR-BR1 (CD4+PD1 : PanCK) comparison has p = 0.028. Error bars show the standard error of the mean. (**b**) Bar plots showing the ratio of CD8+ (left) and CD8+/PD1+ double-positive (right) cells to PanCK+ cells in whole Ultivue tissue slides from average risk (AR; n = 10), high-risk *BRCA1* (HR-BR1; n = 11) and high-risk *BRCA2* (HR-BR2; n = 8) donors. The p-values are calculated with one-way non-parametric Wilcox test with * (**) indicating p < 0.05 (p < 0.005) respectively. The AR vs HR-BR1 (CD8 : PanCK) comparison has p = 0.0041, AR vs HR-BR2 (CD8 : PanCK)

comparison has p = 0.049 and AR vs HR-BR1 (CD8+PD1 : PanCK) comparison has p = 0.028. Error bars show the standard error of the mean. (**c**) Bar plots showing the percentage of TIGIT+ (left), HAVCR2+ (middle) and GZMH+ (right) cells from the immunofluorescence images (n = 2 for AR and HR-BR2, n = 3 for HR-BR1) shown in Fig. 5 and Supplementary Figs. 19–22. (**d**) Bar plot showing the percentage of CD3+ cells located in non-epithelial epithelial areas (ie. not intercalated with the epithelium) in HR-BR1 (n = 11) and HR-BR2 (n = 8) donors compared to AR (n = 10) donors in whole Ultivue slides (see Methods). No significant differences were found with one-way non-parametric Wilcox test. Error bars show the standard error of the mean. (**e**) Example segmentation and calling of positive cells of an immunofluorescence (GZMH) image (AR n = 2, HR-BR1 n = 3, HR-BR2 n = 2).

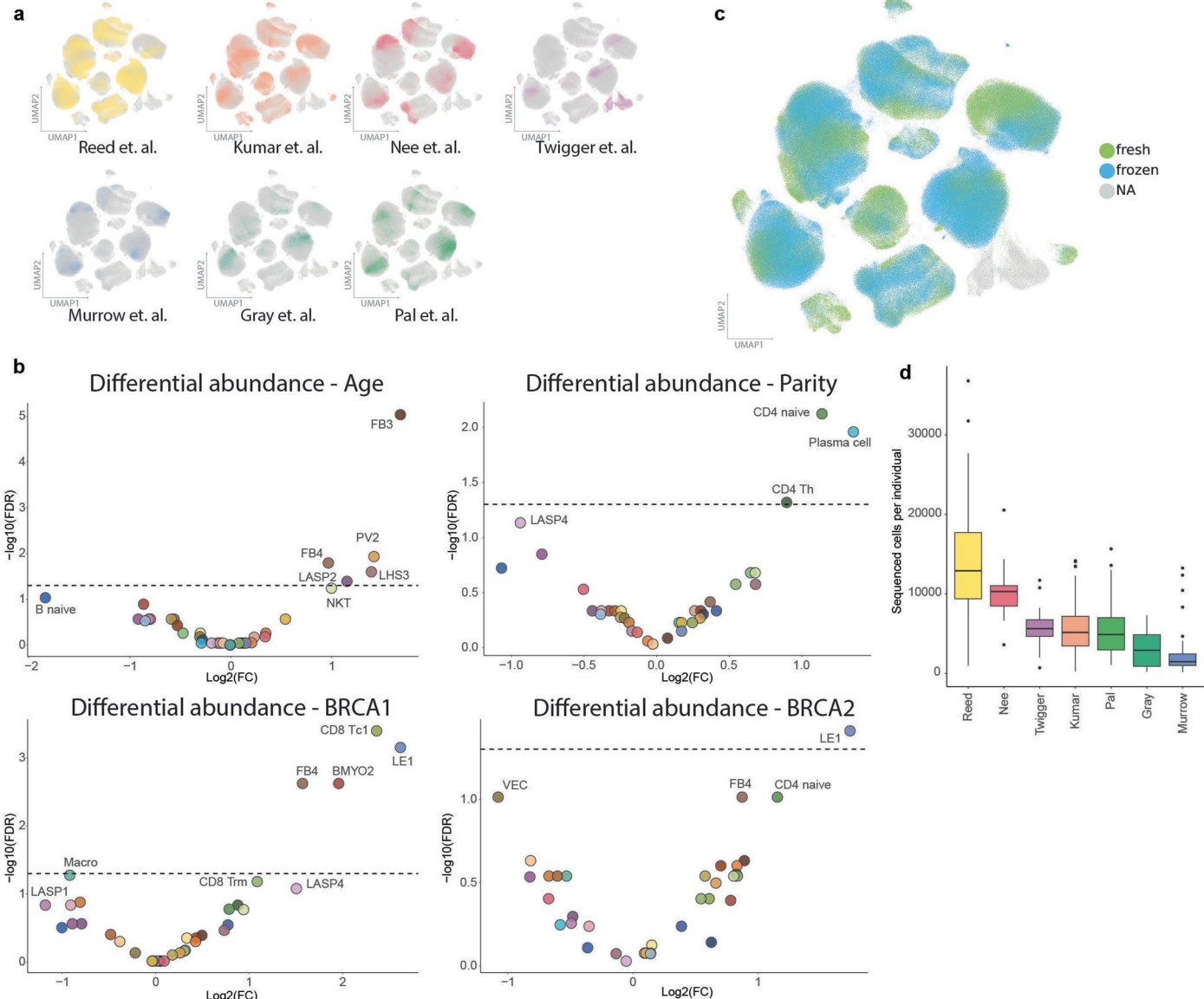

**Extended Data Fig. 10 | The integrated Human Breast Cell Atlas (iHBCA) highlights effects of tissue preparation and confirms Milo differential abundance results.** (**a**) The global iHBCA uniform manifold approximation and projection (UMAP) plots coloured individually by each dataset (rest plotted underneath in grey). (**b**) Standard negative-binomial differential abundance analysis performed on the iHBCA for each of the comparisons previously explored using Milo on the Human Breast Cell Atlas (HBCA). Here we used the same base blocking terms as used in the milo analysis but with additional terms for dataset and live_sorting status (see Methods for details). (**c**) Global iHBCA

UMAP showing the distribution of cells from fresh and frozen tissue. (**d**) Box and whisker plot showing the number of cells sequenced per individual across datasets. Donor numbers: Reed (n = 55), Nee (n = 22), Twigger (n = 18), Kumar (n = 126), Pal (n = 21), Gray (n = 16), Murrow (n = 28). The boxplot centers show median values while the minima / maxima show the 25th /75th percentiles respectively and whiskers extend to the most extreme datapoint within 1.5 × IQR (inter-quartile range) of the outer hinge of the boxplot. Outliers are then displayed independently.

# Reporting Summary

## Statistics

For all statistical analyses, confirm that the following items are present in the figure legend, table legend, main text, or Methods section.

| n/a | Confirmed | |
|---|---|---|
| ☐ | ☒ | The exact sample size (*n*) for each experimental group/condition, given as a discrete number and unit of measurement |
| ☐ | ☒ | A statement on whether measurements were taken from distinct samples or whether the same sample was measured repeatedly |
| ☐ | ☒ | The statistical test(s) used AND whether they are one- or two-sided *Only common tests should be described solely by name; describe more complex techniques in the Methods section.* |
| ☐ | ☒ | A description of all covariates tested |
| ☐ | ☒ | A description of any assumptions or corrections, such as tests of normality and adjustment for multiple comparisons |
| ☐ | ☒ | A full description of the statistical parameters including central tendency (e.g. means) or other basic estimates (e.g. regression coefficient) AND variation (e.g. standard deviation) or associated estimates of uncertainty (e.g. confidence intervals) |
| ☐ | ☒ | For null hypothesis testing, the test statistic (e.g. *F*, *t*, *r*) with confidence intervals, effect sizes, degrees of freedom and *P* value noted *Give P values as exact values whenever suitable.* |
| ☒ | ☐ | For Bayesian analysis, information on the choice of priors and Markov chain Monte Carlo settings |
| ☒ | ☐ | For hierarchical and complex designs, identification of the appropriate level for tests and full reporting of outcomes |
| ☒ | ☐ | Estimates of effect sizes (e.g. Cohen's *d*, Pearson's *r*), indicating how they were calculated |

*Our web collection on statistics for biologists contains articles on many of the points above.*

## Software and code

Policy information about availability of computer code

| Data collection | No software was used for data collection |
|---|---|
| Data analysis | Cell Ranger Single-Cell Software Suite (v6.0.2) was used for demultiplexing, barcode assignment and UMI quantification. All downstream computation analyses were performed in R and Python using standard functions unless otherwise indicated. The specific package and language versions used for each section of analysis are specified in conda yaml files included the HBCA GitHub repository referenced in Code Availability of the text (https://github.com/MarioniLab/hbca). The following packages were used for scRNA-seq data analysis: Cell Ranger Single-Cell Software Suite (v6.0.2), Vireo (v0.5.6), DropletUtils (v1.12.1), Scrublet (v0.2.3), scanpy (v1.8.2), edgeR (v3.36.0), Milo (v1.3.1), Cell Chat (v1.6.0), inferCNV (v. 1.10.0), CellTypist (v0.1.9). FlowJo V10 was used to analyse flow cytometry data. Qupath v0.4.3, ImageJ v1.54f and Halo v3.6.4134.137 and HighPlex FL v4.2.14 were used to analyse Ultivue and Immunofluorescence images |

For manuscripts utilizing custom algorithms or software that are central to the research but not yet described in published literature, software must be made available to editors and reviewers. We strongly encourage code deposition in a community repository (e.g. GitHub). See the Nature Portfolio guidelines for submitting code & software for further information.

## Data

Policy information about availability of data

All manuscripts must include a data availability statement. This statement should provide the following information, where applicable:

- Accession codes, unique identifiers, or web links for publicly available datasets
- A description of any restrictions on data availability
- For clinical datasets or third party data, please ensure that the statement adheres to our policy

The authors declare that all data supporting the findings of this study and unprocessed images are available within the article and its supplementary information files or from the corresponding author upon reasonable request. The raw sequencing data and individual sample processed matrices are available on Array Express E-MTAB-13664. Processed data can also be explored and downloaded at the CellXGene site (https://cellxgene.cziscience.com/collections/cd9a09e2-b440-4887-9163-6f8c684c7ced). The trained CellTypist logistic regression models for label transfer can be found and downloaded from DOI: 10.5281/zenodo.10044650.

## Research involving human participants, their data, or biological material

Policy information about studies with human participants or human data. See also policy information about sex, gender (identity/presentation), and sexual orientation and race, ethnicity and racism.

| | |
|---|---|
| Reporting on sex and gender | The samples included in the study were collected from female individuals only, therefore findings only apply to female individuals and no sex-based analysis was performed. No information was collected on gender, therefore no gender-based analysis could be performed. |
| Reporting on race, ethnicity, or other socially relevant groupings | Metadata linked to the tissue bank samples was provided by BCN. Ethnicity was reported in Extended Data Figure 1 and Supplementary Table 1 when available, however it was not taken into account in the data analysis as it was only reported for a fraction of the samples. |
| Population characteristics | All primary human breast tissue was derived from women undergoing reduction mammoplasties with no known genetic history (n = 22) and risk reduction prophylactic mastectomies from women with germline BRCA1 or BRCA2 mutations or other family histories (n = 27) and contralateral mastectomies from BRCA1 carriers that had breast cancer in one breast and had the second breast removed to reduce the risk of further tumours (n = 6 ). No specific age-range was selected. |
| Recruitment | Participants were not specifically recruited for this study and are part of bigger cohorts where recruitment was not based on the parameters of interest for this analysis. |
| Ethics oversight | All primary human breast tissue was obtained from the Breast Cancer Now Tissue bank, as approved by REC (15/EE/0192). |

Note that full information on the approval of the study protocol must also be provided in the manuscript.

# Field-specific reporting

Please select the one below that is the best fit for your research. If you are not sure, read the appropriate sections before making your selection.

☒ Life sciences ☐ Behavioural & social sciences ☐ Ecological, evolutionary & environmental sciences

For a reference copy of the document with all sections, see nature.com/documents/nr-reporting-summary-flat.pdf

# Life sciences study design

All studies must disclose on these points even when the disclosure is negative.

| | |
|---|---|
| Sample size | Sample size was limited by availability of material. |
| Data exclusions | One 10X sequencing lane belonging to donor 1016CP was removed in early quality control stages due to the sample reading poor quality control metrics and showing large quantities of debris. This was determined a failed lane of sequencing and thus ignored from further analysis. No further data exclusions were made. |
| Replication | This was an atlas study looking at 55 donors. |
| Randomization | We used randomization to group samples for sequencing to minimize possible batch effects (see methods for more details). In statistical testing we blocked for effects of age and parity where possible to minimise confounding (see methods for specific details). |
| Blinding | Blinding was not relevant for this study as no treatments were provided to the participants. |

# Reporting for specific materials, systems and methods

We require information from authors about some types of materials, experimental systems and methods used in many studies. Here, indicate whether each material, system or method listed is relevant to your study. If you are not sure if a list item applies to your research, read the appropriate section before selecting a response.

## Materials & experimental systems

| n/a | Involved in the study |
|---|---|
| ☐ | ☒ Antibodies |
| ☐ | ☒ Eukaryotic cell lines |
| ☒ | ☐ Palaeontology and archaeology |
| ☒ | ☐ Animals and other organisms |
| ☒ | ☐ Clinical data |
| ☒ | ☐ Dual use research of concern |
| ☒ | ☐ Plants |

## Methods

| n/a | Involved in the study |
|---|---|
| ☒ | ☐ ChIP-seq |
| ☐ | ☒ Flow cytometry |
| ☒ | ☐ MRI-based neuroimaging |

## Antibodies

| Antibodies used | CD45-APC (Biolegend, clone H130,1:100), CD31-APC (Biolegend, clone WM-59, 1:100), EPCAM-AF488 (Biolegend, clone 9C4, 1:50), CD49f-PE/Cy7 (Biolegend, clone GoH3, 1 µg ml−1, 1:200),PanCK (1:1000, Novus Bio NBP2-29429), HAVCR2 (1:150, Abcam ab241332), TIGIT (1:200, Abcam ab243903), GZMH (1:200, Atlas Antibodies HPA029200), goat anti-rat AlexaFluor 488, anti-mouse AlexaFluor 568 (1:200, Invitrogen). |
|---|---|
| Validation | Antibodies were used according to manufacturer specification and validated by manufacturers in house. Immunofluorescence antibodies were validated based on staining pattern and cellular morphology only. Flow cytometry antibodies have been extensively validated in the Khaled lab and all the clones used are extensively implemented in the mammary gland biology field. |

## Eukaryotic cell lines

Policy information about cell lines and Sex and Gender in Research

| Cell line source(s) | HuMECs, Thermo Fisher Scientific A10565 |
|---|---|
| Authentication | No authentication was undertaken. |
| Mycoplasma contamination | The cell line was not tested for mycoplasma. cells were used as spike-in controls for 10X preps |
| Commonly misidentified lines (See ICLAC register) | no commonly misidentified cell lines were used in the study |

## Flow Cytometry

### Plots

Confirm that:

☒ The axis labels state the marker and fluorochrome used (e.g. CD4-FITC).

☒ The axis scales are clearly visible. Include numbers along axes only for bottom left plot of group (a 'group' is an analysis of identical markers).

☒ All plots are contour plots with outliers or pseudocolor plots.

☒ A numerical value for number of cells or percentage (with statistics) is provided.

### Methodology

| Sample preparation | Frozen vials of epithelial-enriched or stromal-enriched fractions were defrosted by gently diluting the material in 50 mL of cold Tissue Preparation Medium (TPM, RPMI 1640 + 25 mM HEPES and 2 mM L-glutamine (Sigma R5886), 5% foetal bovine serum (FBS, Gibco), 100 units/mL penicillin and 0,1 mg/mL streptomycin sulphate (Gibco), washed in PBS without Calcium and Magnesium (D8537, Sigma). Samples were centrifuged at 400 g for 5 minutes and resuspended in 2 mL of freshly prepared PBS + 0,025% Trypsin, 0,1 g/L EDTA (HyClone SV30031.01, Fisher Scientific) and 0,4 mg/mL Deoxyribonuclease 1 (DNase) (10104159001, Boehringer/Roche Diagnostics) previously warmed to 37°C. Samples were then incubated at 37°C with pipetting up and down for 30 seconds every 2-3 minutes until smoothly digested or up to a maximum of 10 minutes. Next, samples were washed in 40 mL of TPM and centrifuged for 20 minutes at 400 g with slow break. The pellet was resuspended by pipetting up and down in 200 µL of TPM +10 µL of 10 mg/mL DNase until homogeneous, then diluted in 25 mL of TPM and filtered through a 40 µm cell strainer (352354, Corning) into a 50 mL tube. After centrifugation for 5 minutes at 400 g, the pellet was resuspended by pipetting up and down in 200 µL of CPM (Cell Preparation Medium, RPMI 1640 + 1% FBS, 100 units/mL penicillin, 0,1 mg/mL streptomycin sulphate) + 10 µL of 10 mg/mL DNase until homogeneous, then washed in 3-6 mL of CPM. 30,000 cells were resuspended into 48 µL of HF (Hank's balanced salt solution (Gibco)+1% FBS) into low binding tubes. 400 Human mammary epithelial cells (HuMECs, Thermo Fisher Scientific A10565) were added as spike-in, and samples were submitted for scRNAseq (unsorted fraction). For the epithelial-enriched fraction only, the rest of the processed |
|---|---|

sample was stained with the following primary antibodies: CD45-APC (Biolegend, clone H130,1:100, staining most hemopoietic cells), CD31-APC (Biolegend, clone WM-59, 1:100, staining endothelial cells), EPCAM-AF488 (Biolegend, clone 9C4, 1:50), CD49f-PE/Cy7 (Biolegend, clone GoH3, 1 μg ml−1, 1:200). DAPI was used to detect dead cells. Cells were filtered through a cell strainer (Partec) before sorting. Sorting of cells was done using a FACS Aria Fusion sorter. Single-stained control cells were used to perform compensation manually and unstained cells were used to set gates. After doublets, dead cells and contaminating haematopoietic and endothelial cells (referred to as lineage) were gated out (Supplementary Fig 2), up to 30,000 luminal progenitors were sorted for scRNAseq (with the addition of 400 HuMECs as spike-in).

Instrument FACS Aria Fusion

Software FlowJo

Cell population abundance Purity of samples was not determined post-sorting.

Gating strategy After doublets, dead cells and contaminating haematopoietic and endothelial cells (referred to as lineage) were gated out, EPCAM+, CD49F+ luminal progenitors cells were sorted. The gating strategy is reported in Supplementary Fig 2. Starting cell population values for FSC include all cells between 25k and 250k, and the whole range of SSC was included.

☒ Tick this box to confirm that a figure exemplifying the gating strategy is provided in the Supplementary Information.

