## [Peer Review File · Nature Genetics]

Peer Review Information

Manuscript Title: A single-cell atlas enables mapping of homeostatic cellular shifts in the adult human breast

Corresponding author name(s): Professor Walid (T.) Khaled, Dr John (C) Marioni

Reviewer Comments & Decisions:

Decision Letter, initial version:

6th Jun 2023

Dear Dr Khaled,

I am sorry that it has taken so long to return this decision to you. Thank you for your patience.

Your Article, "A Human Breast Cell Atlas Mapping the Homeostatic Cellular Shifts in the Adult Breast" has now been seen by 3 referees. You will see from their comments below that while they find your work of interest, some important points are raised. We are interested in the possibility of publishing your study in Nature Genetics, but would like to consider your response to these concerns in the form of a revised manuscript before we make a final decision on publication.

You'll see that a concern shared by all reviewers was the lack of validation of key results. We agree that this aspect of the work should be fortified and we'd suggest following the guidance provided by the referees; Reviewer #3 has suggested comparisons with existing datasets and orthogonal validation through experimental approaches (such as IHC, immunofluorescence or in situ hybridisation as suggested by Reviewer #1). All other comments should be addressed in full, either through additional analyses or through textual edits where appropriate.

We therefore invite you to revise your manuscript taking into account all reviewer and editor comments. Please highlight all changes in the manuscript text file. At this stage we will need you to upload a copy of the manuscript in MS Word .docx or similar editable format.

*2) If you have not done so already please begin to revise your manuscript so that it conforms to our Article format instructions, available [here](http://www.nature.com/ng/authors/article_types/index.html). Refer also to any guidelines provided in this letter.

[redacted]

We hope to receive your revised manuscript within four to eight weeks. If you cannot send it within this time, please let us know.

Sincerely,

Safia Danovi

Editor
Nature Genetics

Referee expertise:

Referee #1: scRNAseq, breast cancer

Referee #2: breast pathology

Referee #3: breast development

Reviewers' Comments:

Reviewer #1:

Remarks to the Author:

The authors provide a substantial cellular atlas of non-malignant breast cells, from 55 human donors, using single cell RNA-Seq. This cohort includes pre- and post-menopausal donors as well as donors with wild type and mutant BRCA1/2 germline mutation status. The authors focus on analysing differences between these two classes to identify cellular and molecular features associated with age & parity or germline BRCA1/2 gene status.

The major finding is that various cell types change with both factors, in particular the appearance of a CD8 T cell population with some features of exhaustion.

Overall the strength of the manuscript is as a resource for the community, due to its scale, which includes ~ 3x more total donors than prior work (Pal et al., 2021). However, it is worth noting that the number of healthy reduction mammoplasty donors (22 donors) is less than a recent report (28 donors) (Murrow et al., 2022). The data is high quality and processed and presented clearly.

The manuscript has two primary weaknesses: 1) A relatively shallow description of the data, making it unclear which of the findings are of most significance and novelty. (2) An absence of validation using orthogonal methods. Without these, it is difficult to be convinced by the reported changes.

Specific comments:

Key differences identified through analysis of scRNA-seq data must be validated using orthogonal methods- for instance IHC, immunofluorescence or in situ hybridisation. This should be relatively simple for many of the findings, for example changes in immune and vascular cells with parity. Expansion to a larger cohort would further test the validity of these findings.

Six BRCA1/2 mutant breast samples were taken from women with a contralateral cancer. There is a reasonable possibility that the presence of the cancer could cause systemic effect, influencing the the normal tissue. This may be especially significant in immune populations. Please test whether removal of these 6 samples from the analysis of BRCA1/2 tissue alters the findings.

Was BRCA1/2 testing conducted in all cases to confirm that the control group does not contain BRCA1/2 carriers?

How does this analysis account for differences in hormonal status of participants- which is impacted by estrous cycle and more importantly menopause.

The authors conclude that "Outside the epithelial compartment, there were very few significant differences". However the authors rely on a single method (Milo) to identify changes in cellular abundance. A second method, for instance based on more conventional high resolution clustering, should be employed to validate the changes (and lack of changes) observed using the graph-based method. This will also be valuable in biologically annotating the results, as several of the findings reveal changes in a subset of cells within a cluster(See Fig 3c and 3d). This raises the question of "what is the biological meaning and significance of these changes?"

Previous work has revealed the existence of an abnormal luminal progenitor, and an increased role for PR, RANK and RANKL, in premalignant breast tissue from BRCA1 carriers (RANK Ligand as a Potential Target for Breast Cancer Prevention in BRCA1-Mutation Carriers | Nature Medicine, n.d.). This should be investigated using this dataset

Under what conditions were samples frozen at the time of biobanking? Is there data on the effect of that freezing condition on cellular proportions or gene expression?

The authors suggest that 2 patient-specific clusters may represent cells 'indicative of early signs of disease'. This is an interesting suggestion but not supported by evidence. Please provide data, for instance copy number inference, to examine whether these cells have acquired genomic changes.

The changes in CD8 T cells in BRCA1/2 carriers are interesting, though again not explored deeply, beyond GSEA. These cells should be phenotyped more deeply to explore signatures of exhaustion/dysfunction, tissue residency, memory, etc. If this is the primary finding of significance, as suggested by the abstract, then substantially more work is required to support this finding. There are prior reports of T cell clonal expansion and exhaustion in the premalignant tissues of BRCA1 carriers(Yu et al., 2022),(Ogony et al., 2023), so the novelty of this finding needs to more carefully elaborated. Furthermore, the finding should be discussed in the context of other data reporting a role for BRCA1 in T cell function(Wu et al., 2023).

The Cellchat analysis (Figure 4) is speculative and disconnected to other findings.

Minor comments:

Prior work has labelled differentiated ER+ epithelial cells as "Mature Luminal"(Pal et al., 2021), rather than "Hormone sensing" as used here. Please explain the difference in terminology

Please comment on any divergence in the cell types detected in this study versus other studies such as (Pal et al., 2021) and also on cell types expected but not observed, such as neutrophils, adipocytes etc

Similarly, compare top cell type markers reported here to prior studies, to support the claim of "novel cell type markers"

The Milo method, terminology and depiction of its data (eg Fig 3b & 3d) should be explained at first use

It appears that many cellular subsets increase in fig 4a, without many related decreases. Are these increases at the expense of other cell types or is this an artefact of using scRNA-Seq to estimate changes in cellular abundance?

Murrow, L. M., Weber, R. J., Caruso, J. A., McGinnis, C. S., Phong, K., Gascard, P., Rabadam, G., Borowsky, A. D., Desai, T. A., Thomson, M., Tlsty, T., & Gartner, Z. J. (2022). Mapping hormone-regulated cell-cell interaction networks in the human breast at single-cell resolution. *Cell Systems*, 13(8), 644-664.e8. <https://doi.org/10.1016/j.cels.2022.06.005>

Ogony, J., Hoskin, T. L., Stallings-Mann, M., Winham, S., Brahmabhatt, R., Arshad, M. A., Kannan, N., Peña, A., Allers, T., Brown, A., Sherman, M. E., Visscher, D. W., Knutson, K. L., Radisky, D. C., & Degnim, A. C. (2023). Immune cells are increased in normal breast tissues of BRCA1/2 mutation carriers. *Breast Cancer Research and Treatment*, 197(2), 277–285. <https://doi.org/10.1007/s10549-022-06786-y>

Pal, B., Chen, Y., Vaillant, F., Capaldo, B. D., Joyce, R., Song, X., Bryant, V. L., Penington, J. S., Di Stefano, L., Tubau Ribera, N., Wilcox, S., Mann, G. B., kConFab, Papenfuss, A. T., Lindeman, G. J., Smyth, G. K., & Visvader, J. E. (2021). A single-cell RNA expression atlas of normal, preneoplastic and tumorigenic states in the human breast. *The EMBO Journal*, 40(11), e107333.

<https://doi.org/10.15252/embj.2020107333>

RANK ligand as a potential target for breast cancer prevention in BRCA1-mutation carriers | *Nature Medicine*. (n.d.). Retrieved May 29, 2023, from <https://www.nature.com/articles/nm.4118>

Wu, B., Qi, L., Chiang, H.-C., Pan, H., Zhang, X., Greenbaum, A., Stark, E., Wang, L.-J., Chen, Y., Haddad, B. R., Clagett, D., Isaacs, C., Elledge, R., Horvath, A., Hu, Y., & Li, R. (2023). BRCA1 deficiency in mature CD8+ T lymphocytes impairs antitumor immunity. *Journal for ImmunoTherapy of Cancer*, 11(2), e005852. <https://doi.org/10.1136/jitc-2022-005852>

Yu, X., Lin, W., Spirtos, A., Wang, Y., Chen, H., Ye, J., Parker, J., Liu, C. C., Wang, Y., Quinn, G., Zhou, F., Chambers, S. K., Lewis, C., Lea, J., Li, B., & Zheng, W. (2022). Dissection of transcriptome dysregulation and immune characterization in women with germline BRCA1 mutation at single-cell resolution. *BMC Medicine*, 20(1), 283. <https://doi.org/10.1186/s12916-022-02489-9>

Reviewer #2:

Remarks to the Author:

In this study Reed et al sought to characterize the cell composition of the normal human breast and define the changes associated with biological and environmental factors. The authors generated a scRNAseq Human Breast Cell Atlas through sequencing >800,000 cells from 55 donors, including patients who underwent reduction mammoplasties, BRCA1/BRCA2 germline mutation carriers, and patients with breast cancer family history who underwent prophylactic mastectomies. Epithelial and stromal/immune enriched compartments were isolated and scRNA-seq was conducted. The authors report changes in breast cell composition under physiologic conditions, such as an enrichment of a subset of epithelial subclusters with age, as well as changes in the cellular composition of epithelial, stromal and immune compartments with parity. The authors also investigated the effect of BRCA1 and

BRCA2 germline mutations on normal breast cell composition seeking to identify changes that might contribute to a higher risk of breast cancer development. Their analyses revealed shifts in the proportions of cell type subclusters in BRCA1/2 germline mutation carriers, such as decreased LP proportions in both HR-BR1 and -BR2 cohorts, as well as LP4, HS3 and BSL2 enrichment in BRCA1 samples. The authors also described changes in the immune compartment of BRCA1/2 donors, including an enrichment in IFNG+ T cells in BRCA1 donors and immune checkpoint receptors and PDL1 expression in both BRCA1 and BRCA2 cohorts. The authors should be commended for compiling this human breast cell atlas, as it might constitute a great resource for subsequent studies. The work the authors present, however, seems merely descriptive. Most importantly, there is no evidence to suggest that any of the findings described by the authors could contribute to a higher risk of breast cancer development or play roles in tumor initiation.

Major comments:

- Given that the authors have analyzed normal tissue of BRCA1/2 germline mutation carriers in cancer patients, it would be helpful to conduct the analysis of breast cancers of such cases to determine whether the changes in normal breast composition in HR-BR1 and HR-BR2 described by the authors mirror changes in tumor composition in BRCA1/2 cancers. Such analyses could link the authors' findings and carcinogenesis.

- There are assumptions made by the authors that are not quite supported by the data. For instance, the authors report changes in the endothelial compartment with shifts towards the EC angiogenic tip and LEC1 populations in HR-BR2 donors. The authors speculate that this indicates dysregulated vasculature and lymphatic development suggesting that HR-BR2 donors have a perturbed vascular microenvironment that predates aberrant epithelial expansion. There is, however, no evidence to support such contention.

Minor comments

- The authors conducted an analysis comparing HR-BR1/2 cohorts to donors with breast cancer predispositions of unknown origin. This analysis does not seem informative without further information from this last group of donors. Were these donors genetically profiled? Was there germline sequencing conducted? Was promoter methylation analysis of hereditary predisposition conducted.

Reviewer #3:

Remarks to the Author:

Summary of the key results, originality and significance

In this study, Reed, Pensa et al. perform single cell RNA sequencing (scRNA-seq) on samples from 55 donors that had undergone reduction mammoplasties or risk-reducing mammoplasties. These samples were sourced from the Human Breast Cell Atlas and, thus, include significant metadata and clinical annotation, which is both a strength and a significant consideration (see below). This large (800,000+) scRNA-seq dataset (+ metadata) may prove beneficial to future studies exploring breast cancer, and the submitted paper presents a very deep characterization of the dataset from which the authors draw hypotheses related to the effects of age, parity, and BRCA mutation status on the composition of the human breast. However, a significant consideration and concern is that many of

the discoveries are correlative or speculative, lacking validation to strengthen findings, and some comparisons may be underpowered and/or confounded by highly correlated classifications (e.g. age and parity) in the cohort of reduction mammoplasty samples.

Data, methodology, conclusions (robustness, validity, reliability), and suggested improvements

Figure 1 and Supplemental Figure 1 provide an overview of the scRNA-seq analysis of the samples from 55 donors, including metadata and clinical annotations. Aspects of this dataset are analyzed in the subsequent three Main Figures and Supplemental Figures. Supplemental Figure 1, and its inclusion of 13 variables, highlights a strength of the dataset. That is, the dataset can be potentially mined for insights into expression changes associated with age, parity, mutation status, and others. But, many of these variables are not universally annotated. When considering samples defined by these variables individually, the cohort sizes are rather limited. In addition, the dataset is not unique/original for many of the presented analyses, and, as shown in Figure 1, the scRNA-seq data are not validated in situ or complemented with other experimental analyses.

It is quite important to validate scRNA-seq data, and complement the generated hypotheses/observations with subsequent functional experiments. For example, Kessenbrock and colleagues generated and analysed a scRNA-seq dataset of samples from $n=11$ BRCA1 mutation carriers and $n=11$ non-carrier controls (Nature Genetics 2023, PMID: 36914836); importantly, the observations/hypotheses inferred from their scRNA-seq resource were validated and complemented by extensive in situ and functional in vitro and in vivo experiments. Visvader and colleagues generated and analysed a scRNA-seq dataset from a cohort of tissue specimens from normal or preneoplastic BRCA1+/- tissue ($n=28$), and $n=34$ tumors, including transcriptomes of 340,000 cells (EMBO J 2021, PMID: 33950524); importantly, the observations/hypotheses drawn from this scRNA-seq resource were also validated and complemented by in situ experiments. Finally, Brugge and colleagues (Dev Cell 2022, PMID: 35617956) integrate their scRNA-seq dataset (tissues from noncarriers ($n = 3$) and carriers of mutations in BRCA1 ($n = 6$), BRCA2 ($n = 6$), or RAD51C ($n = 1$)) with other complementary analyses, such as mass cytometry, high-throughput immunostaining and organoid cultures. These examples, and others, illustrate the importance of in situ validation and functional studies to complement scRNA-seq datasets. The current submission is lacking these critical validation and functional studies.

Figure 1 outlines a scRNA-seq data resource with samples from $n=11$ BRCA1 carriers, $n=11$ BRCA2 carriers, and $n=22$ reduction mammoplasty donors (assumed noncarrier). But, the inferred cell types and interactions extracted from these data are not validated in situ. In addition, the authors have missed an opportunity to integrate and compare with the publicly-available scRNA-seq datasets outlined above (PMIDs: 36914836, 33950524, 35617956, and others), as shown recently for four publicly-available scRNA-seq datasets of mouse mammary glands (Communications Biol 2021, PMID: 34079055).

Figure 2 describes a general overview of the cell types identified, as defined by the relative expression of various marker gene products. The output of these scRNA-seq datasets are obviously reliant on the inputted material, including the age, parity, BMI, risk, ethnicity, smoking, menstrual cycle phase, etc of the donor, but also the processing of the tissue (dissociation) and the selection of cells sequenced.

With regards to biological variables, the metadata captures age, parity, BMI, BRCA status, and other variables for some of the donor samples but this data is not uniformly captured throughout the cohort

(55 donors). Moreover, the metadata does not capture menstrual cycle phase, which is known to alter cell-cell interactions and transcriptional cell state (Cell Systems 2022, PMID: 35863345). Given the documented variability in samples isolated from the breast (person-to-person; age; parity; high-risk carrier), and the inconsistent capture of these variables across these samples, it is absolutely essential to validate the hypotheses that are inferred through this large scRNA-seq dataset.

Validation of this data could be accomplished via comparison with other published and complementary datasets as well as through the use of complementary methodologies and experiments. For example, the authors highlight LP5 as a small population of proliferating LP cells. A recent published scRNA-seq dataset (PMID: 36914836) indicates an increased proportion of proliferative epithelial cells in BRCA1 carriers. Is the LP5 population more prominent in BRCA1 carriers in this dataset? Conversely, fibroblasts from BRCA1 carriers are postulated to express a pre-cancer associated fibroblast phenotype characterized by MMP3 expression (PMID: 36914836). Thus, is the FB1 population identified in this dataset more prominent in BRCA1 carriers? The authors conclude that “all major cell types (are) represented in all donors regardless of BRCA, age or parity status”. Does this conclusion indicate that BRCA1 carrier samples are not distinguished by proliferative epithelial cells and a pre-CAF phenotype characterized by MMP3 expression? The agreement, or disagreement, between the cell populations inferred from this dataset and other related scRNA-seq data should be examined more completely.

In addition to the biological variables, the output of this dataset is dependent on the processing of the tissue and of the sequenced cells. The authors indicate that samples were pre-processed by the tissue bank, but it isn't clearly stated what this pre-processing entailed.

The manuscript does outline the gating strategy that discriminated (excluded) doublets, and Supplemental Figure 3 and 4 provide the individual FACS profiles for the samples. As is usual, “several QC and doublet calling steps” reduced the dataset from more than a million cells to 800,000 cells. After excluding doublets at two levels, the dataset was used to examine the expression of alveolar-like genes (Supplemental Figure 8) and the upregulation of milk-biosynthesis (Supplemental Figure 12). The dataset was also integrated with scRNA-seq data (PMID: 35091553) from milk derived epithelial cells (Supplemental Figure 17).

Importantly for this analysis, it is known that milk-producing alveolar cells in the pregnant breast, and pregnant mammary glands in other mammals, are binucleate, and appear to be fragile and reduced upon dissociation and FACS (Nat Commun 2016, PMID: 27102712). Therefore, it is at least possible that the dataset presented here, which is used to infer the proportion of cells expressing alveolar-like or milk-biosynthesis genes, is an underestimate of these populations due to physical- or bioinformatics-based criteria that eliminate or exclude binucleate cells; possibly, it is more likely that binucleate cells are not present in the non-pregnant human breast in average risk or in high risk donors. But, such variables associated with the processing, sequencing, and analysis of the scRNA-seq dataset demand that the conclusions inferred, and/or the hypotheses proposed, are validated by complementary methodologies.

Figure 3 exclusively considers the scRNA-seq data from 22 reduction mammoplasty samples to consider how age and parity impact the composition of the breast. According to data included in Supplementary Table 1, however, eight of eight young samples (under 29) are non-parous (with one unknown) and only one other sample (40 yo) is non-parous. Thus, the reported effect of parity (Fig. 3c,d) is likely to be more reflective of a young donor age, as the average age of non-parous donor samples (n=7, 25.9 yo) is significantly younger than the average age of the parous donor samples

(n=13, 46.6 yo).

The majority of the results section (5 of 9 pages) presents the findings of the analysis of scRNA-seq data comparing high risk (HR-BR1 or HR-BR2 carriers) and average risk (AR) donors. The initial finding is a large increase in proportions of immune cell subclusters CD4 T cell, CD8 T cell 1/2/3 and NK1/2 (Fig. 4a). This finding agrees with recent published IHC analysis of normal/benign breast lobules, which revealed elevated densities of immune cells (CD4+, CD8+, CD68+, and CD11c+) in HR prophylactic mastectomies (n= 19 BRCA1/2 carriers) (Breast Cancer Res Treatment 2023, PMID: 36380012). Further, the scRNA-seq dataset suggests a CD8 T-cell phenotype with a pro-inflammatory role; importantly, this is a testable hypothesis that the authors do not explore further. Rather, the authors highlight shifts in the hormone sensing populations correlating with expression of genes related to stress and heat shock, which they speculate may indicate populations that harbor DNA damage. Again, this is a testable hypothesis that is not explored further. The authors suggest evidence for upregulation of milk-biosynthesis or alveolar-like genes in both HR-BR1 and HR-BR2 cells. Again, this is a potentially interesting observation given the group's prior identification of abnormal alveologenesis-like differentiation in Brca1-mutant mice (Nat Commun 2021, PMID: 33686070), which also seems to occur in another Brca1-mutant mouse model (Biorxiv 2023, PMID: 37163037). But, this observation is not validated via in situ analyses. Disappointingly, another testable hypothesis/observation is left untested and without validation; that is, the scRNA-seq dataset suggests that "HR-BR2 donors have a perturbed vascular microenvironment", which could be a result of aberrant SEMA6 and ANGPT signalling. Taken together, a significant consideration and concern for this dataset is that many of the discoveries are correlative or speculative. The observations lack validation to strengthen the findings. And, some comparisons may be underpowered and/or confounded by highly correlated classifications (e.g. age and parity) in the cohort of reduction mammoplasty samples.

Author Rebuttal to Initial comments

Reviewer 1

Remarks to the Author:

The authors provide a substantial cellular atlas of non-malignant breast cells, from 55 human donors, using single cell RNA-Seq. This cohort includes pre- and post-menopausal donors as well as donors with wild type and mutant BRCA1/2 germline mutation status. The authors focus on analysing differences between these two classes to identify cellular and molecular features associated with age & parity or germline BRCA1/2 gene status. The major finding is that various cell types change with both factors, in particular the appearance of a CD8 T cell population with some features of exhaustion. Overall the strength of the manuscript is as a resource for the community, due to its scale, which includes ~ 3x more total donors than prior work (Pal et al., 2021). However, it is worth noting that the number of healthy reduction mammoplasty donors (22 donors) is less than a recent report (28 donors) (Murrow et al., 2022). The data is high quality and processed and presented clearly.

The manuscript has two primary weaknesses: 1) A relatively shallow description of the data, making it unclear which of the findings are of most significance and novelty. (2) An absence of validation using orthogonal methods. Without these, it is difficult to be convinced by the reported changes.

Specific comments:

Key differences identified through analysis of scRNA-seq data must be validated using orthogonal methods- for instance IHC, immunofluorescence or in situ hybridisation. This should be relatively simple for many of the findings, for example changes in immune and vascular cells with parity. Expansion to a larger cohort would further test the validity of these findings.

*We thank the reviewer for their comments. We have taken on board their suggestion and have performed orthogonal validation on tissue sections provided by the BCN tissue bank for 30 of the donors in our study. In particular, we used the Ultivue immunofluorescence multiplex staining kit to assess the protein expression and spatial distribution of CD8, CD4, PD1, PDL1, CD68, CD3, FOXP3 and panCK. In addition, we also performed Immunofluorescence for the immune check point markers TGIT, TIM3 and the NK marker GZMH on seven donors. We now include this new data in **Figures 4, 5 and Supplementary Figures 16, 17, 18, 19, 20, 21 and***

***22.** These new results validate and support the change in the immune compartment predicted by Milo (**Figure R1 below**). Moreover, the IF analysis also revealed new insights such as the PDL1 expression by CD68+ cells (**Figure 5a, b**) which was validated by further analysis of the scRNAseq data. Additionally, we attempted to stain our samples with markers of the vascular compartment ANGPT2, PXDN and CD31 however, the staining was inconsistent and non-specific. Due to the limited supply of tissue section from the same donors in our cohort we could not perform further optimisations or test other antibodies for these cell types. We have adjusted the text to include the new orthogonal validation methods (see text document highlighting edited text).*

Rebuttal Figure 1

Reed. et al

Rebuttal Figure 1

Ultivue staining showing panCK (white), CD8 (green), CD4 (red) and DAPI (blue) of breast sections from representative AR and HR donors as indicated. Scale bars represent 100 μ m.

Six BRCA1/2 mutant breast samples were taken from women with a contralateral cancer. There is a reasonable possibility that the presence of the cancer could cause systemic effect, influencing the normal tissue. This may be especially significant in immune populations. Please test whether removal of these 6 samples from the analysis of BRCA1/2 tissue alters the findings.

We agree with the reviewer that the inclusion of the 6 donors with contralateral cancer could be confounding, which is why they had been already excluded from the original Milo abundance testing. We now make it clear in the text that our Milo analysis did not include these 6 samples.

Was BRCA1/2 testing conducted in all cases to confirm that the control group does not contain BRCA1/2 carriers?

BRCA1/2 testing was not conducted on reduction mammoplasties as per clinical guidelines in the UK. BRCA testing is only offered to individuals post genetic counselling.

How does this analysis account for differences in hormonal status of participants- which is impacted by estrous cycle and more importantly menopause.

Data on hormonal status/oestrus and menopause status was not available for most of the donors in our cohort. We should also add that there is a debate in the field around the best methods to accurately

determine hormonal status retrospectively. We agree with the reviewer that this could be important in regulating cellular fluctuations, and it is information that should be collected from the donors in future prospective studies.

The authors conclude that “Outside the epithelial compartment, there were very few significant differences”. However, the authors rely on a single method (Milo) to identify changes in cellular abundance. A second method, for instance based on more conventional high resolution clustering, should be employed to validate the changes (and lack of changes) observed using the graph-based method. This will also be valuable in biologically annotating the results, as several of the findings reveal changes in a subset of cells within a cluster (See Fig 3c and 3d). This raises the question of “what is the biological meaning and significance of these changes?”

*We thank the reviewer for the suggestion. We have now taken two steps to answer this point. First, we integrated our dataset with six other published datasets to form the integrated HBCA (iHBCA) (See response to point below and **Figure 6**) and second, we performed pseudobulk differential abundance on the iHBCA. Overall, this analysis validated the results generated by the Milo analysis (**Figure 6a and Supplementary Figure 23 which is included below Figure R2**).*

Previous work has revealed the existence of an abnormal luminal progenitor, and an increased role for PR, RANK and RANKL, in premalignant breast tissue from BRCA1 carriers (RANK Ligand as a Potential Target for Breast Cancer Prevention in BRCA1-Mutation Carriers | Nature Medicine, n.d.). This should be investigated using this dataset

*We agree with the reviewer that RANK signalling appears to play an important role in both healthy and diseased breast physiology. Though it was not a focus of our study the HBCA is an excellent resource for the community to further explore these pathways. We provide some summary plots for RANK and RANKL expression trends within our data in **Figure R3** which supports an increase of RANK signalling in BRCA1 carriers (Nolan et al., 2016).*

Under what conditions were samples frozen at the time of biobanking?

One of the main reasons we chose to use the BCN tissue bank was that the processing of the tissue was standardised for all samples. Prior to freezing, fresh tissues were minced and digested O/N in 1 mg/mL Collagenase 1A and 1 mg/mL Hyaluronidase in an orbital shaker, washed 2-3 times and sedimented to allow for the separation between epithelial-enriched and stromal fractions, which are then cryostored.

Is there data on the effect of that freezing condition on cellular proportions or gene expression?

*We have now integrated our dataset with six other published datasets to form the integrated HBCA (iHBCA) (**Figure 6**). This integration revealed a large degree of consistency (**Figure R4**) in the cell types captured across the datasets despite having varying tissue collection, storage and preparation methods*

(Figure 6 and Supplementary Figure 23). However, the different tissue processing protocols contribute to a degree of variation in proportions of the cell types captured (Supplementary Figure 23).

Rebuttal Figure 2

(a) A schematic showing the curation of the iHBCA combining seven of the largest single cell RNA sequencing datasets for the breast. The diagram highlights the composition and sample heterogeneity captured by the iHBCA. The central plot shows a global uniform manifold approximation and projection (UMAP) representation of the dataset coloured by transferred subcluster annotations (see Fig 2) from the Human Breast Cell Atlas (HBCA). Annotation labels were mapped using CellTypist logistic regression models (see methods).

(b) Standard negative-binomial differential abundance analysis performed on the iHBCA for each of the comparisons previously explored using Milo on the Human Breast Cell Atlas (HBCA). Here we used the same base blocking terms as used in the milo analysis but with additional terms for dataset and live_sorting status (see methods for details).

Rebuttal Figure 3

Reed. et al

Rebuttal Figure 3

(a) Dotplot showing the expression of RANK ligand and receptor across the epithelial subclusters.

(b) Dotplots showing the expression of RANK ligand (TNFSF11) expression in luminal hormone sensing cells (left) and RANK receptor (TNFRSF11A) expression in luminal adaptive secretory precursor cells (right) across average-risk (AR), high-risk BRCA1 carrier (HR-BR1) and high-risk BRCA2 carrier (HR-BR2) donors.

The authors suggest that 2 patient-specific clusters may represent cells ‘indicative of early signs of disease’. This is an interesting suggestion but not supported by evidence. Please provide data, for instance copy number inference, to examine whether these cells have acquired genomic changes.

*We thank the reviewer for their comment and we have since explored the nature of these DDC cells in greater detail. We found that on top of mixed marker expression they showed increased breast cancer gene signatures for $HER2^+$ and basal-like subtypes with respect to the other epithelial cells (**Supplementary Figure 12**). Additionally, we have now performed copy number inference on these two donors (plus two mammoplasty controls) identifying putative CNV profiles in both DDC1 and DDC2 (**Supplementary Figure 13**).*

The changes in CD8 T cells in BRCA1/2 carriers are interesting, though again not explored deeply, beyond GSEA. These cells should be phenotyped more deeply to explore signatures of

exhaustion/dysfunction, tissue residency, memory, etc. If this is the primary finding of significance, as suggested by the abstract, then substantially more work is required to support this finding.

*We thank the reviewer for the comment and suggestion which we also addressed in the response to the first question. First, we further characterised the immune cell compartment gene signatures which led to a higher resolution cell identification, including characterising tissue residency and the functional role of the different CD8 T cell populations. Second, we used the Ultivue immunofluorescence multiplex staining kit to assess the protein expression and spatial distribution of CD8, CD4, PD1, PDL1, CD68, CD3, FOXP3 and panCK. In addition, we also performed Immunofluorescence for the immune check point markers TGIT, TIM3 and the NK marker GZMH on seven donors. We now include this new data in **Figures 4, 5 and Supplementary Figures 16, 17, 18, 19, 20, 21 and 22**. These new results validate and support the change in the immune compartment predicted by Milo (**Figure R1 below**). Moreover the IF analysis also revealed new insights such as the PDL1 expression by CD68+ cells (**Figure 5a,b**) which was validated by further analysis of the scRNAseq data.*

There are prior reports of T cell clonal expansion and exhaustion in the premalignant tissues of BRCA1 carriers (Yu et al., 2022), (Ogony et al., 2023), so the novelty of this finding needs to more carefully elaborated. Furthermore, the finding should be discussed in the context of other data reporting a role for BRCA1 in T cell function (Wu et al., 2023).

We thank the reviewer for highlighting these papers which used IHC and scRNAseq approaches on breast and fallopian tube samples respectively from BRCA carriers. Our scRNAseq and IF validations are in general agreement with these studies, that there is an increase in CD8+ T cells in these pre-malignant samples. Our study reveals for the first time the presence of an immune exhausted phenotype in the Breast which has not been reported before in the human. Interestingly, a similar phenotype was reported in the Fallopian tube (Yu X et al. 2022). Based on these findings, future studies looking at therapeutic cancer prevention in BRCA Carriers is merited and could offer BRCA carriers alternative clinical options to preventative surgeries. The reviewer also highlights the study by Wu et al. 2023 which utilised an Lck-Cre driven Brca1 knockout mouse to investigate T-cell function in response to a Brca WT tumor cell line. It is difficult for us to make direct comparisons as human samples are heterozygous germline BRCA carriers.

The Cellchat analysis (Figure 4) is speculative and disconnected to other findings.

*In light of the orthogonal validation and new data we include in the revised manuscript, we have now moved the CellChat analysis into the discussion (**Supplementary Figure 24**) and modified the text to reflect this (See text comparison document).*

Minor comments:

Prior work has labelled differentiated ER+ epithelial cells as “Mature Luminal”(Pal et al., 2021), rather than “Hormone sensing” as used here. Please explain the difference in terminology

*We recently participated in a global meeting with a number of authors representing most single cell RNA studies of the human breast hosted by CZI. The aim of the meeting was to find consensus on the terminology to use for the different cell types of the breast. We have now updated our manuscript accordingly, and added a guide table as a reference to minimise confusion between the interchangeable nomenclature used in the community (**Supplementary Table 2**).*

Please comment on any divergence in the cell types detected in this study versus other studies such as (Pal et al., 2021) and also on cell types expected but not observed, such as neutrophils, adipocytes etc. Similarly, compare top cell type markers reported here to prior studies, to support the claim of “novel cell type markers”

*We thank the reviewer for the suggestion. Our iHBCA revealed a large degree of consistency in the cell types captured across the datasets despite having varying tissue collection, storage and preparation methods (**Figure R4 Figure 6 and Supplementary Figure 23**). However, the different tissue processing protocols contribute to a degree of variation in proportions of the cell types captured, thus explaining most divergences observed.*

Rebuttal Figure 4

Reed. et al

Rebuttal Figure 4

A set of six confusion matrices showing the cell type/subcluster comparisons between each of the published datasets and our own subcluster annotations. Each cell (row A, column B) shows the percentage of cells of type A in the original dataset that are mapped to cell type B in our HBCA subcluster annotations. Note: LC1/2 cells from the Twigger dataset are cells thought to only appear in the lactating gland and are hence absent from the HBCA cohort causing their nonsensical logistic regression mapping.

The Milo method, terminology and depiction of its data (eg Fig 3b & 3d) should be explained at first use *We have amended the text to introduce and describe the Milo analysis.*

It appears that many cellular subsets increase in fig 4a, without many related decreases. Are these increases at the expense of other cell types or is this an artefact of using scRNA-Seq to estimate changes in cellular abundance?

We thank the reviewer for their comment. Not all the subclusters (rows) within the Milo beeswarm plots have the same number of cells. The visually dominating upregulation seen in the immune cells appears to skew the abundance results towards overall increases, but this is only the case as there are generally less total immune cells compared to epithelial and stromal compartments. Decreases seen in the epithelial and stromal compartments appear less extreme in proportion but will actually reflect greater absolute cell number changes in composition due to simply having more cells within these clusters.

Reviewer 2

Remarks to the Author:

In this study Reed et al sought to characterize the cell composition of the normal human breast and define the changes associated with biological and environmental factors. The authors generate >800,000 cells from 55 donors, including patients who underwent reduction mammoplasties, BRCA1/BRCA2 germline mutation carriers, and patients with breast cancer family history who underwent prophylactic mastectomies. Epithelial and stromal/immune enriched compartments were isolated and scRNA-seq was conducted. The authors report changes in breast cell composition under physiologic conditions, such as an enrichment of a subset of epithelial subclusters with age, as well as changes in the cellular composition of epithelial, stromal and immune compartments with parity. The authors also investigated the effect of BRCA1 and BRCA2 germline mutations on normal breast cell composition seeking to identify changes that might contribute to a higher risk of breast cancer development. Their analyses revealed shifts in the proportions of cell type subclusters in BRCA1/2 germline mutation carriers, such as decreased LP proportions in both HR-BR1 and -BR2 cohorts, as well as LP4, HS3 and BSL2 enrichment in BRCA1 samples. The authors also described changes in the immune compartment of BRCA1/2 donors, including an enrichment in IFNG+ T cells in BRCA1 donors and immune checkpoint receptors and PDL1 expression in both BRCA1 and BRCA2 cohorts. The authors should be commended for compiling this human breast cell atlas, as it might constitute a great resource for subsequent studies. The work the authors present, however, seems merely descriptive. Most importantly, there is no evidence to suggest that any of the findings described by the authors could contribute to a higher risk of breast cancer development or play roles in tumor initiation.

Major comments:

- Given that the authors have analyzed normal tissue of BRCA1/2 germline mutation carriers in cancer patients, it would be helpful to conduct the analysis of breast cancers of such cases to determine whether the changes in normal breast composition in HR-BR1 and HR-BR2 described by the authors mirror changes in tumor composition in BRCA1/2 cancers. Such analyses could link the authors' findings and carcinogenesis.

Most of the tissues obtained were collected from donors prior to any tumour development. Correlating cellular changes that occur precancer to the fully developed tumour is challenging in the absence of longitudinal samples – which is not feasible to obtain from the breast. Longitudinal/time-resolved samples are feasible in the mouse which we have recently performed on a Brca1 mouse model (Bach, Pensa et al. 2021). We learnt from our mouse analysis that changes observed in the tissue precancer may not necessarily be present as the disease progresses, and likely reflect transitional cellular changes that precede the establishment of the tumour phenotype. A direct comparison of precancer tissue to the advanced disease might mask pre-malignant events that could be potentially exploited for early detection and treatment of these HR-donors. We have compared the gene expression profile of the two donor derived clusters (DDC1 and DDC2) to the gene expression profiles previously obtained from

scRNAseq of tumour samples and found some correlation with the various breast cancer subtypes (Supplementary Figure 12).

- There are assumptions made by the authors that are not quite supported by the data. For instance, the authors report changes in the endothelial compartment with shifts towards the EC angiogenic tip and LEC1 populations in HR-BR2 donors. The authors speculate that this indicates dysregulated vasculature and lymphatic development suggesting that HR-BR2 donors have a perturbed vascular microenvironment that predates aberrant epithelial expansion. There is, however, no evidence to support such contention.

*We thank the reviewer for their comments. We have taken on board their suggestion and have performed orthogonal validation on tissue sections provided by the BCN tissue bank for 30 of the donors in our study. In particular, we used the Ultivue immunofluorescence multiplex staining kit to assess the protein expression and spatial distribution of CD8, CD4, PD1, PDL1, CD68, CD3, FOXP3 and panCK. In addition, we also performed Immunofluorescence for the immune check point markers TGIT, TIM3 and the NK marker GZMH on seven donors. We now include this new data in **Figures 4, 5 and Supplementary Figures 16, 17, 18, 19, 20, 21 and***

22. *These new results validate and support the change in the immune compartment predicted by Milo (Figure R1 below). Moreover the IF analysis also revealed new insights such as the PDL1 expression by CD68+ cells (Figure 5a,b) which was validated by further analysis of the scRNAseq data. Finally, we attempted to stain our samples with markers of the vascular compartment ANGPT2, PXDN and CD31 however, the staining was inconsistent and non-specific. Due to the limited supply of tissue section from the same donors in our cohort we could not perform further optimisations or test other antibodies for these cell types. We have adjusted the text to include the new orthogonal validation methods and shift emphasis towards validated results (see highlighted text document).*

Minor comments

- The authors conducted an analysis comparing HR-BR1/2 cohorts to donors with breast cancer predispositions of unknown origin. This analysis does not seem informative without further information from this last group of donors. Were these donors genetically profiled? Was there germline sequencing conducted? Was promoter methylation analysis of hereditary predisposition conducted.

We thank the reviewer for their comment. We agree that such analysis would benefit from additional information on the mutational status of the samples. However, we have not been able to perform this type of tests at this stage. We have removed the analysis of donors with breast cancer predisposition of unknown origin from the paper to focus instead on other results. Our integrated HBCA dataset (Figure 6) contains many more of these donor samples, and thus could provide a particularly useful resource to explore the impact of non-BRCA related hereditary breast cancer risk in future studies.

Reviewer 3

Remarks to the Author:

Summary of the key results, originality and significance

In this study, Reed, Pensa et al. perform single cell RNA sequencing (scRNA-seq) on samples from 55 donors that had undergone reduction mammoplasties or risk-reducing mammoplasties. These samples were sourced from the Human Breast Cell Atlas and, thus, include significant metadata and clinical annotation, which is both a strength and a significant consideration (see below). This large (800,000+) scRNA-seq dataset (+ metadata) may prove beneficial to future studies exploring breast cancer, and the submitted paper presents a very deep characterization of the dataset from which the authors draw hypotheses related to the effects of age, parity, and BRCA mutation status on the composition of the human breast. However, a significant consideration and concern is that many of the discoveries are correlative or speculative, lacking validation to strengthen findings, and some comparisons may be underpowered and/or confounded by highly correlated classifications (e.g. age and parity) in the cohort of reduction mammoplasty samples.

Data, methodology, conclusions (robustness, validity, reliability), and suggested improvements

Figure 1 and Supplemental Figure 1 provide an overview of the scRNA-seq analysis of the samples from 55 donors, including metadata and clinical annotations. Aspects of this dataset are analyzed in the subsequent three Main Figures and Supplemental Figures. Supplemental Figure 1, and its inclusion of 13 variables, highlights a strength of the dataset. That is, the dataset can be potentially mined for insights into expression changes associated with age, parity, mutation status, and others. But, many of these variables are not universally annotated. When considering samples defined by these variables individually, the cohort sizes are rather limited. In addition, the dataset is not unique/original for many of the presented analyses, and, as shown in Figure 1, the scRNA-seq data are not validated in situ or complemented with other experimental analyses.

We thank the reviewer for the comments. Due to the nature of the samples used in the dataset, which belong to a tissue bank, we are not able to collect retrospectively the missing metadata from all donors. We agree with the reviewer that this is important information that should be collected for prospective studies. In terms of uniqueness of the dataset, compared to other atlasing efforts we have two distinguishing points: 1) we have sequenced significantly more cells per donor than any other dataset published (particularly across such wide variety of samples) and 2) to our knowledge this is the first scRNAseq dataset to include BRCA2 donors analysed at such depth. These two unique features enable us to perform high resolution differential abundance analysis such as Milo and identify rare cell clusters such as DDC1 and DDC2.

It is quite important to validate scRNA-seq data, and complement the generated hypotheses/observations with subsequent functional experiments. For example, Kessenbrock and colleagues generated and analysed a scRNA-seq dataset of samples from n=11 BRCA1 mutation carriers and n=11 non-carrier controls (Nature Genetics 2023, PMID: 36914836); importantly, the

observations/hypotheses inferred from their scRNA-seq resource were validated and complimented by extensive in situ and functional in vitro and in vivo experiments. Visvader and colleagues generated and analysed a scRNA-seq dataset from a cohort of tissue specimens from normal or preneoplastic BRCA1+/- tissue (n=28), and n=34 tumors, including transcriptomes of 340,000 cells (EMBO J 2021, PMID: 33950524); importantly, the observations/hypotheses drawn from this scRNA-seq resource were also validated and complimented by in situ experiments. Finally, Brugge and colleagues (Dev Cell 2022, PMID: 35617956) integrate their scRNA-seq dataset (tissues from noncarriers (n = 3) and carriers of mutations in BRCA1 (n = 6), BRCA2 (n = 6), or RAD51C (n = 1)) with other complementary analyses, such as mass cytometry, high-throughput immunostaining and organoid cultures. These examples, and others, illustrate the importance of in situ validation and functional studies to complement scRNA-seq datasets. The current submission is lacking these critical validation and functional studies.

*We thank the reviewer for their comments. We have taken on board their suggestion and have performed orthogonal validation on tissue sections provided by the BCN tissue bank for 30 of the donors in our study. In particular, we used the Ultivue immunofluorescence multiplex staining kit to assess the protein expression and spatial distribution of CD8, CD4, PD1, PDL1, CD68, CD3, FOXP3 and panCK. In addition, we also performed Immunofluorescence for the immune check point markers TGIT, TIM3 and the NK marker GZMH on seven donors. We now include this new data in **Figures 4, 5 and Supplementary Figures 16, 17, 18, 19, 20, 21 and***

22. *These new results validate and support the change in the immune compartment predicted by Milo (**Figure R1 below**). Moreover the IF analysis also revealed new insights such as the PDL1 expression by CD68+ cells (**Figure 5a,b**) which was validated by further analysis of the scRNAseq data. Finally, we attempted to stain our samples with markers of the vascular compartment ANGPT2, PXDN and CD31 however, the staining was inconsistent and non-specific. Due to the limited supply of tissue section from the same donors in our cohort we could not perform further optimisations or test other antibodies for these cell types. We have adjusted the text to include the new orthogonal validation methods (see highlighted text document).*

Figure 1 outlines a scRNA-seq data resource with samples from n=11 BRCA1 carriers, n=11 BRCA2 carriers, and n=22 reduction mammoplasty donors (assumed noncarrier). But, the inferred cell types and interactions extracted from these data are not validated in situ. In addition, the authors have missed an opportunity to integrate and compare with the publicly- available scRNA-seq datasets outlined above (PMIDs: 36914836, 33950524, 35617956, and others), as shown recently for four publicly-available scRNA-seq datasets of mouse mammary glands (Communications Biol 2021, PMID: 34079055).

*We thank the reviewer for the comment and suggestion. We have now integrated our dataset with six other published datasets to form the integrated HBCA (iHBCA). This integration revealed a large degree of consistency (**Figure R4**) in the cell types captured across the datasets despite having varying tissue collection, storage and preparation methods (**Figure 6 and Supplementary Figure 23**). However, the different tissue processing protocols contribute to a degree of variation in proportions of the cell types*

captured. In addition, we performed pseudobulk differential abundance on the iHBCA which validated the results generated by the Milo analysis (Supplementary Figure 23 which is included below Figure R2). Figure 2 describes a general overview of the cell types identified, as defined by the relative expression of various marker gene products. The output of these scRNA-seq datasets are obviously reliant on the inputted material, including the age, parity, BMI, risk, ethnicity, smoking, menstrual cycle phase, etc of the donor, but also the processing of the tissue (dissociation) and the selection of cells sequenced. With regards to biological variables, the metadata captures age, parity, BMI, BRCA status, and other variables for some of the donor samples but this data is not uniformly captured throughout the cohort (55 donors). Moreover, the metadata does not capture menstrual cycle phase, which is known to alter cell-cell interactions and transcriptional cell state (Cell Systems 2022, PMID: 35863345). Given the documented variability in samples isolated from the breast (person-to-person; age; parity; high-risk carrier), and the inconsistent capture of these variables across these samples, it is absolutely essential to validate the hypotheses that are inferred through this large scRNA-seq dataset. Validation of this data could be accomplished via comparison with other published and complementary datasets as well as through the use of complementary methodologies and experiments. Data on hormonal status/oestrus and menopause status was not available for most of the donors in our cohort. We should also add that there is a debate in the field around the best methods to accurately determine hormonal status retrospectively. We agree with the reviewer that this could be important in regulating cellular fluctuations, and it is information that should be collected from the donors in future prospective studies. In terms of variability of cell captured our integrated iHBCA revealed a large degree of consistency (Figure R4) in the cell types captured across the datasets despite having varying tissue collection, storage and preparation methods (Figure 6 and Supplementary Figure 23). However, the different tissue processing protocols contribute to a degree of variation in proportions of the cell types captured. The previously mentioned IF stainings and pseudobulk differential abundance testing in the iHBCA cohort also serve as validation of our analysis through complementary methodologies.

For example, the authors highlight LP5 as a small population of proliferating LP cells. A recent published scRNA-seq dataset (PMID: 36914836) indicates an increased proportion of proliferative epithelial cells in BRCA1 carriers. Is the LP5 population more prominent in BRCA1 carriers in this dataset?

We thank the reviewer for their suggestion, however, we found the proportions of proliferating LASP cells (LASP5) across the cohort too small to make robust comparisons between average-risk and high-risk donors (~0.03% of all cells with most originating from a few specific donors). These LASP5 cells show expression profiles more closely matching the G2/M phase of cell cycle, whereas the study mentioned (Nee et al., 2023) characterised proportions of S phase cells in BRCA1 carriers. This could be another avenue which the iHBCA with its increased sample size could assist future studies into the proliferation dynamics of the breast.

Conversely, fibroblasts from BRCA1 carriers are postulated to express a pre-cancer associated fibroblast phenotype characterized by MMP3 expression (PMID: 36914836). Thus, is the FB1 population identified in this dataset more prominent in BRCA1 carriers? The authors conclude that “all major cell types (are)

represented in all donors regardless of BRCA, age or parity status". Does this conclusion indicate that BRCA1 carrier samples are not distinguished by proliferative epithelial cells and a pre-CAF phenotype characterized by MMP3 expression? The agreement, or disagreement, between the cell populations inferred from this dataset and other related scRNA-seq data should be examined more completely. *In our dataset, the FB population regardless of MMP3 expression had a slight enrichment in both BRCA1 and BRCA2 donors (Figure 2, Figure 4 and Supplementary Figure 15). We don't see a particularly strong enrichment of MMP3 high FB therefore, it is possible that there is a general upregulation of MMP3 in FBs as opposed to expansion of MMP3 high 'pre-CAF' FB. Our data is not in disagreement with Nee et al., but it is worth noting that in the Nee et al. paper they identify the differential gene expression of MMP3 on all FBs while in our analysis we look at the subpopulations of FB and use the higher resolution Milo approach to identify which of these are enriched in BRCA1/2 donors.*

In addition to the biological variables, the output of this dataset is dependent on the processing of the tissue and of the sequenced cells. The authors indicate that samples were pre-processed by the tissue bank, but it isn't clearly stated what this pre-processing entailed.

We thank the reviewer for the comment. One of the main reasons we chose to use the BCN tissue bank was that the processing of the tissue was standardised for all samples. Prior to freezing, fresh tissues were minced and digested O/N in 1 mg/mL Collagenase 1A and 1 mg/mL Hyaluronidase in an orbital shaker, washed 2-3 times and sedimented to allow for the separation between epithelial-enriched and stromal fractions, which are then cryostored.

The manuscript does outline the gating strategy that discriminated (excluded) doublets, and Supplemental Figure 3 and 4 provide the individual FACS profiles for the samples. As is usual, "several QC and doublet calling steps" reduced the dataset from more than a million cells to 800,000 cells. After excluding doublets at two levels, the dataset was used to examine the expression of alveolar-like genes (Supplemental Figure 8) and the upregulation of milk- biosynthesis (Supplemental Figure 12). The dataset was also integrated with scRNA-seq data (PMID: 35091553) from milk derived epithelial cells (Supplemental Figure 17).

Importantly for this analysis, it is known that milk-producing alveolar cells in the pregnant breast, and pregnant mammary glands in other mammals, are binucleate, and appear to be fragile and reduced upon dissociation and FACS (Nat Commun 2016, PMID: 27102712). Therefore, it is at least possible that the dataset presented here, which is used to infer the proportion of cells expressing alveolar-like or milk-biosynthesis genes, is an underestimate of these populations due to physical- or bioinformatics-based criteria that eliminate or exclude binucleate cells; possibly, it is more likely that binucleate cells are not present in the non- pregnant human breast in average risk or in high risk donors. But, such variables associated with the processing, sequencing, and analysis of the scRNA-seq dataset demand that the conclusions inferred, and/or the hypotheses proposed, are validated by complementary methodologies.

We thank the reviewer for highlighting this point. While it is possible we may have lost some binucleate alveolar cells due to their fragile nature during sample preparation, the exclusion of doublets via gating strategy only applies to the LP population that was enriched by FACS. However, for each of the donors we also sequenced non-sorted epithelia, therefore they should have been sequenced in relevant proportions here. The initial round of computational doublet detection was completed using Scrublet looks at simulated doublet profiles of cells combined within the dataset and scores cells based on their similarity to these doublet profiles. Milk producing (binuclear or not) alveolar cells are a distinct cell type, so they should display a unique transcriptomic profile and thus have low scrublet scores generally. So, it is unlikely that they would be removed in this stage. The second round of doublet detection combined the scrublet scores with fine clustering and marker gene expression manually, again making it unlikely to result in the exclusion of alveolar cells. We have now specified in the relevant part of the text that doublet calling was computational doublet calling.

Figure 3 exclusively considers the scRNA-seq data from 22 reduction mammoplasty samples to consider how age and parity impact the composition of the breast. According to data included in Supplementary Table 1, however, eight of eight young samples (under 29) are non-parous (with one unknown) and only one other sample (40 yo) is non-parous. Thus, the reported effect of parity (Fig. 3c,d) is likely to be more reflective of a young donor age, as the average age of non-parous donor samples (n=7, 25.9 yo) is significantly younger than the average age of the parous donor samples (n=13, 46.6 yo).

We thank the reviewer for this important point. In 2021, the average age of mothers who gave birth in England and Wales was 30.9 years (Office of National Statistics UK) which is reflected in the cohort of donors in our study. However, we use statistical methods to account for the independent impact of ageing. If the nulliparous samples completely overlapped with young donors and parous with the older donors and our statistical approach failed, then we would expect our tests to show parity have a near identical effect to that of ageing. As this is not observed (in fact they are more so opposing despite the parous donors being on average older than nulliparous donors), this is unlikely to be an issue.

The majority of the results section (5 of 9 pages) presents the findings of the analysis of scRNA-seq data comparing high risk (HR-BR1 or HR-BR2 carriers) and average risk (AR) donors. The initial finding is a large increase in proportions of immune cell subclusters CD4 T cell, CD8 T cell 1/2/3 and NK1/2 (Fig. 4a). This finding agrees with recent published IHC analysis of normal/benign breast lobules, which revealed elevated densities of immune cells (CD4+, CD8+, CD68+, and CD11c+) in HR prophylactic mastectomies (n= 19 BRCA1/2 carriers) (Breast Cancer Res Treatment 2023, PMID: 36380012). Further, the scRNA-seq dataset suggests a CD8 T-cell phenotype with a pro-inflammatory role; importantly, this is a testable hypothesis that the authors do not explore further. Rather, the authors highlight shifts in the hormone sensing populations correlating with expression of genes related to stress and heat shock, which they speculate may indicate populations that harbor DNA damage. Again, this is a testable hypothesis that is not explored further. The authors suggest evidence for upregulation of milk-biosynthesis or alveolar-like genes in both HR-BR1 and HR-BR2 cells. Again, this is a potentially interesting observation given the group's prior identification of abnormal alveologenesis-like differentiation in Brca1-mutant mice (Nat

Commun 2021, PMID: 33686070), which also seems to occur in another Brca1-mutant mouse model (Biorxiv 2023, PMID: 37163037). But, this observation is not validated via in situ analyses. Disappointingly, another testable hypothesis/observation is left untested and without validation; that is, the scRNA-seq dataset suggests that “HR-BR2 donors have a perturbed vascular microenvironment”, which could be a result of aberrant SEMA6 and ANGPT signalling. Taken together, a significant consideration and concern for this dataset is that many of the discoveries are correlative or speculative. The observations lack validation to strengthen the findings. And, some comparisons may be underpowered and/or confounded by highly correlated classifications (e.g. age and parity) in the cohort of reduction mammoplasty samples.

*We thank the reviewer for their comments. We have taken on board their suggestion and have performed orthogonal validation on tissue sections provided by the BCN tissue bank for 30 of the donors in our study. In particular, we used the Ultivue immunofluorescence multiplex staining kit to assess the protein expression and spatial distribution of CD8, CD4, PD1, PDL1, CD68, CD3, FOXP3 and panCK. In addition, we also performed Immunofluorescence for the immune check point markers TGIT, TIM3 and the NK marker GZMH on seven donors. We now include this new data in **Figures 4, 5 and Supplementary Figures 16, 17, 18, 19, 20, 21 and 22**. These new results validate and support the change in the immune compartment predicted by Milo (**Figure R1 below**). Moreover, the IF analysis also revealed new insights such as the PDL1 expression by CD68+ cells (**Figure 5a, b**) which was validated by further analysis of the scRNAseq data. Finally, we attempted to stain our samples with markers of the vascular compartment ANGPT2, PXDN and CD31 however, the staining was inconsistent and non-specific. Due to the limited supply of tissue section from the same donors in our cohort we could not perform further optimisations or test other antibodies for these cell types. We have adjusted the text to include the new orthogonal validation methods (see highlighted text document). We thank reviewer 1 and 3 for highlighting papers which used Immunofluorescence and scRNAseq approaches on breast and fallopian tube samples respectively from BRCA carriers. Our scRNAseq and IF validations are in general agreement with these studies, that there is an increase in CD8+ T cells in these pre-malignant samples. Our study reveals for the first time the presence of an immune exhausted phenotype in the Breast which has not been reported before in the human. Interestingly, a similar phenotype was reported in the Fallopian tube (Yu X et al. 2022). Based on these findings, future studies looking at therapeutic cancer prevention in BRCA Carriers is merited and could offer BRCA donors alternative clinical options to preventative surgeries. The reviewer also highlights the study by Wu et al. 2023 which utilised an Lck-Cre driven Brca1 knockout mouse to investigate T-cell function in response to a Brca WT tumor cell line. It is difficult for us to make direct comparisons as human samples are heterozygous germline BRCA donors. Finally, with regards to the expression of CSN2, we have previously shown evidence of these cells in our 2021 paper (PMID: 33686070), in which we isolated RNA from an independent cohort of AR and HR-BR1 samples from the same tissue bank, and showed by qPCR and Bulk-RNAseq expression of CSN2 in a proportion of the donors analysed.*

Decision Letter, first revision:

22nd Nov 2023

Dear Professor Khaled,

Your Article, "A human breast cell Atlas enables mapping of homeostatic cellular shifts in the adult breast" has now been seen by 3 referees. You will see from their comments below that while they find your work of interest, some important points are raised. We are interested in the possibility of publishing your study in Nature Genetics, but would like to consider your response to these concerns in the form of a revised manuscript before we make a final decision on publication.

Overall, all three reviewers consider the work to have been improved by your revisions. Reviewer #3 continues to think that further validation could have been performed but while we'd welcome these data, their absence would not preclude our interest in the paper. However, we consider that other requests for more robust quantification and methodological clarifications should all be addressed in full. Reviewer #1 has also requested that your control group be tested for BRCA1/2 germline mutations and if you are able to provide these data, please include them. Please address R1's other queries.

We therefore invite you to revise your manuscript taking into account these reviewer and editor comments. Please highlight all changes in the manuscript text file. At this stage we will need you to upload a copy of the manuscript in MS Word .docx or similar editable format.

*2) If you have not done so already please begin to revise your manuscript so that it conforms to our Article format instructions, available [here](http://www.nature.com/ng/authors/article_types/index.html). Refer also to any guidelines provided in this letter.

[redacted]

We hope to receive your revised manuscript within four to eight weeks. If you cannot send it within this time, please let us know.

Sincerely,

Safia Danovi
Editor
Nature Genetics

Reviewers' Comments:

Reviewer #1:

Remarks to the Author:

Thankyou for the revisions, they have gone a long way towards addressing my concerns

Regarding BRCA1/2 testing of patients, I understand that this is not clinically indicated, but the researchers could nonetheless test the samples to ensure that the control group is not contaminated with sample/s that carry unknown BRCA1/2 mutations.

The authors do not appear to have quantitated the changes in cellular proportion between AR and HR cases, as detected by IF. Staining is shown in Fig 4C and Fig S16-S19 but no quantitation with statistical analysis is provided.

The number of individuals contributing to each datapoint should be stated in each legend- for example in Fig 5c, how many patients are represented?

A comment should be made in the discussions to highlight that hormonal status/oestrus and menopause status was not measured and could influence the results for each individual

Reviewer #2:

Remarks to the Author:

The authors have addressed my comments satisfactorily. Thank you.

Reviewer #3:

Remarks to the Author:

The authors have revised their manuscript with the following major changes: (1) the authors have performed multiplex immunofluorescence imaging of markers for T cell populations (CD4+, CD8+, and FoxP3+), macrophages (CD68+), and immune checkpoint proteins (PDL1, PD1) in FFPE samples. These orthologous validation experiments agree with their scRNA-seq data. The multiplex IF imaging strengthens their finding that an altered immune infiltrate is present in BRCA carriers, and is suggestive that immune cell exhaustion/immune escape may be present in the HR-BRCA donor population; (2) the authors integrate their dataset with 6 other large scRNA-seq datasets to compile a large breast atlas, which will be a valuable resource to researchers. The authors have also uploaded the integrated Human Breast Cell Atlas dataset on cellxgene; and (3) the authors have further investigated the two unique donor derived clusters through the prediction of CNVs in individual cells from these clusters, which were then compared to CNV profiles from molecular breast cancer subtypes. Their new analysis strengthens their conclusion that these were not erroneously annotated clusters that were composed of low-quality cells.

These major changes have improved the manuscript but many conclusions drawn in this manuscript still remain untested because the depth of orthogonal validation remains rather shallow.

Major comments

1. The authors cite the limited supply of tissue from the same donors as a justification for not optimizing the detection of markers for the vascular compartment. However, the authors propose that these alterations are generalizable and, thus, these changes can/should be confirmed in tissues from other donors.
2. The authors propose age-related changes to the LSAP compartment inferred by Milo analysis of their scRNA seq dataset; but the authors do not validate this observation.
3. The authors do perform immunofluorescence analysis of immune cells in samples from HR-BR1, HR-BR2 and AR patients. But, the authors do not provide measurements of these cellular subsets and rather rely on qualitative assessments (Figure 4b,c). As mentioned in a prior review, the levels of immune cells have been measured in tissues from BR1/2 carriers (median cell counts per mm²) and found to be significantly higher (PMID: 36380012). Thus, the current analysis does not provide further insight to that which has already been published. It is a lost opportunity that the authors chose not to

1. Measure absolute cell numbers (counts per mm²) and 2. Extract spatial information about the topology of these immune cells with respect to the epithelia and other immune components.
4. The authors identify changes in the expression of immune checkpoint markers in the immune populations in HR-BR tissues suggestive of potential immune exhaustion, which the authors present as potential evidence that "immune escape mechanisms could manifest in non-cancerous tissues during very early stages of tumor initiation". This is an attractive hypothesis that, nevertheless, remains speculative. The authors should test this hypothesis and provide evidence from functional in vitro assays with immune cells isolated from HR-BR1/2 carrier samples.

Minor comments:

1. The multiplex immunofluorescence analysis does not quantify the density of T cells or macrophages. Additionally, there is no quantification of the immune check point markers in the supplemental figures. Finally, the authors might consider to determine spatial topologies for the immune cells.
2. In the multiplex immunofluorescence images, some show large ducts while others show lobules. Does immune cell density differ between sites? If so, did the authors ensure that lobules are being compared to lobules, and ducts be compared to ducts?

Author Rebuttal, first revision:

Reviewer #1:

Remarks to the Author:

Thank you for the revisions, they have gone a long way towards addressing my concerns Regarding BRCA1/2 testing of patients, I understand that this is not clinically indicated, but the researchers could nonetheless test the samples to ensure that the control group is not contaminated with sample/s that carry unknown BRCA1/2 mutations.

We have checked with the tissue bank regarding this point and to the best of their knowledge the donors in our control group were not BRCA1/2 patients. However, there is an ethical and legal barrier to us sequencing these samples to check the BRCA1/2 mutations. The approved ethics for the tissue bank does not allow us to sequence the samples to specifically look for BRCA1/2 mutations.

The authors do not appear to have quantitated the changes in cellular proportion between AR and HR cases, as detected by IF. Staining is shown in Fig 4C and Fig S16-S19 but no quantitation with statistical analysis is provided.

We thank the reviewer for this comment. We have now quantified the IFs as requested and the trends observed generally confirm our Milo analysis, such as the increased proportions of CD8+ in HR-BR1 vs AR. In addition, this enabled us to extract spatial information from our results. For example, we observed that the enrichment of PD1+ cells observed in the HR donors compared to AR, is specifically localised in non-epithelial areas. The new results have now been added to the manuscript (**Figure 5e and Supplementary Figure 23**). One caveat of quantifying a single section of a tissue is that it is not representative of the whole tissue and does not give the same statistical power of scRNAseq. The scRNAseq data was obtained after processing frozen epithelial-enriched and stromal-enriched samples obtained from the tissue bank. Prior to freezing, the tissue is manually minced and then digested in parallel batches. These are then pooled back together and mixed prior to freezing into vials. We

therefore believe the single cell data will reflect the general tissue composition more accurately than one single tissue section. This is an important issue for all spatial genomics studies which we believe need to include multiple serial sections across the tissue in their analysis to be truly representative. The number of individuals contributing to each datapoint should be stated in each legend- for example in Fig 5c, how many patients are represented?
This information had now been added to the Figure legends.

A comment should be made in the discussions to highlight that hormonal status/oestrus and menopause status was not measured and could influence the results for each individual

This has now been added to the discussion.

Reviewer #2:

Remarks to the Author:

The authors have addressed my comments satisfactorily. Thank you.
We thank the reviewer for their endorsement of our study.

Reviewer #3:

Remarks to the Author:

The authors have revised their manuscript with the following major changes: (1) the authors have performed multiplex immunofluorescence imaging of markers for T cell populations (CD4+, CD8+, and FoxP3+), macrophages (CD68+), and immune checkpoint proteins (PDL1, PD1) in FFPE samples. These orthologous validation experiments agree with their scRNA-seq data. The multiplex IF imaging strengthens their finding that an altered immune infiltrate is present in BRCA carriers, and is suggestive that immune cell exhaustion/immune escape may be present in the HR-BRCA donor population; (2) the authors integrate their dataset with 6 other large scRNA-seq datasets to compile a large breast atlas, which will be a valuable resource to researchers. The authors have also uploaded the integrated Human Breast Cell Atlas dataset on cellxgene; and (3) the authors have further investigated the two unique donor derived clusters through the prediction of CNVs in individual cells from these clusters, which were then compared to CNV profiles from molecular breast cancer subtypes. Their new analysis strengthens their conclusion that these were not erroneously annotated clusters that were composed of low-quality cells.

These major changes have improved the manuscript but many conclusions drawn in this manuscript still remain untested because the depth of orthogonal validation remains rather shallow.

Major comments

1. The authors cite the limited supply of tissue from the same donors as a justification for not optimizing the detection of markers for the vascular compartment. However, the authors propose that these

alterations are generalizable and, thus, these changes can/should be confirmed in tissues from other donors.

We agree with the Reviewer that it would be important to validate these findings, however, in the absence of unique markers for these cell types and material to perform further testing, this was not possible at this stage. We however, adjusted the text in light of these results to highlight the validated immune cell fluctuations, and removed any speculation related to the findings related to the vascular compartment.

2. The authors propose age-related changes to the LSAP compartment inferred by Milo analysis of their scRNA seq dataset; but the authors do not validate this observation.

LASPs are proportionately very rare in the breast, therefore it would be very hard to find these cells in sufficient numbers to allow for the validation of these observations. In addition, one section is a very small area of the tissue and therefore might not be representative of the whole tissue.

3. The authors do perform immunofluorescence analysis of immune cells in samples from HR- BR1, HR-BR2 and AR patients. But, the authors do not provide measurements of these cellular subsets and rather rely on qualitative assessments (Figure 4b,c). As mentioned in a prior review, the levels of immune cells have been measured in tissues from BR1/2 carriers (median cell counts per mm²) and found to be significantly higher (PMID: 36380012). Thus, the current analysis does not provide further insight to that which has already been published.

It is a lost opportunity that the authors chose not to 1. Measure absolute cell numbers (counts per mm²) and 2. Extract spatial information about the topology of these immune cells with respect to the epithelia and other immune components.

We thank the reviewer for this comment. We have now quantified the IFs as requested and the trends observed generally confirm our Milo analysis, such as the increased proportions of CD8+ in HR-BR1 vs AR. In addition, this enabled us to extract spatial information from our results. For example, we observed that the enrichment of PD1+ cells observed in the HR donors compared to AR, is specifically localised in non-epithelial areas. The new results have now been added to the manuscript (**Figure 5e and Supplementary Figure 23**). One caveat of quantifying a single section of a tissue is that it is not representative of the whole tissue and does not give the same statistical power of scRNAseq. The scRNAseq data was obtained after processing frozen epithelial-enriched and stromal-enriched samples obtained from the tissue bank. Prior to freezing, the tissue is manually minced and then digested in parallel batches. These are then pooled back together and mixed prior to freezing into vials. We therefore believe the single cell data will reflect the general tissue composition more accurately than one single tissue section. This is an important issue for all spatial genomics studies which we believe need to include multiple serial sections across the tissue in their analysis to be truly representative.

4. The authors identify changes in the expression of immune checkpoint markers in the immune populations in HR-BR tissues suggestive of potential immune exhaustion, which the authors present as potential evidence that “immune escape mechanisms could manifest in non-cancerous tissues during very early stages of tumor initiation”. This is an attractive hypothesis that, nevertheless, remains

speculative. The authors should test this hypothesis and provide evidence from functional in vitro assays with immune cells isolated from HR- BR1/2 carrier samples.

We thank the Reviewer for this comment; however, we believe this is beyond the scope of this manuscript. We are confident the rich datasets we include in our study and the newly added integrated Human Breast Cell Atlas would be the foundation of many investigations by the breast cancer research community. We will be looking in the future at the exploration of some of these findings in experimental systems.

Minor comments:

1. The multiplex immunofluorescence analysis does not quantify the density of T cells or macrophages. Additionally, there is no quantification of the immune check point markers in the supplemental figures. Finally, the authors might consider to determine spatial topologies for the immune cells. Please see response to major comment n. 3

2. In the multiplex immunofluorescence images, some show large ducts while others show lobules. Does immune cell density differ between sites? If so, did the authors ensure that lobules are being compared to lobules, and ducts be compared to ducts?

We thank the reviewer for this comment. We have not performed such spatial quantification. We believe drawing spatial conclusions based on a single section representing a very small fraction of the whole tissue would be error prone.

Decision Letter, second revision:

14th Dec 2023

Dear Dr Khaled,

Thank you for submitting your revised manuscript "A human breast cell Atlas enables mapping of homeostatic cellular shifts in the adult breast" (NG-A62214R1). We assessed your revisions in-house and I'm pleased to say that we'll be happy in principle to publish it in Nature Genetics, pending minor revisions to satisfy our editorial and formatting guidelines.

Sincerely,

Safia Danovi
Editor
Nature Genetics

Final Decision Letter:

9th Feb 2024

Dear Dr Khaled,

I am delighted to say that your manuscript "A single-cell atlas enables mapping of homeostatic cellular shifts in the adult human breast" has been accepted for publication in an upcoming issue of Nature Genetics.

Your paper will be published online after we receive your corrections and will appear in print in the next available issue. You can find out your date of online publication by contacting the Nature Press Office (press@nature.com) after sending your e-proof corrections.

Before your paper is published online, we shall be distributing a press release to news organizations worldwide, which may very well include details of your work. We are happy for your institution or funding agency to prepare its own press release, but it must mention the embargo date and Nature Genetics. Our Press Office may contact you closer to the time of publication, but if you or your Press

Office have any enquiries in the meantime, please contact press@nature.com.

Please note that *Nature Genetics* is a Transformative Journal (TJ). Authors may publish their research with us through the traditional subscription access route or make their paper immediately open access through payment of an article-processing charge (APC). Authors will not be required to make a final decision about access to their article until it has been accepted. Find out more about Transformative Journals

Authors may need to take specific actions to achieve compliance with funder and institutional open access mandates. If your research is supported by a funder that requires immediate open access (e.g. according to Plan S principles) then you should select the gold OA route, and we will direct you to the compliant route where possible. For authors selecting the subscription publication route, the journal's standard licensing terms will need to be accepted, including [a href="https://www.nature.com/nature-portfolio/editorial-policies/self-archiving-and-license-to-publish](https://www.nature.com/nature-portfolio/editorial-policies/self-archiving-and-license-to-publish). Those licensing terms will supersede any other terms that the author or any third party may assert apply to any version of the manuscript.

If you have not already done so, we invite you to upload the step-by-step protocols used in this manuscript to the Protocols Exchange, part of our on-line web resource, natureprotocols.com. If you complete the upload by the time you receive your manuscript proofs, we can insert links in your article that lead directly to the protocol details. Your protocol will be made freely available upon publication of

your paper. By participating in natureprotocols.com, you are enabling researchers to more readily reproduce or adapt the methodology you use. Natureprotocols.com is fully searchable, providing your protocols and paper with increased utility and visibility. Please submit your protocol to <https://protocolexchange.researchsquare.com/>. After entering your nature.com username and password you will need to enter your manuscript number (NG-A62214R2). Further information can be found at <https://www.nature.com/nature-portfolio/editorial-policies/reporting-standards#protocols>

Sincerely,

Safia Danovi
Editor
Nature Genetics